# Brightening triplet excitons enable high-performance white-light emission in organic small molecules via integrating $n-\pi^*/\pi-\pi^*$ transitions

Qing Yang[1], Xinyi Yang [1]✉, Yixuan Wang[1], Yunfan Fei[2], Fang Li[2], Haiyan Zheng[2], Kuo Li[2], Yibo Han [3], Takanori Hattori[4], Pinwen Zhu[1], Shuaiqiang Zhao[5], Leiming Fang [6], Xuyuan Hou[1], Zhaodong Liu[1], Bing Yang[5] & Bo Zou [1]✉

Luminescent materials that simultaneously embody bright singlet and triplet excitons hold great potential in optoelectronics, signage, and information encryption. However, achieving high-performance white-light emission is severely hampered by their inherent unbalanced contribution of fluorescence and phosphorescence. Herein, we address this challenge by pressure treatment engineering via the hydrogen bonding cooperativity effect to realize the mixture of $n-\pi^*/\pi-\pi^*$ transitions, where the triplet state emission was boosted from 7% to 40% in isophthalic acid (IPA). A superior white-light emission based on hybrid fluorescence and phosphorescence was harvested in pressure-treated IPA, and the photoluminescence quantum yield was increased to 75% from the initial 19% (blue-light emission). In-situ high-pressure IR spectra, X-ray diffraction, and neutron diffraction reveal continuous strengthening of the hydrogen bonds with the increase of pressure. Furthermore, this enhanced hydrogen bond is retained down to the ambient conditions after pressure treatment, awarding the targeted IPA efficient intersystem crossing for balanced singlet/triplet excitons population and resulting in efficient white-light emission. This work not only proposes a route for brightening triplet states in organic small molecules, but also regulates the ratio of singlet and triplet excitons to construct high-performance white-light emission.

White-light emission based on organic materials has attracted particular attention on account of their unique merits in diversified practical applications including panel displays and light-emitting diodes[1–4]. However, their maximum internal quantum efficiency is limited to 25% based on quantum spin statistics, as triplet excitons are typically dark owing to their spin-forbidden nature[5,6]. To conquer this issue, hybrid fluorescence and phosphorescence multicolor-encoded materials with the advantageous spin-flipping process are applied to brighten the "dark" triplet states to boost the emission efficiency, as well as to enable color variability[7,8]. Notably, since triplet excitons are prone to

[1]State Key Laboratory of Superhard Materials, Synergetic Extreme Condition High-Pressure Science Center, College of Physics, Jilin University, Changchun, China. [2]Center for High Pressure Science and Technology Advanced Research, Beijing, China. [3]Wuhan National High Magnetic Field Center and School of Physics, Huazhong University of Science and Technology, Wuhan, China. [4]J-PARC Center, Japan Atomic Energy Agency, Tokai, Ibaraki, Japan. [5]State Key Laboratory of Supramolecular Structure and Materials, College of Chemistry, Jilin University, Changchun, China. [6]Institute of Nuclear Physics and Chemistry, China Academy of Engineering Physics, Mianyang, China. ✉e-mail: yangxinyi@jlu.edu.cn; zoubo@jlu.edu.cn

dissipate through non-radiative transition and luminescence quenching at room temperature, realizing highly efficient white-light emission in these phosphors still remains a grand challenge. Conventional approaches focus on organometallic complexes with large spin-orbital coupling (SOC) to allow efficient bright triplet states emission, nevertheless, often suffering unsustainable resources and high toxicity[9,10]. In light of this, purely organic room temperature phosphorescence (RTP) has been increasingly regarded as a promising alternative based on its low toxicity, flexibility, and limited processability[11–14].

Efficient intersystem crossing (ISC) and suppressing non-radiative dissipation have been employed as key concerns to achieve superior RTP[15,16]. In this regard, the research community has concentrated on halogen bonding[17,18], $n$–$\pi$ transition[19], intermolecular charge transfer[20], and steric effects[21] to improve hyperfine coupling (HFC), induce large SOC and minimize singlet-triplet energy gap ($\Delta E_{ST}$) for accelerated ISC process. Suppressing non-radiative dissipation via creating a rigid environment[22–24] has sparked vigorous investigations into manipulating phosphorescence performance[25]. Recently, stimuli-responsive RTP emission was proposed by grinding or thermal annealing the non-compact packing materials[26]. As an independent thermodynamic parameter and a clean tuning knob, pressure can be used for continuous modification in both intermolecular interactions and energy bands[27–34]. Therefore, we speculate that the high-pressure engineering might directly modulate the intrinsic SOC and introduce multifarious singlet-triplet energy separation to regulate the ISC process for white-light emission.

Herein, we employ a facile physical strategy, pressure treatment engineering, to achieve high-performance white-light emission with chromaticity coordinates of (0.28, 0.36) from the initial blue-light-emitting isophthalic acid (IPA), where the photoluminescence quantum yield (PLQY) increased from 19% to 75%. The pressure treatment engineering tremendously boosts the mixture of $n$–$\pi^*$/$\pi$–$\pi^*$ transition configurations in targeted triplet states via strengthening hydrogen bonds, which triggers an enhancement of SOC favoring efficient ISC. Moreover, attributed to the enhanced intermolecular (O–H···O) electrostatic environment, as well as a pressure-treated misaligned $\pi$–$\pi$ stacking configuration, the narrowed $\Delta E_{ST}$ was also harvested in the targeted sample, which further facilitates the ISC process. A high fraction of triplet excitons is then generated to balance the distribution of fluorescence and phosphorescence, thus endowing the targeted IPA with superior hybrid fluorescence and phosphorescence white-light emission at ambient conditions. Meanwhile, the non-radiative dissipation of singlet and triplet excitons are substantially suppressed by the enhanced hydrogen bonds, empowering the treated IPA to carry distinguishable higher fluorescent and phosphorescent efficiency.

## Results

### In situ high-pressure optical properties of IPA

The single IPA molecules in the twisted layers were arranged in zigzag forms via hydrogen-bond interaction and the twisted layers were offset face-to-face staked along the normal direction (Fig. 1a). The intrinsic intermolecular hydrogen-bond interactions and $\pi$–$\pi$ staking motifs endow the solid IPA materials with RTP emission by SOC and HFC effect[20,35,36]. Under the 355 nm laser excitation, a blue-light emission with PLQY of 19% was observed based on the pristine configuration, and accompanied by a green RTP emission after turning off the laser excitation (Fig. 1b). The initial PL spectrum of IPA was composed of duel emission bands with a primary strong one centered at 402 nm and another weak one in the range of 450–720 nm under the 355 nm laser excitation (Supplementary Fig. 1a). After turning off the laser excitation, the lower-energy band was measured located ~530 nm (Supplementary Fig. 1b). The time-resolved PL decays of these two species indicate their lifetimes of 6.81 ns (402 nm) and 888.6 ms (530 nm),

respectively (Supplementary Fig. 2). These properties reflected that the short-lived emission should originate from singlet excited state, while the long-lived one belonged to triplet excitons emission. The corresponding fluorescence quantum yield ($\Phi_F$) and phosphorescence quantum yield ($\Phi_P$) of the pristine sample separately reached 12% and 7% (Supplementary Fig. 3a).

Considering the fluorescence-dominated weak blue-light emission and limited phosphorescence efficiency of the initial sample, high-pressure processing was expected to alter the singlet/triplet excitons distribution by directly regulating the intermolecular interactions and staking configuration. In situ high-pressure PL measurements were thus carried out up to 18.0 GPa with a pressure transmitting medium (PTM) of silicone oil. The emission of the sample experienced a stark persistent increase with the maximum value exceeding 9.7 times from 1 atm to 9.0 GPa and then followed by a decrease in intensity (Fig. 1c and Supplementary Fig. 1c, e). A weak white-light emission with chromaticity coordinates of (0.30, 0.38) was observed at 18.0 GPa (Supplementary Fig. 4). With the decrease of pressure, the PL intensity gradually enhanced (Supplementary Fig. 1d, e). Upon releasing pressure to the ambient conditions, the targeted sample displayed a high-performance white-light emission with PLQY of 75% and chromaticity coordinates of (0.28, 0.36) (Fig. 1d, e). We also explored the high-pressure optical evolutions with different PTMs, including liquid argon (Supplementary Fig. 5) and nitrogen (Supplementary Fig. 6). Their PL evolutions under high pressure still experienced a similar trend to that of silicon oil, where the white-light emission was also harvested after releasing pressure completely. These similar results manifested that the presently used PTMs (silicone oil, liquid argon, and nitrogen) could not influence the current results: the PL enhancement under high pressure and the harvest of white-light emission after pressure treatment. Furthermore, the treated sample still displayed a high-performance white-light emission after ~400 days (Supplementary Fig. 7). In order to further explore the influence of pressure treatment on RTP properties, the relevant characterization of RTP emission was performed. We measured the yellowish-green afterglow emission at ~573 nm (Supplementary Fig. 8) after turning off the UV lamp, which could be captured by the naked eye even lasting for 8 s (Fig. 1b, Supplementary Fig. 9, and Supplementary Movie). The time-resolved PL decay measurements of the targeted IPA revealed a longer phosphorescence lifetime of 900.2 ms (573 nm) (Supplementary Fig. 2). The fluorescence peak was fitted at 481 nm (Fig. 1f). $\Phi_P$ and $\Phi_F$ unexpectedly increased to 40% and 35% from 7% (1 atm) and 12% (1 atm) (Supplementary Fig. 3b and Supplementary Table 1).

We further analyzed the in situ high-pressure ultraviolet-visible (UV-vis) absorption profiles to elucidate the electronic behaviors (Supplementary Fig. 10). At ambient conditions, the absorption peak was located at 295 nm, where the absorption edge was at ~307 nm. With the increase of pressure to ~21.9 GPa, the absorption edge exhibited a continuous redshift. After pressure was completely released, the absorption peak at ~317 nm in the targeted sample redshifted by about 22 nm compared with the pristine sample. We then estimated the bandgap changes before and after pressure treatment. We found that the bandgap narrowed from 4.20 eV to 3.91 eV, illustrating the reduction of relevant excited-states energies in targeted IPA.

### Structural evolution of IPA under pressure

Following the optical changes above, we devoted efforts to the corresponding structural evolution to explore the mechanism. We have conducted in situ high-pressure infrared (IR) spectra in Supplementary Fig. 11a. The vibration mode at 1696 cm$^{-1}$ belonged to the stretching vibration of C=O bonds[37], which redshifted by 14 cm$^{-1}$ below 20.0 GPa. This suggested that the O–H···O=C hydrogen bonds were continuously strengthened with the increase of pressure[38,39]. More importantly, the C=O stretching vibrational mode in the pressure-treated IPA was

 

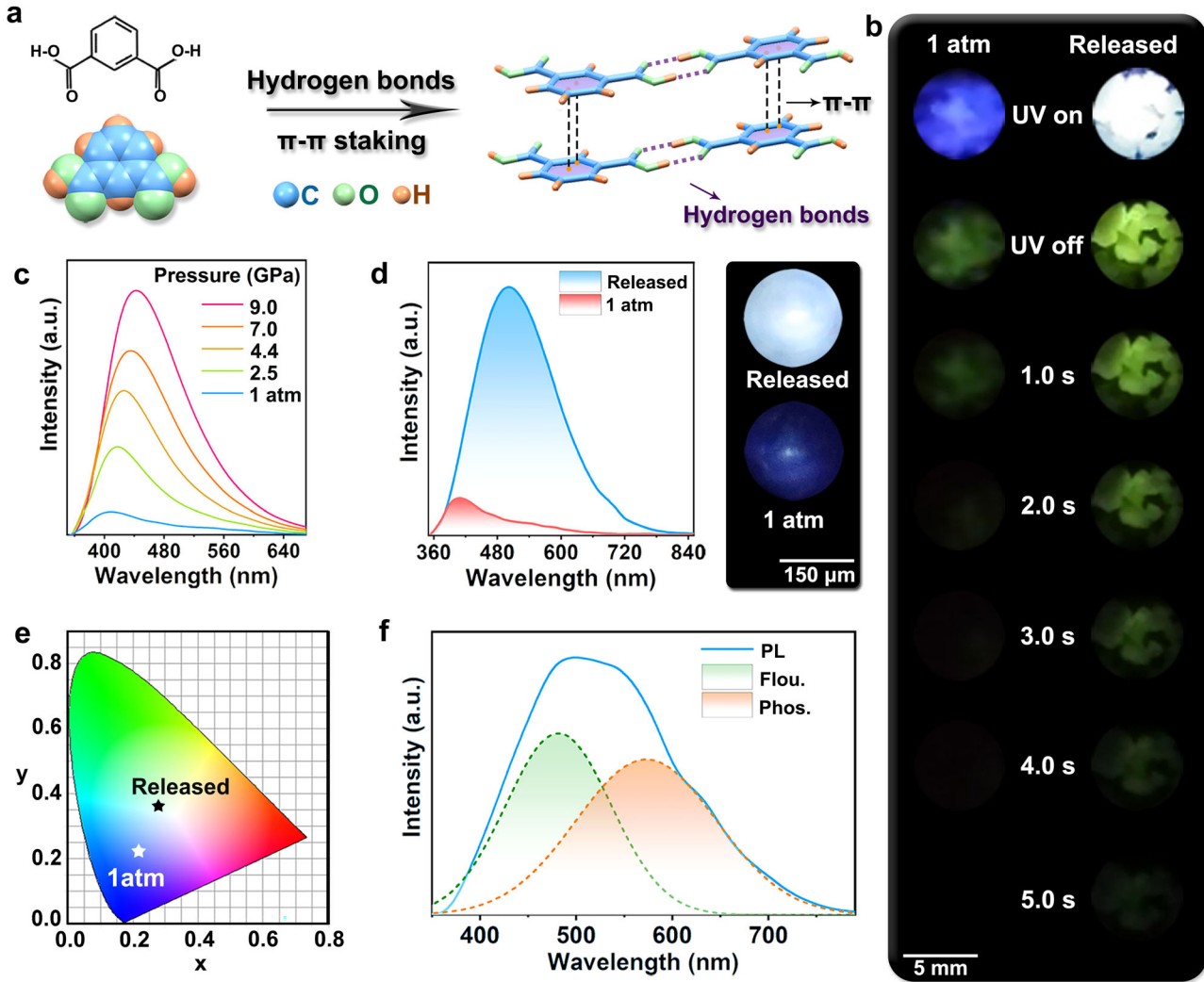

**Fig. 1 | The molecular structure and photoluminescence (PL) properties of IPA.**
**a** The crystal structure of IPA. **b** Photographs of the pristine and pressure-treated IPA taken at different time intervals before and after turning off the laser excitation (355 nm) at ambient conditions. The photographs (UV off) were taken immediately after turning the UV light off. A mass of quenched products (decompression from 20.0 GPa) was harvested by high-pressure experiments in a Walker-type large-volume press. **c** PL spectra of IPA upon compression to 9.0 GPa using symmetric diamond anvil cell (DAC) devices under the 355 nm laser excitation. **d** PL spectra of IPA (left) at ambient conditions and after pressure was released from 18.0 GPa. The corresponding PL photographs (right) were taken in DAC devices. **e** Changes of chromaticity coordinates at ambient conditions (0.22, 0.22) and after releasing pressure from 18.0 GPa (0.28, 0.36). **f** PL (blue solid line), fitting fluorescence (green dashed line), and fitting phosphorescence (orange dashed line) spectra of the targeted IPA treated by the Walker-type large-volume press.

redshifted by 6 cm$^{-1}$ compared to the original state (Fig. 2a). This demonstrates that the enhanced hydrogen bonds have survived down to the ambient conditions after pressure treatment. It is noted that the hydrogen-bond interaction in IPA refers primarily to the electrostatic interaction between oxygen and hydrogen atoms. Such enhancement of the hydrogen bonds indicates that the proton donor of hydrogen atoms has a stronger attraction to the lone pair (*n*) electrons on oxygen atoms. Thus, the *n*-orbitals can be stabilized to the lower energies by the enhanced electrostatic environments[36].

In order to further explore the structural variation upon compression and decompression, we tracked in situ high-pressure angle-dispersive synchrotron X-ray diffraction (ADXRD) experiments from 1 atm to 19.5 GPa (Supplementary Fig. 11b). Upon compression, all Bragg diffraction peaks shifted continuously along the large 2-theta direction and no new peaks appeared. It indicated that the structure did not undergo phase transformation during the whole compression process. We performed Rietveld refinements of ADXRD patterns to obtain detailed lattice parameters upon compression and decompression (Supplementary Fig. 12). The initial IPA possesses monoclinic symmetry

with space group $P2_1/c$ at ambient conditions. As the pressure increased, lattice parameters *a*, *b*, and *c* decreased. The $\beta$ also underwent continuous reduction upon compression, which appeared to be in favor of a lessened spatial overlap of the π-stacked motif (Fig. 2b).

Given that organic IPA consists solely of light-weight elements such as C, H, and O, which displayed a weaker scatter ability in ADXRD experiments. We further performed in situ time-of-flight (TOF) neutron diffraction to delve more deeply into the structural evolution upon compression and decompression (Supplementary Figs. 13–16, Supplementary Table 2, and Fig. 2c). As pressure was increased from ambient conditions to 19.8 GPa, all diffraction patterns moved toward smaller *d*-spacing, indicating lattice compression (Supplementary Fig. 13 and Fig. 2c). In addition, several peaks of (033), (110), and (022) planes in the recovered neutron spectrum located at smaller *d*-values (the blue and magenta lines in Fig. 2d). We also conducted experiments on the recovered sample after removing the gasket (the green and orange lines in Fig. 2d). The consistent shift in the (033), (110), and (022) planes indicated that these changes were indeed derived from the pressure treatment engineering (Supplementary Fig. 17). Notably, the

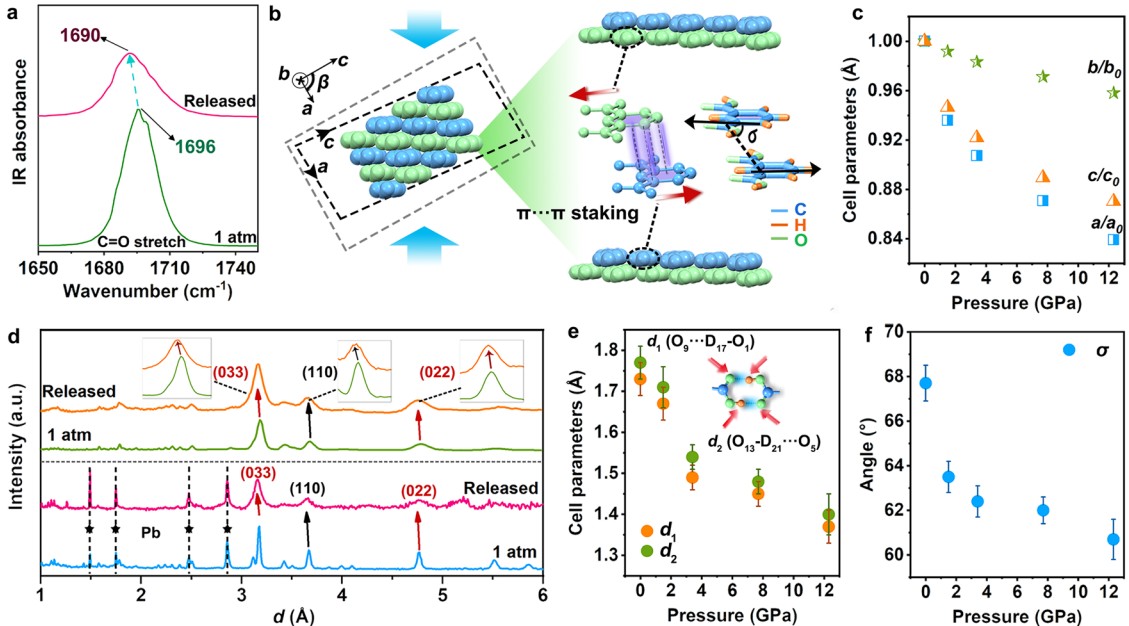

**Fig. 2 | Crystal structure evolution upon compression and decompression. a** IR spectra of IPA in the region of C=O stretching vibrational mode ν (C=O) at ambient conditions and after pressure was released from 20.0 GPa. **b** Molecular packing along the *b*-axis (left). The schematic diagram of the parallel misalignment angle (*σ*) of IPA (right). **c** The compression rate of three lattice constants (*a*, *b*, *c*) at different pressures. **d** Time-of-flight (TOF) neutron diffraction patterns of IPA-*d₆*. The TOF neutron diffraction patterns of the pristine (blue line) and recovered (magenta line) samples were collected at BL11 PLANET in the MLF at J-PARC. The asterisks indicate the peaks of the Pb marker. The neutron diffraction patterns of the pristine (green line) and recovered (orange line) samples were collected at the High-pressure neutron diffractometer (Fenghuang) at the CMRR neutron science platform. **e** Pressure-dependent hydrogen-bond distances $D_{17}$⋯$O_9$ ($d_1$) and $D_{21}$⋯$O_5$ ($d_2$) evolution of IPA. **f** Pressure-dependent *σ* evolution of IPA. The $d_1$, $d_2$, and *σ* of IPA-*d₆* were determined by Rietveld refinement of neutron diffraction patterns. The error bars in (**e**) and (**f**) represent the standard error in the Rietveld refinement.

orientations of (033) and (022) lattice planes are almost perpendicular to the direction of hydrogen bonds, the evolution of their *d*-spacings is closely related to the change of hydrogen-bond distance (Supplementary Fig. 18). Therefore, the reduced *d*-spacings of these almost vertical planes in the targeted IPA could further confirm the result of strengthened hydrogen bonds after pressure treatment[38]. In addition, attributed to such reduced hydrogen-bond distances, coupled with the intensified parallel-displaced π staking arrangement, the *d*-spacing of the (110) plane also exhibited a decrease in the recovered sample.

In this regard, we move forward to investigate the changes of the correlative intermolecular distances $D_{17}$⋯$O_9$ ($d_1$), $D_{21}$⋯$O_5$ ($d_2$), and the parallel misalignment angle (*σ*) to assess the hydrogen bonds and π-stacked configuration of IPA upon compression and decompression. We found that both $d_1$ and $d_2$ displayed a decreasing trend with the increase of pressure, suggesting the strengthened hydrogen bonds at high pressure[40] (Fig. 2e). Moreover, the $d_1$ of the recovered sample decreased from 1.73(4) Å to 1.49(5) Å and $d_2$ decreased from 1.77(4) Å to 1.69(5) Å. Therefore, the hydrogen bonds in the recovered IPA were stronger than those ones before pressure treatment. The *σ* also tended to decrease upon compression (Fig. 2f), and the lesser *σ* value of 65.7(9)° rendered a lessened spatial overlap of the π-stacked motif in the targeted IPA (Supplementary Table 3). Meanwhile, the intermolecular perpendicular distance between benzene rings was compressed from 3.47 Å to 3.43 Å. The intermolecular distance $d_{(C⋯C)}$ between benzene rings did not change significantly, which varied from 3.77(5) Å to 3.75(5) Å (Supplementary Fig. 19). In this regard, the electronic coupling between the π-stacked molecules should following these changes and then contribute to the targeted phosphorescence enhancement.

## SOC, HFC, and *ΔE*_ST effect of IPA

Notably, the decreased hydrogen-bond distances would drive strengthened electrostatic interactions between the proton and the lone pair electrons, leading to the consequent changes in the ISC process[36]. The coupling of the radical ion pairs (RIP) excitons derived from π–π staked IPA molecules is expected to be influenced by the alteration of π–π stacking arrangement[20]. Besides, the parallel-displaced configuration also played a significant role in minimizing energy difference *ΔE*(π, π*) of the bonding and antibonding π-type molecular orbitals[41], thereby resulting in the alteration of *ΔE*_ST. Drawing upon the aforementioned considerations, we systematically analyze the influence of these structural changes on the ISC process from the following three aspects: SOC, HFC, and *ΔE*_ST. To begin with, the narrowed hydrogen-bond distance was considered a vital factor for efficient ISC. To decipher the mechanism, we performed the SOC coefficients (*ξ*) and the natural transition orbitals (NTOs) on the singlet and triplet states based on the pristine and pressure-treated IPA (Fig. 3a and Supplementary Table 4). Before pressure treatment, the $S_1$ and high-lying triplet ($T_5$ and $T_6$) states mainly exhibited the transition character of ¹(*n*, π*) and ³(*n*, π*) (Supplementary Fig. 20). The low-lying $T_n$ (1 ≤ *n* ≤ 4) states were primarily assigned to the transition character of ³(π, π*). After pressure treatment, the $T_5$ and $T_6$ states underwent a noticeable change from ³(*n*, π*) in the original IPA to the mixture of ³(*n*, π*) and ³(π, π*) in pressure-treated IPA (Fig. 3b). According to the El-Sayed's rule[42], a superior SOC between $S_1$ to $T_n$ (5 ≤ *n* ≤ 6) states was expected to be achieved in the targeted sample for efficient ISC. In fact, the calculated *ξ*($S_1$-$T_5$) and *ξ*($S_1$-$T_6$) for targeted IPA were 29.18 cm⁻¹ and 7.71 cm⁻¹, respectively. These values were both larger than those of pristine IPA (with *ξ*($S_1$ − $T_5$) of 28.53 cm⁻¹ and *ξ*($S_1$ − $T_6$) of 0.08 cm⁻¹). Such enhanced SOC could efficiently promote the ISC process for enhanced RTP emission. In addition, we found that the pressure treatment also endowed *ξ* enhancements of other triplet states (Supplementary Table 4). This further boosts the population of triplet excitons, allowing for substantial utilization of triplet-state energies in treated IPA. Therefore, the effective mixture of *n*−π*/π−π* transition configurations tremendously promotes the increase of SOC

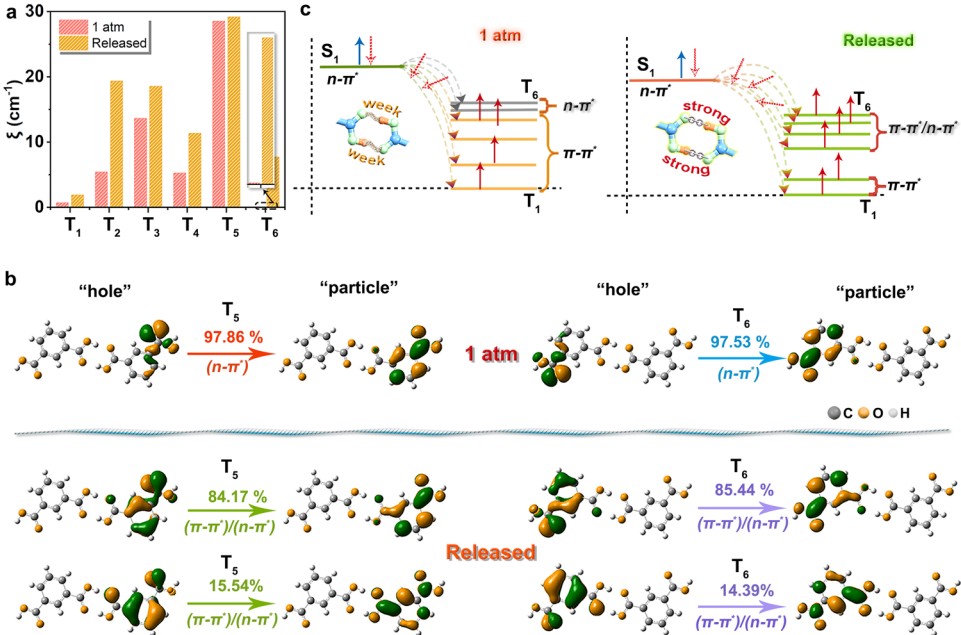

**Fig. 3 | The SOC and NTOs of the pristine and pressure-treated IPA. a** The calculated $\xi$ ($\xi(S_1 - T_6)$, $\xi(S_1 - T_5)$, $\xi(S_1 - T_4)$, $\xi(S_1 - T_3)$, $\xi(S_1 - T_2)$, and $\xi(S_1 - T_1)$) for the pristine and pressure-treated IPA. **b** The NTOs of $T_5$ and $T_6$ in the pristine and treated IPA. At 1 atm, the $T_5$ and $T_6$ states feature $^3(n, \pi^*)$ transition. After pressure treatment (Released), the $T_5$ and $T_6$ states feature a mixture of $^3(\pi, \pi^*)$ and $^3(n, \pi^*)$ transitions. **c** Schematic representation of hydrogen bonds acting on the ISC process. The strengthened hydrogen bonds promote the mixture of $^3(\pi, \pi^*)$ and $^3(n, \pi^*)$ transitions, thus accelerating the ISC process.

for the accelerated spin-flipping process (Fig. 3c). A high fraction of triplet excitons was then generated for drastic phosphorescence enhancement.

Additionally, the misaligned π–π stacking arrangement was prone to alter the coupling of the RIP excitons, which further influenced the HFC effect on the ISC process. To assess the impact of pressure treatment engineering on HFC, we carried out the magnetic-field PL (MPL) spectra of the pristine and pressure-treated IPA (Fig. 4a). The MPL intensity of the pristine sample changed with the magnetic flux density modulation of up to ~2 T. This magnetic-field effects proved the presence of RIP excitons and the HFC mechanism in the initial IPA[20,43]. However, after pressure treatment, we noticed minimal variation in pressure-treated IPA. The little influence of the external magnetic field on PL indicated that the RIP excitons were almost absent in the pressure-treated IPA[43]. In order to explore the relationship between the parallel-displaced arrangement and the radical pairs, we conducted the molecular orbitals of π–π staking dimers before and after pressure treatment. We found that the pristine IPA mainly exhibited charge-transfer-like π–π* transition between two IPA molecules, thus benefiting the presence of weakly coupled radical pairs. However, the pressure-treated IPA mainly shows an intramolecular n–π* transition, which is not conducive to the existence of radical pairs and the supplemental HFC channel (Supplementary Fig. 21). Therefore, the effect of HFC was negligible after pressure treatment owing to the misaligned π–π stacking arrangement.

Besides the HFC effect, this parallel-displaced configuration also contributes to minimizing energy difference $\Delta E(\pi, \pi^*)$. Notably, the enhanced hydrogen bonds could stabilize the n-orbitals to the lower energies by the enhanced electrostatic environments. These modifications would consequently induce variations in the relative energy levels of singlet and triplet states, along with changes in $\Delta E_{ST}$, which is closely associated with the ISC process. Therefore, the energies for both singlet and triplet excited states, as well as $\Delta E_{ST}$ were calculated in detail. We converged on investigations of the lowest singlet state ($S_1$) and six triplet states ($T_1$–$T_6$) below $S_1$[44]. Compared with the initial states, the treated sample featured various degrees of lowered energy

levels, coinciding with the experimental fluorescence and phosphorescence emission redshift (Fig. 4b and Supplementary Table 5). Meanwhile, the $\Delta E(S_1 - T_6)$, $\Delta E(S_1 - T_5)$, $\Delta E(S_1 - T_4)$, $\Delta E(S_1 - T_3)$, and $\Delta E(S_1 - T_2)$ decreased after pressure treatment (Fig. 4c, d and Supplementary Table 6). Based on the energy band theory[45],

$$K_{ISC} \propto \xi^{2*}\exp[-(\Delta E_{ST})^2] \qquad (1)$$

a reduced $\Delta E_{ST}$ would gift a higher degree of bright triplet excitons through an accelerated ISC process (where the $K_{ISC}$ is the ISC rate constant). Undoubtedly, the ISC rate was expected to be further expedited followed by these reduced energy gaps.

In this regard, the pressure treatment engineering synergistically improved SOC and lowered the $\Delta E_{ST}$ via strengthened hydrogen bonds, as well as parallel-displaced π–π staking motif, leading to an efficient spin-flipping process. Moreover, the strengthened hydrogen bonds would also suppress the non-radiative dissipation, benefiting the emission enhancement. In order to quantitatively analyze the effects of the pressure treatment engineering on ISC and radiative transition, we calculated the ISC, radiative, and non-radiative decay rates in the light of the standard methods using the measured quantum yields and lifetimes (Supplementary Table 1). According to the following equation[16],

$$\Phi_P = \Phi_{ISC} \frac{K_P}{K_P + K_{nr}} \qquad (2)$$

$$\tau_P = \frac{1}{K_P + K_{nr}} \qquad (3)$$

After pressure treatment, the estimated rate constant of the phosphorescence non-radiative decay ($K_{nr}$) remarkably decreased from 1.047 s⁻¹ to 0.667 s⁻¹. The significant increase of $K_P$ from 0.079 s⁻¹ to 0.444 s⁻¹ made a further contribution to the higher $\Phi_P$ (40%) in the targeted sample. Meanwhile, the fluorescent non-radiative decay $K_{nr}'$ remained a considerable decrease from 0.119 ns⁻¹ to 0.037 ns⁻¹ after an

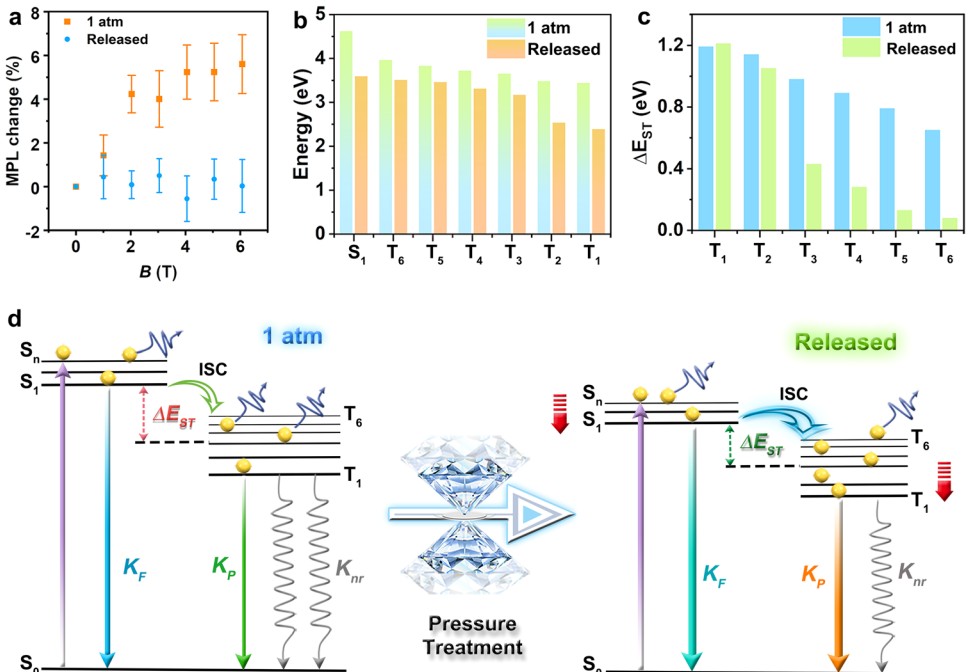

**Fig. 4 | The effects of HFC and singlet-triplet energy gap ($\Delta E_{ST}$) on the ISC process of IPA before and after pressure treatment. a** Magnetic-field effects on the PL intensity of IPA. The data at 1 atm and released represent the mean value from five and three experiments, respectively. The error bars are ±s.e.m. **b** The energies of singlet and triplet states of IPA before and after pressure treatment.

**c** The calculated $\Delta E_{ST}$ ($\Delta E(S_1 - T_6)$, $\Delta E(S_1 - T_5)$, $\Delta E(S_1 - T_4)$, $\Delta E(S_1 - T_3)$, $\Delta E(S_1 - T_2)$, and $\Delta E(S_1 - T_1)$) for the pristine and targeted IPA. **d** Proposed energy transfer processes for fluorescence and phosphorescence in IPA before and after pressure treatment. The narrowed $\Delta E_{ST}$ effectively accelerated the ISC process after pressure treatment.

entire compression cycle of 1 atm to 20.0 GPa, resulting in the enhancement of $\Phi_F$ from 12% to 35%. Moreover, we received a rapid ISC rate (0.059 ns⁻¹) in the pressure-treated sample from the original 0.010 ns⁻¹, which sufficiently triggered the balanced contribution of singlet and triplet excitons to the white-light emission.

## Discussion

In summary, we report a convenient and facile strategy to evoke triplet excitons based on pure organic small molecule, IPA, for high-efficiency white-light emission by pressure treatment engineering. In contrast to current chemical techniques in manipulating ISC by introducing diverse heavy atoms or chromophores, our strategy using pressure treatment engineering can effectively facilitate spin-flipping through the hydrogen bonding cooperativity effect and parallel-displaced π–π staking arrangement. The strengthened hydrogen-bond interaction in the pressure-treated structure also ensured the suppression of non-radiative dissipation for further emission enhancement. Pressure treatment engineering has endowed the IPA with optimized phosphorescent emission via elaborate structural regulation, which would provide valid clues to broaden perspectives for designing and developing high-performance phosphors at ambient conditions.

## Methods

### Characterization and high-pressure generation

IPA was purchased from Aladdin and used directly without purification. The crystal structure at ambient conditions was obtained from the Cambridge Crystallographic Data Center (CCDC) with CCDC number 1108747. The refinements of ADXRD patterns were performed using the GSAS software[46]. All in situ high-pressure experiments were performed using a symmetric diamond anvil cell (DAC) apparatus at room temperature. The sample and a small ruby ball were loaded into the ~150 μm-diameter DAC chamber consisting of a T301 steel gasket pre-indented to a thickness of 45 μm. The pressure calibration was determined utilizing the standard ruby fluorescent technique[47]. In

high-pressure experiments, silicone oil with a viscosity of 10 cst was utilized as the PTM for ADXRD, optical absorption, and PL experiments, while the KBr was employed as the PTM for IR measurements. These PTMs did not have any detectable effect on the behavior of IPA under pressure. All of the measurements were performed at room temperature.

### In situ high-pressure PL, UV-vis absorption, IR spectra, and ADXRD experiments

A 355 nm line of a UV DPSS laser with a power of 4.8 mW was used for the PL measurements. The in situ high-pressure steady-state PL spectra of IPA were collected by a modified spectrophotometer (Ocean Optics, QE65000) at various pressures. PL micrographs of the samples were obtained using a camera (Canon Eos 5D mark II) equipped with a microscope (Eclipse TI-U, Nikon). The camera can record the photographs under the same conditions including exposure time and intensity. The chromaticity coordinates $(x, y)$ were calculated from the fluorescence data (355–800 nm) using the CIE1931xy.V.1.6.0.2a software package. The color of the fluorescent emission was identified by the CIE colorimetry system. Any colors could be described by the chromaticity $(x, y)$ coordinates on the CIE diagram. The absorption spectra were performed with a Deuterium-Halogen light source and recorded with an optical fiber spectrometer (Ocean Optics, QE65000). IR spectra measurements of IPA were carried out by Nicolet iN10 microscope spectrometer (Thermo Fisher Scientific, USA) using liquid-nitrogen-cooled CCD. High-pressure experiments on IPA were further carried out in a Walker-type JLUHC-1000 LVP to guarantee the quantity of quenched products (after pressure was released from 20 GPa) for further PL, phosphorescence, PLQY, and time-resolved PL decays measurements. The PLQY was measured with Hamamatsu multi-channel analyzer c10027[48]. In situ high-pressure ADXRD patterns were obtained with a wavelength of 0.6199 Å at beamline 15U1, Shanghai Synchrotron Radiation Facility (SSRF), China. CeO₂ was applied to the standard sample to do the calibration. The FIT2D program was used to

integrate the two-dimensional images, and the one-dimensional diffraction angle 2-theta diagram was obtained. All the high-pressure experiments were conducted at room temperature.

## Magnetic photoluminescence (MPL) experiments

The MPL experiments were performed at Wuhan National High Magnetic Field Center. For the MPL measurements, multimode optical fibers were used to transmit both the excitation and emission light beams. The PL was excited by a solid-state laser with a wavelength of 360 nm and recorded by a spectrometer composed of a monochromator (SP500l, Andor) and an electron multiplied charge-coupled device (EMCCD, Newton 970P, Andor). The pristine IPA was purchased from Aladdin. The pressure-treated IPA sample was harvested by the Kawai-Type Large Volume Press device at the B1 station, Synergetic Extreme Condition User Facility (SECUF). These samples (~10 mg, diameter 3 mm, thickness 1 mm) were placed into a liquid helium cryostat at the center of a pulsed magnet with the designed peak field of 50-T and pulse duration of 300 ms. In this situation, the EMCCD was exposed every millisecond synchronous to the time period of the magnetic field pulse.

## In situ high-pressure neutron diffraction

Given the considerable incoherent scattering cross-section of hydrogen leading to a notable background contribution in neutron diffraction data, we opted for deuterated IPA (IPA-$d_6$) in the in situ neutron diffraction experiment[49]. The neutron diffraction data up to 19.8 GPa were collected at BL11 PLANET in the Materials and Life Science Experimental Facility (MLF) at Japan Proton Accelerator Research Complex (J-PARC)[50]. Before the high-pressure experiment, the fresh sample with a Pb marker was loaded into a vanadium can to collect the neutron diffraction data. The data were used as an ambient-pressure reference. Then, a VX4 Paris–Edinburgh (PE) Press equipped with sintered diamond double toroidal anvils with a diameter of 3 mm was used to apply the pressure. Titanium–Zirconium alloy gaskets were used as the sample chamber, along with a Pb pressure marker. Liquid argon was used as PTM in high-pressure neutron diffraction experiments. An automatic hydraulic oil syringe pump was used to drive the system and the pressure was estimated according to the equation of state (EOS) of Pb[51,52]. The data were collected in the range of 0–19.8 GPa. The rates of compression and decompression were kept at 10 bar/min (~0.2 GPa/min) below 500 bar and 5 bar/min (~0.1 GPa/min) above 500 bar. The pattern of the sample at each pressure was collected for 2 h. The crystal structures and atomic positions of IPA-$d_6$ were determined by Rietveld refinement using the GSASII software package. The high-pressure structural data was provided in the Supplementary Information.

The recovered sample was re-synthesized using a VX3 PE Press without PTM and Pb marker. The encapsulated gaskets were subsequently removed to completely release the pressure. The original and recovered samples (90 mg) were then loaded into vanadium cans with a cut diameter of 6 mm, respectively to collect the neutron diffraction data. This measurement was performed at High pressure neutron diffractometer (Fenghuang) at China Mianyang Research Reactor's (CMRR) neutron science platform[53,54]. And 26 h were spent on the collecting pattern of each sample.

## First-principle calculations

Calculations on the excited-states energies and the NTOs were carried out using the Gaussian 09 (version D.01) package on a PowerLeader cluster[55]. The excited state energies were calculated by time-dependent density functional theory (TD-DFT) at the level of CAM-B3LYP/6-31G(d, p). The NTOs were evaluated with the dominant "hole"-"particle" pair contributions. The SOC coefficients were quantitatively estimated at the level of CAM-B3LYP/6-31G(d,p) by the Beijing density function program[56-60].

## Data availability

The authors declare that the main data supporting our findings of this study are contained within the paper and Supplementary Information. All other relevant data are available from the corresponding author upon request. Source data are provided with this paper.

## Code availability

Gaussian 09 code is available for download on the developer page: https://gaussian.com/.

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

## Acknowledgements

This work is supported by the National Natural Science Foundation of China (Nos. 12274177 and 22022101). The authors also acknowledge the support of the National Key Research and Development Program of China (2019YFA0708502). Angle dispersive XRD measurements were performed at the BL15U1 beamline, SSRF. The Kawai-type large-volume press experiments were performed at the B1 station, Synergetic Extreme Condition User Facility (SECUF). Neutron diffraction data were collected at BL11 PLANET of the Japan Proton Accelerator Research Complex

under the J-PARC user programs (Proposal Nos. 2023B0100 and 2024A0054) and High-pressure neutron diffractometer (Fenghuang) at China Mianyang Research Reactor's (CMRR) neutron science platform. The MPL experiments were performed at Wuhan National High Magnetic Field Center. The authors thank Dr. Jun Abe for supporting the neutron diffraction experiment.

## Author contributions

X.Y. and B.Z. designed the project and supervised the work. Q.Y. performed the in situ high-pressure PL, UV-vis Absorption, and IR spectra measurements. Q.Y., X.Y., and Y.W. conducted the ADXRD, time-resolved PL decay curves, and PLQY experiments. Y.F., F.L., H.Z., K.L., T.H., and L.F. carried out neutron diffraction measurements. Q.Y. and Y.H. collected the MPL data. P.Z., X.H., and Z.L. synthesized the pressure-treated IPA sample with Large-Volume Press. S.Z. and B.Y. performed the calculations. Q.Y., X.Y., and B.Z. wrote the manuscript. All authors discussed the results and commented on the manuscript.

## Competing interests

The authors declare no competing interests.
