## [Peer Review File · Nature Communications]

Brightening triplet excitons enable high-performance white-light emission in organic small molecule via integrating $n-\pi^*/\pi-\pi^*$ transitionsEditorial Note: Parts of this Peer Review File have been redacted as indicated to remove third-party material where no permission to publish could be obtained.

Reviewers' comments:

Reviewer #1 (Remarks to the Author):

This manuscript reports the observation of the giant afterglow's intensification in isophthalic acid (IPA) after a compression/decompression cycle up to 20 GPa. The white emission deriving from the strong intensity increase of both phosphorescence and fluorescence could be of a certain appeal in optical applications. The system has been characterized by spectroscopic, diffractometric, and computational techniques for providing a complete description of the structural changes that lead to the observed phenomena.

The work is of a certain interest but there are so many flaws to hide the scientific content. The following list addresses the most important shortcomings even though several other disputable sentences or words (es: splendid line 64, triumphantly line 165, excitingly line 183, obvious line 229) are disseminated along the manuscript.

a) the remarkable afterglow emission is attributed to the strengthening of the H-bonds with pressure and their metastable recover at ambient conditions. However, no quantitative proofs of this relation is provided and, in addition, reference to a previous accurate study is completely missing. A superficial account of the existing literature is widespread in the manuscript especially as it concerns previous high pressure studies of organic crystals. The paper I was referring (PCCP 2015, 17, 15989) reports an accurate study of the IPA afterglow phosphorescence which allows, also through magnetic measurements, to assign this emission to the spin exchange between a radical ion pair. The packing of IPA molecules promotes the formation of charge transfer complexes, or their derived radical pairs, so that the two electron spins over the pair are only weakly coupled allowing the spin exchange and then the intersystem crossing which is at the basis of the phosphorescence. The H-bond can have an effect, but to demonstrate its relevance I would like to see quantitative proofs like the measurement of the atomic positions vs P.

b) the sentence on page 6 "The carboxyl units locked by hydrogen bonds enable IPA to exhibit a green phosphorescence." is neither demonstrated nor a citation is provided, how do they reach such a conclusion?

c) the photoluminescence measurements have been performed using the 355 laser line, but the authors never discuss if some reactivity occurs at high pressure under irradiation, a quite common occurrence.

d) on page 8 lines 146-149 contain an incredible number of perplexing statements. Line 146: lattice parameters I presume. Line 147: how much does the monoclinic angle reduce? Fig. 10 in the supplementary (incredibly) does not show numbers. Line 148: they state that the monoclinic angle reduction with pressure reduces the partial overlap of the π -stacked motif; are they sure? I would say the opposite. Line 149: radioluminescence? Where is the ionizing radiation?

e) Figure 2: panels b, c and d are exchanged.

f) from the structural data the authors calculate the pressure evolution of the intra- and intermolecular dihedral angles. How do they achieve this result? They were able to refine the atomic positions from the XRD data? I don't think so looking to the XRD patterns quality. All the dihedral angles are reported to reduce of less than 1 degree, at the most 0.76 for the intermolecular one (the adverb triumphantly used here is ridiculous) but I would like to see the error bars on the data to understand their reliability. In addition, they never mention if, and in case which one, PTM used for the XRD experiments. This is crucial to evaluate the reliability of the data. Silicon oil has been used for the emission experiments, it is nearly hydrostatic only up few (5-6) GPa.

g) Concerning the HOMO-LUMO calculations is never specified which kind of system they computed: single molecule, cluster, crystal cell... How the pressure effects are computed? Imposing the lattice parameters? The recovered material is simulated as an ordered structure?

h) The only evidence provided of the enforcement of the hydrogen bond with pressure is the red shift of the carbonyl frequency. The shift is quantified in 14 cm^{-1} up to 20 GPa. It would be nice to have an explanation of how they catch the peak maximum above 10 GPa since the spectra are extremely broad and unresolved. However, this shift is very small being $<1\%$, and is much lower than that of the recovered material (6 cm^{-1}) that could be simply due to residual stress in the sample. It would be also interesting to see the IR spectrum of the recovered material in order to judge which is the quality of the sample and if some reaction occurred.

i) On the basis of the H-bond enforcement, they also speculate about the stabilization of the non-bonding orbitals and in the variations of the relevant rate constants characterizing the radiative

and non radiative decays. Again all qualitative statements.

l) Lines 210-213: No details are provided on how they computed the hydrogen bond binding energy especially for the sample recovered after the pressure treatment. How do they compute this energy if the local structure of the recovered material is not known. Regardless of how this energy is computed, 1.35 eV is huge considering the minor effect (if any) observed on the C=O stretching frequency.

m) All the discussion about the absorption spectrum (lines 227-236) is absolutely nonsense. They cannot estimate the band gap from an out-of-scale spectrum. They must measure the peak maximum since the onset of the absorption not only depends on the peak maximum but also on the intensity. According to their method, Figure 13 in the supplementary material, the onset can just apparently move to the red when the absorption band intensifies.

This manuscript although reporting an interesting phenomenon is extremely weak and misleading undeserving, in my opinion, the publication in a scientific journal.

Reviewer #2 (Remarks to the Author):

This work realizes the harvesting of bright white-light emission through modulating the balanced population of singlet/triplet excitons in an organic small molecule IPA through high pressure processing. Through both calculation and experiments, it demonstrates that the enhanced hydrogen bonds boosts the mixture of $n\text{-}\pi^*/\pi\text{-}\pi^*$ transition configurations that is in favor of the promoted intersystem crossing for white-light emission. The result is unique, interesting, and important. The manuscript is well written and should be published in Nature Communications after minor revisions.

1. The authors obtained a brilliant white-light emission based on balanced fluorescence and phosphorescence species after decompression from 18 GPa. What about decompression from lower pressures? Any detailed data would be valuable.
2. The PL intensity continuously increases before 9.0 GPa due to the red-shift of C=O stretching vibration. At higher pressures, this mode continuously red-shifts, while the PL intensity decreases. Discussion on the possible reason/mechanism would be appreciated by the reader.
3. The white-light emission remained for about 400 days after releasing from pressure. What was the storage/environmental condition/s the sample is located?
4. The binding energy of the hydrogen bond before pressure processing should be provided to clarify the enhancement of hydrogen bond before and after compression.
5. Some minor errors or typos, examples below:
 - a. Page 3 line 45 , "remains as a grand challenge" to "as remains a grand challenge";
 - b. The representation of T5 state in figure 4b should be consistent with the one in Supplementary (Figure 14);
 - c. Page 3, line 42 "as well as color variability" to "as well as to enable color variability"?

Reviewer #3 (Remarks to the Author):

The authors report an hybrid fluorescence and phosphorescence white-light emission with a 75% photoluminescence quantum yield. Through compression and decompression of isophthalic acid (an organic molecule) they are able to mix $n\text{-}\pi^*/\pi\text{-}\pi^*$ transition configurations, leading to balanced singlet and triplet excitons distribution.

The hybrid fluorescence/phosphorescence phenomena has been extensively studied, and most recently the influence of the pressure on fluorescence (Nature Communications, 13, 5234 (2022)).

The main contribution of the work presented is mixing both worlds, pressure and hybrid fluorescence/phosphorescence.

The results are well presented, although the hydrogen-bonding/electron delocalization effects should be better explained.

The triplet transitions should also be also more clearly explained, to be understood for a wider audience.

I recommend the publication of this work if these issues are solved.

For your guidance, itemized response to each reviewers' comments is appended below.

Reply to Reviewer #1

Comments: This Manuscript reports the observation of the giant afterglow's intensification in isophthalic acid (IPA) after a compression/decompression cycle up to 20 GPa. The white emission deriving from the strong intensity increase of both phosphorescence and fluorescence could be of a certain appeal in optical applications. The system has been characterized by spectroscopic, diffratometric, and computational techniques for providing a complete description of the structural changes that lead to the observed phenomena.

The work is of a certain interest but there are so many flaws to hide the scientific content. The following list addresses the most important shortcomings even though several other disputable sentences or words (es: splendid line 64, triumphantly line 165, excitingly line 183, obvious line 229) are disseminated along the Manuscript.

Author reply: Thanks for the reviewer's kind comments with positive affirmation. First of all, we acknowledge your comments and suggestions very much, which are valuable in improving the quality of our manuscript. Furthermore, we have tried our best to check the terms of the whole manuscript to make them concise and clarity, which is highlighted in red and can be found in **Lines 28, 63-64, 67, 175, 197, 257 and 320**. In addition, we also added experimental and theoretical data to further support our conclusion in accordance with your guidance. Please see below for the details of all the improvements.

Comments 1: The remarkable afterglow emission is attributed to the strengthening of the H-bonds with pressure and their metastable recover at ambient conditions. However, no quantitative proofs of this relation is provided and, in addition, reference to a previous accurate study is completely missing. A superficial account of the existing literature is widespread in the Manuscript especially as it concerns previous high pressure studies of organic crystals. The paper I was referring (PCCP 2015, 17, 15989) reports an accurate study of the IPA afterglow phosphorescence which allows, also through magnetic measurements, to assign this emission to the spin exchange between a radical ion pair. The packing of IPA molecules promotes the formation of charge transfer complexes, or their derived

radical pairs, so that the two electron spins over the pair are only weakly coupled allowing the spin exchange and then the intersystem crossing which is at the basis of the phosphorescence. The H-bond can have an effect, but to demonstrate its relevance I would like to see quantitative proofs like the measurement of the atomic positions vs P.

Author reply: Thanks for the reviewer's comments. Generally, the hydrogen-bond distance and angles have been regarded as important geometric parameters to estimate the strength of hydrogen bonds¹⁻³. In the present work, we further performed the angle-dispersive synchrotron X-ray diffraction (ADXRD) and infrared (IR) spectra in order to quantitatively prove the enhanced hydrogen bonds. **We have collected and analyzed atomic positions, the hydrogen-bond distance and angle at various pressures. The reduced hydrogen-bond distance and increased hydrogen bond angles directly confirm the strengthening of hydrogen bonds.**

(1) First of all, based on the ADXRD spectra, we have collected atomic positions, the hydrogen-bond distance and angle by the refinement of the diffraction patterns at different pressures. The method of the refinements can be found in the experimental method part. The obtained atomic positions of H₁₆, O₂, O₃, and H₅ at different pressures are shown in Figure R1. In order to illustrate the hydrogen bond changes with pressure, we measured the distance and angles of hydrogen bonds upon compression in Added Supplementary Fig. 12a. As the pressure increased to 12.4 GPa, the hydrogen-bond distance of H₁₆···O₂ (d₁) decreased from 1.624 Å to 1.546 Å and H₅···O₃ (d₂) decreased from 1.705 Å to 1.620 Å. This indicated the strengthening of hydrogen bonds at high pressure². After releasing pressure completely, the d₁ and d₂ were reduced to 1.613 Å and 1.694 Å, respectively. This suggested that the pressure-treated hydrogen bonds were stronger than their initial state. In addition, the O₄–H₁₆···O₂ angle (α_1) and O₁–H₅···O₃ angle (α_2) gradually increased with increasing pressure, which further contributed to enhancing hydrogen bonds under high pressure (Added Supplementary Fig. 12b)¹. After decompression to ambient pressure, α_1 and α_2 remained the same as the initial values, indicating the enhanced hydrogen bonds in pressure-treated IPA were mainly rooted in the decreased hydrogen-bond distance (Added Supplementary Fig. 12c). Moreover, the enlarged 2-theta values of (062) and (042) lattice planes in the pressure-treated ADXRD pattern demonstrated a reduced *d*-spacings of these two lattice planes after pressure treatment (Revised Supplementary Fig. 11b, c). It is noted that the orientation of (062) and (042) lattice planes is almost perpendicular to the direction of the hydrogen

bond, and their variation of d -spacings is closely related to the change of hydrogen-bond distance (Revised Supplementary Fig. 11a)⁴. Therefore, the reduced d -spacings of (062) and (042) lattice planes further confirm the acquisition of the reduced hydrogen-bond distance in pressure-treated IPA. **In addition, the experimental IR data were analyzed in detail to further confirm the observed strengthened hydrogen bonds under high pressure and after pressure treatment.** Upon compression, the C=O stretching vibration mode ($\nu(\text{C}=\text{O})$) redshifted by 14 cm^{-1} below 20.0 GPa (Revised Supplementary Fig. 8b). It suggested that the O–H \cdots O=C hydrogen bonds were continuously strengthened with the increase of pressure^{4,5}. The redshift of 6 cm^{-1} in pressure-treated structure also proved the enhancement of hydrogen bonds after pressure treatment (Revised Fig. 2b). **In summary, these experimental data gave us enough evidences to support the conclusion of the enhanced hydrogen bonds caused by pressure treatment.**

Figure R1. a The structure of IPA. Selected atomic positions of b H₁₆, c O₂, d O₃, and e H₅ at different pressures.

Added Supplementary Fig. 12. **a** Pressure-dependent hydrogen-bond distance $H_{16}\cdots O_2$ (d_1) and $H_5\cdots O_3$ (d_2) evolution of IPA. **b** Pressure-dependent $O_4-H_{16}\cdots O_2$ angle (α_1) and $O_1-H_5\cdots O_3$ angle (α_2) evolution of IPA. **c** Schematic diagram of hydrogen bonds of IPA before and after pressure treatment. The hydrogen-bond distance (d_1 and d_2) and angles (α_1 and α_2) are also labeled for clarity.

Revised Supplementary Fig. 11. **a** The lattice planes of (062) and (042) in IPA. **b** ADXRD patterns of the IPA crystal upon decompression ($\lambda = 0.6199 \text{ \AA}$). **c** The 2-theta value of the selected lattice planes at 1 atm and after releasing pressure completely.

Revised Supplementary Fig. 8. **a** ADXRD patterns of the IPA crystal under different pressures ($\lambda = 0.6199 \text{ \AA}$). **b** Selected IR spectra of IPA in the region of C=O stretching vibrational mode (v(C=O)) upon compression.

Revised Fig. 2 | Crystal structure evolution upon compression and decompression. **a** The

compression rate of three lattice constants (a , b , c) at different pressures. **b** IR spectra of IPA in the region of C=O stretching vibrational mode $\nu(\text{C}=\text{O})$ at ambient conditions and after pressure was released from 20.0 GPa. **c** Schematic illustrations of the intramolecular dihedral angle (δ_1 and δ_2) between the benzene ring and carboxyl groups as well as the intermolecular dihedral angle (θ) between two benzene rings of the dimer connected by hydrogen bonds. **d** The Jablonski diagram. **e** Hydrogen bond binding energy of the pressure-treated IPA.

(2) In addition, we carefully checked and cited new references to avoid the superficial account of the existing literature. Kuno et al.'s paper⁶ mentioned by the reviewer is very helpful to our work, so it is cited in our revised manuscript. In crystal IPA, the single IPA molecules are linked by **hydrogen bonds** to form a zigzag chain. The chains are stacking offset face-to-face via **π - π interaction**. On one hand, the hydrogen bonds could convert the lowest singlet excited state S_1 from $^1(n,\pi^*)$ to $^1(\pi,\pi^*)$ type through intermolecular electrostatic interaction, which accelerates the intersystem crossing (ISC) for room temperature phosphorescence (RTP) emission⁷. On the other hand, the π -stacked motif of crystal IPA promotes the formation of charge transfer complexes, or their derived radical pairs, benefiting the RTP through hyperfine coupling⁶. Therefore, both of these two effects, hyperfine coupling and electrostatic hydrogen-bond interaction, would affect the RTP of crystal IPA.

In the present work, we further analyze the changes of hydrogen bonds and π -stacked motif in detail upon compression and decompression. The reduced hydrogen-bond distance and redshifted $\nu(\text{C}=\text{O})$ mode have proved the enhancement of hydrogen bonds after pressure treatment. Therefore, the enhancement of the hydrogen bond should be related to the resulting RTP enhancement of the pressure-treated IPA. To investigate whether the π -stacked motif contributed to the enhanced RTP in the pressure-treated structure, we added *in situ* high-pressure Raman measurements to explore the evolution of π -packing motif upon compression and decompression (Figure R2). The peaks located at 1001 cm^{-1} signify $\nu(\text{CC})_{\text{ring}}$ stretching vibration. Upon compression to 7.0 GPa, the $\nu(\text{CC})_{\text{ring}}$ mode split into two peaks, indicating the strong π - π interaction and tighter π -stacked motif due to the compressed configuration⁸⁻¹⁰. After pressure was completely released, the $\nu(\text{CC})_{\text{ring}}$ mode recovered to the initial state. This suggested that **the forced packing of IPA molecules was restored to the original state after pressure treatment**. Notably, the enhanced hydrogen bonds in the pressure-treated IPA have been concluded by the combination of the ADXRD and IR spectra. **Therefore, the enhanced hydrogen bonds have dominated the resulting phosphorescence enhancement in the pressure-**

treated IPA. In the present work, we found that the strengthened hydrogen bonds successfully converted the natural transition-orbital (NTOs) type of triplet states (Revised Fig. 4b). The NTOs of T_5 and T_6 states underwent a noticeable change from $^3(n, \pi^*)$ in the original IPA to the mixture of $^3(n, \pi^*)$ and $^3(\pi, \pi^*)$ in the pressure-treated IPA. Considering the $^1(n, \pi^*)$ type of S_1 state (Supplementary Fig. 16 in revised Supplementary Information), the ISC from S_1 to T_5 and T_6 was favored in pressure-treated structure based on the El-Sayed's rule¹¹. Therefore, the ISC was then accelerated for RTP enhancement after pressure treatment.

The revised details are highlighted in red and can be found in Lines 135, 137-183, 364-376, Revised Fig. 2 and 4 and their captions of the revised manuscript, as well as Lines 85-95, revised Supplementary Fig. 8, 11, Added Supplementary 12 and their captions of the revised Supplementary Information.

Figure R2. Raman spectra upon compression and decompression. The red arrow indicates $\nu(\text{CC})_{\text{ring}}$ stretching vibration.

Revised Fig. 4 | The SOC and NTOs of the pristine and pressure-treated IPA. **a** The calculated ζ ($\zeta(S_1-T_6)$, $\zeta(S_1-T_5)$, $\zeta(S_1-T_4)$, $\zeta(S_1-T_3)$, $\zeta(S_1-T_2)$, $\zeta(S_1-T_1)$) for the pristine and pressure-treated IPA. **b** The NTOs of T_5 and T_6 state in the pristine and treated IPA. **c** Schematic representation of hydrogen bonds acting on ISC process.

References:

1. Arunan, E. et al. Definition of the hydrogen bond (IUPAC Recommendations 2011). *Pure Appl. Chem.* **83**, 1637-1641 (2011).
2. Shi, B. et al. Short hydrogen-bond network confined on COF surfaces enables ultrahigh proton conductivity. *Nat. Commun.* **13**, 6666 (2022).
3. Emamian, S., Lu, T., Kruse, H. & Emamian, H. Exploring nature and predicting strength of hydrogen bonds: a correlation analysis between atoms-in-molecules descriptors, binding energies, and energy components of symmetry-adapted perturbation theory. *J. Comput. Chem.* **40**, 2868-2881 (2019).
4. Wang, X. et al. From biomass to functional crystalline diamond nanothread: pressure-induced polymerization of 2,5-furandicarboxylic Acid. *J. Am. Chem. Soc.* **144**, 21837-21842 (2022).
5. Lu, B. et al. Piezochromic luminescence of AIE-active molecular co-crystals: tunable multiple hydrogen bonding and molecular packing. *J. Mater. Chem. C* **6**, 9660-9666 (2018).

6. Kuno, S., Akeno, H., Ohtani, H. & Yuasa, H. Visible room-temperature phosphorescence of pure organic crystals via a radical-ion-pair mechanism. *Phys. Chem. Chem. Phys.* **17**, 15989 (2015).
7. Ma, H. et al. Electrostatic interaction-induced room-temperature phosphorescence in pure organic molecules from QM/MM calculations. *J. Phys. Chem. Lett.* **7**, 2893-2898 (2016).
8. Hu, Y., Kazemian, H., Rohani, S., Huang, Y. & Song, Y. In situ high pressure study of ZIF-8 by FTIR spectroscopy. *Chem. Commun.* **47**, 12694 (2011).
9. Miyazaki, M. & Fujii, M. A structural study on the excimer state of an isolated benzene dimer using infrared spectroscopy in the skeletal vibration region. *Phys. Chem. Chem. Phys.* **19**, 22759-22776 (2017).
10. Yuan, H., Wang, K., Yang, K., Liu, B. & Zou, B. Luminescence properties of compressed tetraphenylethene: the role of intermolecular interactions. *J. Phys. Chem. Lett.* **5**, 2968-2973 (2014).
11. El-Sayed, M. A. Triplet state. Its radiative and non-radiative properties. *Acc. Chem. Res.* **1**, 8–16 (1968)

Comments 2: *The sentence on page 6 “The carboxyl units locked by hydrogen bonds enable IPA to exhibit a green phosphorescence.” is neither demonstrated nor a citation is provided, how do they reach such a conclusion?*

Author reply: We are so sorry for our expressions confusing the reviewer. We intend to express that the crystal IPA has a green phosphorescence emission at ambient conditions. And the emission in the crystal state is related to various intermolecular interactions, in which hydrogen bonds play a significant role in the phosphorescence emission. We make a corresponding modification of this sentence and supply the citation.

The understanding of the phosphorescence of IPA can be found in the reported literatures¹⁻³. At ambient conditions, the isolated IPA molecules in solutions are almost non-emissive owing to the active intramolecular motions (rotations and vibrations) and intermolecular collisions. A unique phenomenon of **crystallization-induced dual emission**, namely, simultaneously **boosted fluorescence and phosphorescence upon crystallization**, is observed in IPA crystals at ambient conditions¹. In crystal IPA, those intramolecular motions and intermolecular collisions are significantly inhibited due to the conformation rigidification resulting from the efficacious intermolecular

interactions, thus diminishing the non-radiative relaxations and boosting the fluorescence and phosphorescence emissions¹. Further research on the phosphorescence of crystal IPA can be found in the previous studies from Ma et al.² and Kuno et al.³. Ma et al. proposed **electrostatic interaction-induced phosphorescence** for IPA². In the crystal IPA, the carboxylic groups are connected by hydrogen bonds (O–H···O) with intermolecular electrostatic interaction. This work suggests that the *n*-orbital represents negative charge in the oxygen atom attracting the positive charge in hydrogen atom, which could be largely stabilized by the electrostatic interaction. At crystalline phase, such intermolecular electrostatic interaction converts the lowest singlet excited state S₁ from ¹(n, π*) to ¹(π, π*) type. Considering the T₆ states belong to ¹(n, π*) state, the ISC from S₁ to T₆ is favored based on the El-Sayed's rule⁴. And the efficient ISC thereby boosts the phosphorescence of the crystal IPA. In addition, Kuo et al. found that the phosphorescence of crystal IPA was also related to the **hyperfine coupling**³. The forced packing of IPA molecules promotes the formation of charge transfer complexes, or their derived radical pairs. The two electron spins over the pair are only weakly coupled allowing the spin exchange and then facilitating ISC to emit phosphorescence of crystal IPA. Therefore, the IPA in crystal state could emit the phosphorescence at ambient conditions.

The revised details are highlighted in red and can be found in **Lines 94–96** in the revised manuscript.

References:

1. Gong, Y. et al. Crystallization-induced dual emission from metaland heavy atom-free aromatic acids and esters. *Chem. Sci.* **6**, 4438 (2015)
2. Ma, H. et al. Electrostatic interaction-induced room-temperature phosphorescence in pure organic molecules from QM/MM calculations. *J. Phys. Chem. Lett.* **7**, 2893-2898 (2016).
3. Kuno, S., Akeno, H., Ohtani, H. & Yuasa, H. Visible room-temperature phosphorescence of pure organic crystals via a radical-ion-pair mechanism. *Phys. Chem. Chem. Phys.* **17**, 15989 (2015).
4. El-Sayed, M. A. Triplet state. Its radiative and non-radiative properties. *Acc. Chem. Res.* **1**, 8–16 (1968)

Comments 3: *The photoluminescence measurements have been performed using the 355 laser line, but the authors never discuss if some reactivity occurs at high pressure under irradiation, a quite common occurrence.*

Author reply: We are grateful for the reviewer's constructive suggestions. According to the reviewer's advice, we performed supplementary experiments to investigate if some reactivity occurs at high pressure under irradiation. In this regard, we have studied *in situ* high-pressure photoluminescence (PL), absorption, Raman, XRD and IR spectra evolution before and after irradiation using 355 nm laser line at high pressure. These data give us enough evidences to rule out the occurrence of some reactivity under irradiation, and the detailed information is as follows:

As shown in Figure R3, the PL spectrum was collected after an entire compression cycle of 1 atm to 20 GPa, where the chromaticity coordinate is (0.29, 0.36). The entire compression and decompression process were carried out without irradiation at all. We directly collected the compressed sample to measure the PL spectrum. This result proved that the production of white-light emission was the result of pressure-treatment engineering. In addition, we performed high-pressure PL spectra under irradiation for 10 and 20 minutes using 355 laser line. It is found that the PL spectra of compressed sample under high pressure remain basically unchanged after continuous irradiation for 10 and 20 minutes using 355 laser line (Figure R4). The absorption spectra before and after irradiation at high pressure were also carried out (Figure R5). We find that the absorption spectra remain the same as the one without irradiation. This indicates that the continuous irradiation using 355 laser line doesn't change the structural properties. Selected Raman and IR spectra before and after irradiation at high pressure were further performed in Figures R6 and R7. It is clearly seen that their spectra after irradiation for 10 and 20 minutes stay the same as the one before irradiation, suggesting that the crystal structure of IPA under high pressure doesn't change after continuous irradiation for 10 and 20 minutes using 355 laser line. **Therefore, the enhancement of emission intensity is not the result of photochemical reactions.**

More importantly, the Raman, IR and XRD spectra of the pressure-treated sample before and after irradiation were also performed in Figures R8-10. The almost unchanged spectra further confirm that **the production of hybrid fluorescence/phosphorescence white-light emission is not the result of a photochemical reaction.**

Figure R3. **a** The PL spectrum of the pressure-treated sample after a compression cycle of 1 atm to 20.0 GPa with the Walker-Type Large-Volume Press. **b** The chromaticity coordinates (0.29, 0.36) of the corresponding pressure-treated IPA.

Figure R4 a-f PL spectra before and after irradiation for 10 and 20 minutes using 355 laser line at different pressures.

Figure R5 Absorption spectra of IPA at different pressures, including the spectra before and after irradiation for 10 and 20 minutes using 355 laser line.

Figure R6 Selected Raman spectra of IPA at different pressures, including the spectra before and after irradiation for 10 and 20 minutes using 355 laser line.

Figure R7 Selected IR spectra of IPA at different pressures, including the spectra before and after irradiation for 10 and 20 minutes using 355 laser line.

Figure R8 Selected Raman spectra of initial and pressure-treated IPA, including the spectra before and after irradiation for 10 and 20 minutes using 355 laser line.

Figure R9 Selected IR spectra of initial and pressure-treated IPA before and after irradiation for 10 minutes using 355 laser line.

Figure R10 XRD spectra of the pressure-treated IPA before and after irradiation for 20 minutes using 355 laser line ($\lambda = 1.54056 \text{ \AA}$).

Comments 4: On page 8 lines 146-149 contain an incredible number of perplexing statements. Line 146: lattice parameters I presume. Line 147: how much does the monoclinic angle reduce? Fig. 10 in the Supplementary (incredibly) does not show numbers. Line 148: they state that the monoclinic angle

reduction with pressure reduces the partial overlap of the π -stacked motif; are they sure? I would say the opposite. Line 149: radioluminescence? Where is the ionizing radiation?

Author reply: Thanks for the reviewer's comments. We provided the numbers for the ordinate in Supplementary Fig. 10a (Original: Supplementary Fig. 10) to clearly display the reduction of monoclinic angle β . The schematic diagram of the parallel misalignment angle (σ) of IPA was also provided to directly confirm the reduced partial overlap of the π -stacked motif in revised Supplementary Information, and the detailed information is as follows:

The monoclinic angle β underwent continuous reduction upon compression from 90.34° at 1 atm to 84.66° at 12.4 GPa. The reduction of β appeared to be in favor of a lessened spatial overlap of the π -stacked motif (Revised Supplementary Fig. 10b). In order to distinguish the changes in the spatial overlap in the π -stacked motif, we conducted the evolution of the parallel misalignment angle (σ) upon compression (Added Supplementary Table 4). The dislocations were activated unremittingly with the decreased σ from 64.73° (1 atm) to 60.99° (12.4 GPa), which directly indicated the reduced overlap of the π -stacked motif (Revised Supplementary Fig. 10c).

In addition, we are so sorry for our expressions confusing the reviewer. The word "radioluminescence" is intended to express the radiative transition, not ionizing radiation. We have made modifications to the expression in the revised manuscript.

The revised details are highlighted in red and can be found in Lines 143-146 of the revised manuscript, as well as revised Supplementary Fig.10, Added Supplementary Table 4 and their captions in the revised Supplementary Information.

Revised Supplementary Fig. 10. **a** Pressure-dependent β evolution of IPA. **b** Molecular packing along the b axis (left). The schematic diagram of the parallel misalignment angle (σ) of IPA (right). **c** Pressure-dependent σ evolution of IPA.

Added Supplementary Table 4. The σ values of IPA upon compression.

Pressure (GPa)	1 atm	1.5	3.3	4.8	6.2	7.2	8.0	10.0	12.4
σ (°)	64.73	63.64	62.86	62.67	62.36	61.68	61.61	61.41	60.99

Comments 5: Figure 2: panels b, c and d are exchanged.

Author reply: Thanks very much for the reviewer's professional knowledge and great patience to spot the mistakes. According to your suggestion, we have revised Figure 2 and its captions in the

revised manuscript. Furthermore, we have tried our best to check the whole manuscript and Supplementary Information to make them concise and clarity.

Revised Fig. 2 | Crystal structure evolution upon compression and decompression. **a** The compression rate of three lattice constants (a , b , c) at different pressures. **b** IR spectra of IPA in the region of C=O stretching vibrational mode $\nu(\text{C}=\text{O})$ at ambient conditions and after pressure was released from 20.0 GPa. **c** Schematic illustrations of the intramolecular dihedral angle (δ_1 and δ_2) between the benzene ring and carboxyl groups as well as the intermolecular dihedral angle (θ) between two benzene rings of the dimer connected by hydrogen bonds. **d** The Jablonski diagram. **e** Hydrogen bond binding energy of the pressure-treated IPA.

Comments 6: From the structural data the authors calculate the pressure evolution of the intra- and intermolecular dihedral angles. How do they achieve this result? They were able to refine the atomic positions from the XRD data? I don't think so looking to the XRD patterns quality. All the dihedral angles are reported to reduce of less than 1 degree, at the most 0.76 for the intermolecular one (the adverb triumphantly used here is ridiculous) but I would like to see the error bars on the data to understand their reliability. In addition, they never mention if, and in case which one, PTM used for the XRD experiments. This is crucial to evaluate the reliability of the data. Silicon oil has been used for the emission experiments, it is nearly hydrostatic only up few (5-6) GPa.

Author reply: We are grateful for the reviewer’s comments. In the present work, the structural information, as well as atomic positions have been collected for each pressure by Rietveld refinements of the ADXRD patterns using the General Structure Analysis System (GSAS) software with low R_p ($< 0.6\%$) and R_{wp} ($< 0.4\%$) values. Then, we measured the intramolecular dihedral angles δ_1 , δ_2 and intermolecular dihedral angle θ based on the GSAS refinements of the ADXRD patterns at different pressures. To further confirm the experimental results, we also conducted high-pressure structures using Vienna ab initio Simulation Package (VASP) simulation code, which also proved the reduction of the angles (δ_1 , δ_2 and θ) upon compression. The details are as follows:

The refinements of ADXRD patterns were performed using the GSAS software (Added Supplementary Fig. 9). The quality of the fitting between the experimental and calculated profile is assessed by the various R parameters like R_p (profile factor) and R_{wp} (weighted profile factor)¹.

$$R_p = \left\{ \frac{\sum_i |I_i^{obs} - I_i^{cal}|}{\sum_i w_i (I_i^{obs})} \right\} \quad (1)$$

$$R_{wp} = \left\{ \frac{\sum_i w_i (I_i^{obs} - I_i^{cal})^2}{\sum_i w_i (I_i^{obs})^2} \right\}^{\frac{1}{2}} \quad (2)$$

Where I_i^{obs} , I_i^{cal} , and “i” indicate the experimental, calculated, and total number of points respectively. And the “w” is the reciprocal of the variance of observation I_i^{obs} . Generally, it is best to obtain the R_{wp} to about 10% or lower².

Here, each crystal structure of IPA under different pressures was refined using the Rietveld method, where the final structure refinement turned out to converge successfully with a low R value. The R parameters on the ADXRD data were provided to estimate the reliability in Added Supplementary Table 3. The Rietveld refinements of the ADXRD patterns at ambient conditions agreed well with the experimental data ($R_{wp} = 0.48\%$, $R_p = 0.30\%$). And the refinements of the ADXRD pattern at high pressure also match the experimental data points very well as shown in Added Supplementary Fig. 9. We thus collected the structural information, including the lattice parameters, intramolecular and intermolecular dihedral angles (δ_1 , δ_2 , and θ), etc, at various pressures based on these refinements. In the present work, the δ_1 decreased from 4.45° to 4.00° as pressure increased to 12.4 GPa. And the δ_2 reduced from 2.77° at 1 atm to 2.50° at 12.4 GPa. After releasing pressure completely, the δ_1 was measured as 3.86° , which was 0.59° smaller than the original state. And δ_2 was measured as 2.40° , which was 0.37° smaller than the original state. These reduced intramolecular dihedral angles trigger the aromatic ring and carboxyl groups to form a better co-facial geometry after pressure treatment, benefiting the electronic delocalization and emission redshift³. In addition, the intermolecular dihedral angle (θ) was reduced by 0.76° when the pressure increased to 12.4 GPa. The lesser value of θ (4.73°) rendered the pressure-treated IPA an optimized planar configuration for a further intermolecular

population of π electrons⁴. We also provide additional error bars on the angles to understand their reliability in Figure R11. In fact, small structural changes could give rise to remarkable changes in physicochemical characteristics, which have been reported in previous works⁵⁻⁷. For example, Mara, M. et al. found that “the selected complexes exhibit large variations in the overall quantum yield despite only minor differences in the chemical structure of the connecting ligand backbones”⁵. Besides, “pressure-induced bulk superconductivity” occurred at a pressure of ~ 0.18 - 0.35 GPa in CrAs with slightly reduced 2-theta values (< 0.1 degrees) of (111) plane have been reported by Yu, Z. et al.⁶ and Wu, W. et al.⁷. Therefore, in the present work, the small changes in dihedral angles (δ_1 , δ_2 , and θ) first increase the planarity of the structure, which promotes the electron delocalization for low-energy π - π^* transitions. Then, the original $^3(n, \pi^*)$ type of triplet states (T_5 and T_6) were converted to a mixture of $^3(n, \pi^*)/^3(\pi, \pi^*)$. Such mixed configuration facilitates the ISC process from S_1 to T_6 or T_5 . A high fraction of triplet excitons was then generated for phosphorescence enhancement, where the phosphorescence efficiency increased from 7 % to 40 %.

To further confirm the experimental results, we also conducted a theoretical structural simulation under high pressure based on VASP simulation code in Added Supplementary Fig. 14. The value of δ_1 decreased from 6.06° to 4.05° as pressure increased to 12.4 GPa, and δ_2 reduced from 3.64° at 1 atm to 2.35° at 12.4 GPa. Meanwhile, the θ reduced from 4.84° to 0.90° upon compression to 12.4 GPa. Notably, the trend of these simulated dihedral angles at high pressure is consistent with the experimental results. Because of the temperature difference between the simulation (0 K) and the experiments (~ 300 K), the values of these dihedral angles were different between the experimental and theoretical data^{8,9}. After pressure was completely released, we also obtained narrowed dihedral angles with $\delta_1 = 5.04^\circ$, $\delta_2 = 3.08^\circ$ and $\theta = 4.81^\circ$. The trends toward lower values further indicate an increase in the degree of planarity after pressure treatment, which enhances the conjugation of the pressure-treated IPA.

In addition, the PTM used for the XRD experiments is silicon oil. According to previous reports, the hydrostatic pressure of silicone fluid in diamond anvil cells can be maintained up to about 15 GPa (Figure R12)^{10,11}. Therefore, in our present study, the hydrostatic pressure environment was maintained relatively well during the whole compression process.

The corresponding revised details are highlighted in red and can be found in Lines 138-146, 184-207, 336, 364-376 of the revised manuscript and lines 103-115, Added Supplementary Fig. 9, 14 and Table 3 and their captions of the revised Supplementary Information.

Added Supplementary Fig. 9. Rietveld refinements of the experimental (red dot), simulated (blue profile), and difference (black line) ADXRD patterns at the pressure of 1 atm, 3.3 GPa, 6.2 GPa and released to ambient pressure. The red vertical markers indicate the corresponding Bragg reflections.

Figure R11 The dihedral angles (δ_1 , δ_2 and θ) with error bars of IPA at different pressures.

Added Supplementary Fig. 14. The calculated pressure-dependent dihedral angles (δ_1 , δ_2 and θ) of IPA based on simulated high-pressure structure using VASP simulation code.

[REDACTED]

Figure R12 Separation of ruby R₁-R₂ lines within different pressure medium, reported in previous work¹¹. This indicates that the hydrostatic pressure can be maintained up to about 15 GPa by using silicone oil as PTM.

Added Supplementary Table 3. The Rietveld refinement values of R_p (profile factor) and R_{wp} (weighted profile factor) of IPA upon compression and decompression. These values indicate the refinements at different pressures are good enough to obtain the structural information.

Pressure (GPa)	R _{wp}	R _p
1 atm	0.38 %	0.30 %

1.5	0.50 %	0.33 %
3.3	0.42 %	0.30 %
4.8	0.43 %	0.29 %
6.2	0.52 %	0.36 %
7.2	0.43 %	0.30 %
8.0	0.37 %	0.26 %
9.1	0.45 %	0.26 %
10.0	0.38 %	0.24 %
12.4	0.48 %	0.32 %
Released	0.32 %	0.20 %

References:

1. Hill, R. J. & Fischer, R. X. Profile agreement indices in Rietveld and pattern-fitting analysis. *J. Appl. Cryst.* **23**, 462-468 (1990).
2. Kumar, V., Kumari, S., Kumar, P., Kar, M. & Kumar, L. Structural analysis by Rietveld method and its correlation with optical properties of nanocrystalline zinc oxide. *Adv. Mater. Lett.* **6**, 139-147 (2015).
3. Bardak, F. et al. Conformational, electronic, and spectroscopic characterization of isophthalic acid (monomer and dimer structures) experimentally and by DFT. *Spectrochim Acta A* **165**, 33-46 (2016).
4. Gong, Y. et al. Crystallization-induced dual emission from metal and heavy atom-free aromatic acids and esters. *Chem. Sci.* **6**, 4438 (2015).
5. Mara, M. et al. Energy transfer from antenna ligand to europium (III) followed using ultrafast optical and X-ray spectroscopy. *J. Am. Chem. Soc.* **141**, 11071–11081 (2019)
6. Yu, Z. et al. Anomalous anisotropic compression behavior of superconducting CrAs under high pressure. *Proc. Natl. Acad. Sci. USA* **116**, 23404 (2019).
7. Wu, W. et al. Superconductivity in the vicinity of antiferromagnetic order in CrAs. *Nat. Commun.* **5**, 5508 (2014).
8. Clark, S. J. et al. First principles methods using CASTEP. *Z. Kristallogr.* **220**, 567–570 (2005).

9. Georgieva, I., Trendafilova, N., Dodoff, N. & Kovacheva, D. DFT study of the molecular and crystal structure and vibrational analysis of cisplatin. *Spectrochim Acta A* **176**, 58-66 (2017).
10. Ragan, D. D., Clarke, D. R. & Schiferl, D. Silicone fluid as a high-pressure medium in diamond anvil cells. *Rev. Sci. Instrum.* **67**, 494-496 (1996).
11. Shu-Jie, Y., Liang-Chen, C. & Chang-Qing, J. Hydrostaticity of pressure media in diamond anvil cells. *Chin. Phys. Lett.* **26**, 096202 (2009).

Comments 7: Concerning the HOMO-LUMO calculations is never specified which kind of system they computed: single molecule, cluster, crystal cell...How the pressure effects are computed? Imposing the lattice parameters? The recovered material is simulated as an ordered structure?

Author reply: Thanks very much for the reviewer's comments. Considering the calculations based on isolated molecular states cannot provide satisfactory results with experimental values¹, the present HOMO-LUMO calculations were computed based on the **single molecule under confined molecular crystal conditions**. In fact, fluorescent and phosphorescent materials are generally trapped in complicated environments, and interact with adjacent molecules continuously²⁻⁵. Therefore, the study of multimolecular systems needs to further consider the environmental effects on molecules. Since all charges around the central IPA molecule fit the electrostatic potential (Added Supplementary Table 6), so the HOMO-LUMO calculations are conducted in consideration of the charges from Electrostatic Potentials using a Grid based method (CHELPG). This method has been proved to have better electrostatic potential reproducibility than other atomic charge models, and could well reflect the electrostatic effect of environmental molecules on the central molecule^{6,7}.

In the present work, we have obtained the crystal structures by the GSAS refinement of the diffraction patterns at different pressures⁸⁻¹¹. And then, these refined structures were used to construct the structural models for HOMO-LUMO calculations of each pressure.

For the pressure-treated IPA, we have measured the ADXRD spectrum in Figure R13. All diffraction peaks positions are almost same with the initial ones except for the shift of (062) and (042) lattice planes. This indicates that the pressure-treated IPA still has an ordered periodic structure. And the symmetry of the pressure-treated structure doesn't change after the pressure treatment, which also belongs to monoclinic symmetry with space group $P 2_1/c$.

The corresponding revised details are highlighted in red and can be found in Lines 207, 209, 212-215 of the revised manuscript and Added Supplementary Table 6 and its captions of the revised Supplementary Information.

Figure R13 ADXR D spectra of the pressure-treated IPA ($\lambda = 0.6199 \text{ \AA}$). The symmetry belongs to monoclinic symmetry with space group $P 2_1/c$, where $a=3.74 \text{ \AA}$, $b=16.25 \text{ \AA}$, $c=11.69 \text{ \AA}$, $\alpha=\gamma=90.00^\circ$, $\beta=90.37^\circ$.

Revised Supplementary Fig. 11. a The lattice planes of (062) and (042) in IPA. b ADXRD patterns of the IPA crystal upon decompression ($\lambda = 0.6199 \text{ \AA}$). c The 2-theta value of the selected lattice planes at 1 atm and after releasing pressure completely.

Added Supplementary Table 6. The calculated HOMO and LUMO of IPA at ground state upon compression and decompression.

Pressure	HOMO	LUMO
1 atm		

3.3 GPa		7.2 GPa		12.4 GPa		Released		
References:

1. Wang, S. P., Wu, X. Z., Kong, S. M., Bai, F. Q. & Zhang, H. X. Refine the evaluation of photophysical properties of organometallic chromophores under confined molecular crystal conditions. *Spectrochim Acta A* **275**, 121168 (2022).
2. Huang, C. & Li, B. Two luminescent iridium complexes with phosphorous ligands and their photophysical comparison in solution, solid and electrospun fibers: decreased aggregation-caused emission quenching by steric hindrance. *Materials* **14**, 5419 (2021).
3. Yu, W., Zhang, H., Yin, P.-A., Zhou, F., Wang, Z., Wu, W., Peng, Q., H. Jiang & Tang, B.Z. Restriction of conformation transformation in excited state: an aggregation-induced emission building block based on stable exocyclic C=N group. *iScience* **23**, 101587 (2020).
4. Gao, X., Zhao, L., Ding, M., Wang, X., Zhai, L. & Ren, X. Insight understanding into influence

- of binding mode of carboxylate with metal ion on ligand-centered luminescence properties in Pb-based coordination polymers, *Chin. Chem. Lett.* **32**, 2423–2426 (2021).
5. Zhang, K., Cai, L., Fan, J., Zhang, Y., Lin, L. & Wang, C.-K. Effect of intermolecular interaction on excited-state properties of thermally activated delayed fluorescence molecules in solid phase: A QM/MM study, *Spectrochim Acta Part A Mol Biomol Spectrosc* **209**, 248–255 (2019).
 6. Breneman, C. M. & Wiberg, K. B. Determining atom-centered monopoles from molecular electrostatic potentials. The need for high sampling density in formamide conformational analysis. *J. Comput. Chem.* **11**, 361–373 (1990).
 7. Tian, L. U. & Fei-Wu, C. Comparison of computational methods for atomic charges. *Acta Phys. Chim. Sin.* **28**, 1-18 (2012).
 8. Yin, X. et al. Doping of charge-transfer molecules in cocrystals for the design of materials with novel piezo-activated luminescence. *Chem. Sci.* **14**, 1479-1484 (2023).
 9. Dong, X., Oganov, A. R., Cui, H., Zhou, X. F. & Wang, H. T. Electronegativity and chemical hardness of elements under pressure. *Proc. Natl. Acad. Sci. USA* **119**, e2117416119 (2022).
 10. Sato, H., Abd Rahman, S. A., Yamada, Y., Ishii, H. & Yoshida, H. Conduction band structure of high-mobility organic semiconductors and partially dressed polaron formation. *Nat. Mater.* **21**, 910-916 (2022).
 11. Wang, X. et al. Pressure-induced iso-structural phase transition and metallization in WSe₂. *Sci. Rep.* **7**, 46694 (2017).

Comments 8: The only evidence provided of the enforcement of the hydrogen bond with pressure is the red shift of the carbonyl frequency. The shift is quantified in 14 cm⁻¹ up to 20 GPa. It would be nice to have an explanation of how they catch the peak maximum above 10 GPa since the spectra are extremely broad and unresolved. However, this shift is very small being <1%, and is much lower than of the recovered material (6 cm⁻¹) that could be simply due to residual stress in the sample. It would be also interesting to see the IR spectrum of the recovered material in order to judge which is the quality of the sample and if some reaction occurred.

Author reply: For the evidence of the enforcement of hydrogen bonds, we have provided IR, XRD, atomic positions and calculated hydrogen bonding energy in our revised manuscript,

please see the above reply to comment 1. In terms of the results of IR spectra, in order to see the evolution of $\nu(\text{C}=\text{O})$ mode under high pressure more clearly, we provide new IR spectra in Figure R14. Upon compression, the $\nu(\text{C}=\text{O})$ mode is gradually redshifted, while the $\nu(\text{CC})_{\text{ring}}$ mode at around 1611 cm^{-1} is gradually blue-shifted. This gives rise to the resulting broad peak at 18.1 GPa. In order to clearly understand the range of redshifts, the IR spectra at high pressure are fitted into two peaks. The final fitted peak position of $\nu(\text{C}=\text{O})$ mode at 20.0 GPa is 1682 cm^{-1} , which redshifted 14 cm^{-1} compared with the initial 1696 cm^{-1} .

In addition, a slight blueshift or redshift of $3\text{-}8\text{ cm}^{-1}$ has been reported to demonstrate the formation of hydrogen bonds¹. Therefore, the redshift of 14 cm^{-1} in our present work could demonstrate the enhancement of hydrogen bonds under high pressure²⁻⁴. In addition, we added the IR spectra of the pressure-treated IPA over 400 days (Figure R15), and the $\nu(\text{C}=\text{O})$ mode still has a redshift of 6 cm^{-1} compared to the initial state. This indicates that the enhanced hydrogen bonds harvested in the pressure-treated IPA is not the results of residual stress. By combing with other experimental data, the redshift of 6 cm^{-1} for IR spectra in the pressure-treated sample is sufficient to indicate that we successfully intercepted the enhanced hydrogen bond after pressure-treatment.

In order to judge the quality of the pressure-treated sample and if some reaction occurred, we have provided the IR spectra of the sample before and after pressure treatment in Figure R15. Comparing the peaks before and after pressure treatment engineering, the pressure-treated spectra show a redshifted $\nu(\text{C}=\text{O})$ mode (Figure R15b), while the rest modes are recovered to the initial state (Figure R15c). It can be found that the quality of the sample is still in good condition, and there is no reaction occurred in the pressure-treated IPA.

The corresponding revised details are highlighted in red and can be found in **Lines 170-183** of the revised manuscript and **revised Supplementary Fig. 8 and its captions** of the revised Supplementary Information.

Figure R14. Selected IR spectra of IPA upon compression. The red dash lines indicate the evolution of $\nu(\text{C}=\text{O})$ upon pressure. The blue dash lines indicate the evolution of $\nu(\text{CC})_{\text{ring}}$ upon pressure. ν : stretching.

Figure R15. a IR spectra of IPA in the range of 600-3800 cm^{-1} before and after pressure treatment. **b** IR spectra of IPA in the region of C=O vibrational modes before and after pressure treatment. **c** Selected IR spectra of IPA in the range of 600-1650 cm^{-1} and 2300-3800 cm^{-1} before and after pressure treatment.

Reference:

1. Hobza, P. & Havlas, Z. k. Blue-shifting hydrogen bonds. *Chem. Rev.* **100**, 4253–4264 (2000).
2. Reynolds, J. & Sternstein, S. S. Effect of pressure on the infrared spectra of some hydrogen-bonded solids. *J. Chem. Phys.* **41**, 47-50 (1964).
3. Wu, M. et al. Pressure-induced restricting intermolecular vibration of a herringbone dimer for significantly enhanced multicolor emission in rotor-free truxene crystals. *J. Phys. Chem. Lett.* **13**, 2493-2499 (2022).
4. Gu, Y. et al. Pressure-induced emission enhancement of carbazole: the restriction of intramolecular vibration. *J. Phys. Chem. Lett.* **8**, 4191-4196 (2017).

Comments 9: On the basis of the H-bond enforcement, they also speculate about the stabilization of the non-bonding orbitals and in the variations of the relevant rate constants characterizing the radiative and non radiative decays. Again all qualitative statements.

Author reply: We appreciate the reviewer for suggestions. In the present work, the conclusion for the stabilization of the non-bonding orbitals (*n*-orbital) was referred to those previously reported works^{1,2}. In fact, the *n*-orbital represents negative charge in the oxygen attracting the positive charge in hydrogen ($\text{O}\cdots\text{H}$). With increasing electrostatic interaction, the *n*-orbital energy was stabilized to a lower energy level¹. This is the origin of the orbital stabilization.

In order to quantitatively understand the changes of ISC, radiative, and non-radiative transitions before and after pressure treatment, we have calculated the values of ISC rate K_{ISC} , phosphorescence radiative rate K_{P} and non-radiative rate K_{nr} , etc in Supplementary Table 1 based on the following equation³,

$$\Phi_{\text{P}} = \Phi_{\text{ISC}} \frac{K_{\text{P}}}{K_{\text{P}} + K_{\text{nr}}} \quad (1)$$

$$\tau_{\text{P}} = \frac{1}{K_{\text{P}} + K_{\text{nr}}} \quad (2)$$

Based on the measured lifetimes and photoluminescence quantum yield (PLQY) of the pristine and pressure-treated sample, we can calculate the radiative decay rate constant of fluorescence $K_F = \Phi_F/\tau_F$, the non-radiative decay rate constant of fluorescence $K_{nr}' = (1-\Phi_F - \Phi_P)/\tau_F$, the ISC rate constant $K_{ISC} = \Phi_P/\tau_F$, the radiative decay rate constant of phosphorescence $K_P = \Phi_P/\tau_P$, the non-radiative decay rate constant of phosphorescence $K_{nr} = (1-\Phi_P)/\tau_P$. Consequently, we found that the K_{ISC} was enlarged from 0.010 ns⁻¹ to 0.059 ns⁻¹. The K_{nr} remarkably decreased from 1.047 s⁻¹ to 0.667 s⁻¹, and the K_P increased from 0.079 s⁻¹ to 0.444 s⁻¹ (Supplementary Table 1).

Supplementary Table 1. Photophysical data of IPA before and after pressure treatment.

Pressure (GPa)	PLQY (%)	Φ_F (%)	Φ_P (%)	τ_F (ns)	τ_P (s)	K_F (ns ⁻¹)	K_{ISC} (ns ⁻¹)	K_P (s ⁻¹)	K_{nr} (s ⁻¹)	K_{nr}' (s ⁻¹)
1 atm	19	12	7	6.81	0.8886	0.018	0.010	0.079	1.047	0.119
Released	75	35	40	6.83	0.9002	0.051	0.059	0.444	0.667	0.037

In the revised version, the revised details are highlighted in red and can be found in **Lines 178-183, 216-226, 231, 235 and 237** of the revised manuscript.

References:

1. Ma, H. et al. Electrostatic interaction-induced room-temperature phosphorescence in pure organic molecules from QM/MM calculations. *J. Phys. Chem. Lett.* **7**, 2893-2898 (2016).
2. Kwon, M. S. et al. Multi-luminescent switching of metal-free organic phosphors for luminometric detection of organic solvents. *Chemi. Sci.* **7**, 2359–2363 (2016)
3. Zhang, Z. Y. et al. A synergistic enhancement strategy for realizing ultralong and efficient room-temperature phosphorescence. *Angew. Chem. Int. Ed.* **59**, 18748-18754 (2020).

Comments 10: Lines 210-213: No details are provided on how they computed the hydrogen bond binding energy especially for the sample recovered after the pressure treatment. How do they compute this energy if the local structure of the recovered material is not known. Regardless of how this energy is computed, 1.35 eV is huge considering the minor effect (if any) observed on the C=O stretching frequency.

Author reply: Thanks very much for the reviewer's comments. The computed method of hydrogen bonding energy was provided in the Methods section of the manuscript. VASP was used for DFT calculations¹. Perdew-Burke-Ernzerhof (PBE) generalized gradient approximation (GGA) exchange-correlation energy functional was adopted in this work². The long-range vdW interactions between atoms were described by the DFT-D3 correction method in Grimme's scheme³. Moreover, we described the interaction between core electrons and valence electrons using the frozen-core projector-augmented wave (PAW) method with a cutoff energy of 500 eV⁴. The convergence tolerances of force and energy on each atom during structure relaxation were less than 0.02 eV/Å and 10⁻⁵ eV, respectively.

The energy of the two IPA molecules (E_1 and E_2) connected by hydrogen bonds was calculated respectively. The energy of the dimer was calculated as E_3 . The hydrogen bonding energy is the $E = (E_1 + E_2) - E_3$. Based on the experimental ADXRD spectra, the structures before and after pressure treatment were refined by Rietveld refinements of the ADXRD patterns using the GSAS software. The pressure-treated IPA still has an ordered periodic structure (Figure R13). And the symmetry of the pressure-treated structure does not change after the pressure treatment, which also belongs to monoclinic symmetry with space group $P 2_1/c$. **Their hydrogen bond bonding energies are thereby calculated by using the known structures.**

Here, **the value of 1.35 eV represents the hydrogen bond bonding energy of pressure-treated IPA after pressure treatment.** It cannot be regarded as an enhanced energy variation of hydrogen bonds.

The revised details are highlighted in red and can be found in **Lines 241-247** of the revised manuscript.

Figure R13 ADXR D spectra of the pressure-treated IPA ($\lambda = 0.6199 \text{ \AA}$). The symmetry belongs to monoclinic symmetry with space group $P 2_1/c$, where $a=3.74 \text{ \AA}$, $b=16.25 \text{ \AA}$, $c=11.69 \text{ \AA}$, $\alpha=\gamma=90.00^\circ$, $\beta=90.37^\circ$.

References:

1. Kresse, G. & Furthmüller, J. Efficient iterative schemes for ab initio total-energy calculations using a plane-wave basis set. *Phys. Rev. B* **54**, 11169 (1996).
2. Perdew, J. P., Burke, K. & Ernzerhof, M. Generalized gradient approximation made simple. *Phys. Rev. Lett.* **77**, 3865 (1996).
3. Grimme, S., Antony, J., Ehrlich, S. & Krieg, H. A consistent and accurate ab initio parametrization of density functional dispersion correction (DFT-D) for the 94 elements H-Pu. *J. Chem. Phys.* **132**, 154104 (2010).
4. Kresse, G. & Joubert, D. From ultrasoft pseudopotentials to the projector augmented-wave method. *Phys. Rev. B* **59**, 1758 (1999).

Comments 11: All the discussion about the absorption spectrum (lines 227-236) is absolutely nonsense. They cannot estimate the band gap from an out-of-scale spectrum. They must measure the peak maximum since the onset of the absorption not only depends on the peak maximum but also on the intensity. According to their method, Figure 13 in the Supplementary material, the onset can just apparently move to the red when the absorption band intensifies.

Author reply: Thanks for the reviewer's comments. Following the reviewer's suggestions, we have added unsaturated absorption spectrum evolution upon compression and decompression. The redshift of the absorption peak with pressure below 1.8 GPa proves that the band gap indeed decreases with pressure. At 1 atm, the peak position was measured at 296 nm. Upon compression to 1.8 GPa, the absorption peak moved to the long wavelength at 299 nm and the absorption edge also redshifted (Figure R16). As pressure further increased, we found that the cut-off of absorption intensity at high pressure was inevitable, and the absorption edge continuously redshifted. After releasing pressure completely, the redshifted broadband absorption spectrum was also retained to the ambient conditions in Figure R17.

Notably, in order to obtain the optical band gap of the **solid material**, it is necessary to measure the absorption edge¹⁻³. It is well known that photons can be absorbed to excite electrons of a **solid system** from valence band to conduction band. The energy ($h\nu$) from the absorbed photon should meet the following equation (1):

$$h\nu = \frac{hc}{\lambda} \geq E_g \quad (1)$$

where $h\nu$ is the photon's energy, h is the Planck's constant, ν is the frequency, c is the speed of light, λ is the corresponding wavelength and E_g is the optical band gap. **This indicates the existence of a long-wavelength limit value $\lambda_0=hc/E_g$, which is the absorption edge of the spectra.** Therefore, the optical band gap of IPA is estimated based on the absorption edge by using the Tauc method.

The Tauc method is based on the assumption that the energy-dependent absorption coefficient α can be expressed by the following equation (2):

$$(\alpha h\nu)^{1/m} = B(h\nu - E_g) \quad (2)$$

where α is the absorption coefficient and B is a constant. The m factor depends on the nature of the electron transition and is equal to 1/2 or 2 for the direct and indirect transition band gaps, respectively^{4,5}.

The band gap of IPA belongs to direct band gap as shown in Figure R18. Therefore, the m factor of IPA is equal to 1/2. We estimate the bandgap of the IPA by extrapolating the linear portion of the $(\alpha d h\nu)^2$ versus $h\nu$ curve in direct bandgap Tauc plots, where a is the absorption coefficient and d is the sample thickness. The x -axis intersection point of the linear fit of the Tauc plot gives an estimate of the optical band gap energy.

The revised details are highlighted in red and could be found in Lines 256, 257, 259-262 of the revised manuscript.

Figure R16 Absorption spectra of IPA under high pressure.

Figure R17 Absorption spectra of IPA before and after pressure treatment.

Figure R18 The calculated band structure of IPA. The calculated results show IPA crystal is a **direct-band gap** material.

References:

1. Sinclair, G. S., Claridge, R. C. M., Kukor, A. J., Hopkins, W. S. & Schipper, D. J. N-Oxide S-O chalcogen bonding in conjugated materials. *Chem. Sci.* **12**, 2304-2312 (2021).
2. Al-Zaabi, U. A., Al-Busafi, S. N., Rasbi, N. K. A. & Suliman, F. O. Synthesis and characterization of tris(5,7-diphenyl-8-quinolinolato) aluminum(III), gallium(III), and indium(III) complexes: Effect of metal ions on the structural, photoluminescence, thermal and electrochemical properties. *J. Mol. Struct.* **1283** (2023).
3. Costa, J. C. S., Taveira, R. J. S., Lima, C. F. R. A. C., Mendes, A. & Santos, L. M. N. B. F. Optical band gaps of organic semiconductor materials. *Opt. Mater.* **58**, 51-60 (2016).
4. J. Tauc, R. Grigorovici & A. Vancu, Optical properties and electronic structure of amorphous germanium. *Phys. Status. Solidi. B* **15**, 627–637 (1966).
5. Makula, P., Pacia, M. & Macyk, W. How to correctly determine the band gap energy of modified semiconductor photocatalysts based on UV-Vis spectra. *J. Phys. Chem. Lett.* **9**, 6814-6817 (2018).

Reply to Reviewer #2

Comments: This work realizes the harvesting of bright white-light emission through modulating the balanced population of singlet/triplet excitons in an organic small molecule IPA through high pressure processing. Through both calculation and experiments, it demonstrates that the enhanced hydrogen bonds boosts the mixture of n-pi*/pi-pi* transition configurations that is in favor of the promoted intersystem crossing for white-light emission. The result is unique, interesting, and important. The Manuscript is well written and should be published in Nature Communications after minor revisions.

Author reply: Thanks for the reviewer's kind comments with the positive affirmation. First of all, we acknowledge your comments and suggestions very much, which are valuable in improving the quality of our manuscript. We revised our manuscript in accordance with your instructive guidance. Please see below for the details of all the improvements.

Comments 1: The authors obtained a brilliant white-light emission based on balanced fluorescence and phosphorescence species after decompression from 18 GPa. What about decompression from lower pressures? Any detailed data would be valuable.

Author reply: We highly appreciate the reviewer for the insightful comments. According to the advice, we added *in-situ* high-pressure PL experiments to investigate the PL evolution upon decompression from lower pressures. Figure R19 showed the different PL spectra of IPA upon decompression from 5.7 GPa and 10.1 GPa, separately. Upon decompression from 5.7 GPa and 10.1 GPa, the PL spectra signified enhanced blue-light emission that was ~1.6 and ~1.2 folds enhancement compared to the original values. Notably, the fluorescent emission was the main component in the pressure-treated PL spectra. When the pressure was completely released from 18.0 GPa, the pressure-treated emission intensity was about 6 times higher than the initial state, accompanied by a balanced population of fluorescence and phosphorescence species for white-light emission (Fig. 1d in revised manuscript). This phenomenon indicated that a stronger PL intensity than the initial state could also be obtained after decompression from lower pressures such as 5.7 GPa or 10.1 GPa. However, the realization of high-performance white-light emission based on balanced population of these two species (fluorescence and phosphorescence) in IPA needs to be decompressed from 18.0 GPa.

Figure R19. **a** PL spectra were collected before and after decompression from 5.7 GPa. **b** PL spectra were collected before and after decompression from 10.1 GPa.

Comments 2: The PL intensity continuously increases before 9.0 GPa due to the red-shift of C=O stretching vibration. At higher pressures, this mode continuously red-shifts, while the PL intensity decreases. Discussion on the possible reason/mechanism would be appreciated by the reader.

Author reply: Thanks very much for the reviewer's constructive comments. The non-monotonic increase of luminescence with pressure is derived from the competition of hydrogen-bond interaction and π - π interaction. **The enhanced hydrogen bonds play a positive role in increasing PL intensity at lower pressure, while the enhancement of π - π interaction would decrease and quench the emission at higher pressure.** The details are as follows:

The stretching vibration of C=O bonds maintained continuous red-shift upon compression to 20.0 GPa, suggesting the hydrogen bonds were strengthened over the entire compression process, which should promote emission enhancement monotonously. However, the gradually pressed closer packing of the π -stacked motif would decrease the emission. In this regard, we have added additional *in situ* high-pressure Raman measurements to directly identify the evolution of π - π interaction upon compression and decompression (Figure R2). The peaks located at 1001 cm^{-1} signified $\nu(\text{CC})_{\text{ring}}$ stretching vibration. Upon compression to around 7.0 GPa, the $\nu(\text{CC})_{\text{ring}}$ mode split into two peaks. This phenomenon suggests the strong π - π interaction and tighter π -stacked motif due to the compressed

configuration, which could decrease the emission intensity^{1,2}. Therefore, the non-monotonic increase of luminescence with pressure is attributed to the competition of hydrogen-bond interaction and π - π interaction. The enhanced hydrogen bonds play a leading role in increasing PL intensity below 9.0 GPa. As the pressure further increased, the enhancement of π - π interaction would decrease the emission intensity.

In addition, after pressure was completely released, the $\nu(\text{CC})_{\text{ring}}$ mode recovered to the initial state, indicating that the negative effect did not survive down to the ambient conditions. Notably, the enhanced hydrogen bonds have been harvested in the pressure-treated structure, in which the stretching vibration of C=O bonds was redshifted by 6 cm^{-1} (Revised Fig. 2b). Therefore, after pressure treatment, the pressure-treated IPA exhibits an enhanced emission with PLQY of 75 %.

Figure R2. Raman spectra upon compression and decompression. The red arrow indicates $\nu(\text{CC})_{\text{ring}}$ stretching vibration.

Revised Fig. 2 | Crystal structure evolution upon compression and decompression. **a** The compression rate of three lattice constants (a , b , c) at different pressures. **b** IR spectra of IPA in the region of C=O stretching vibrational mode $\nu(\text{C}=\text{O})$ at ambient conditions and after pressure was released from 20.0 GPa. **c** Schematic illustrations of the intramolecular dihedral angle (δ_1 and δ_2) between the benzene ring and carboxyl groups as well as the intermolecular dihedral angle (θ) between two benzene rings of the dimer connected by hydrogen bonds. **d** The Jablonski diagram. **e** Hydrogen bond binding energy of the pressure-treated IPA.

References:

1. Hu, Y., Kazemian, H., Rohani, S., Huang, Y. & Song, Y. In situ high pressure study of ZIF-8 by FTIR spectroscopy. *Chem. Commun.* **47**, 12694 (2011).
2. Ma, H. et al. Electrostatic interaction-induced room-temperature phosphorescence in pure organic molecules from QM/MM calculations. *J. Phys. Chem. Lett.* **7**, 2893-2898 (2016).

Comments 3: The white-light emission remained for about 400 days after releasing from pressure. What was the storage/environmental condition/s the sample is located?

Author reply: We are grateful for the reviewer's constructive suggestions. We stored the pressure-treated samples at ambient conditions. The corresponding temperature was about 18-25 °C, and the

humidity was about 40 %-50 %.

We added the storage conditions of the pressure-treated samples in the revised Supplementary Information. The revised details are highlighted in red and could be found in **Lines 44-45** of the revised Supplementary Information.

Comments 4: The binding energy of the hydrogen bond before pressure processing should be provided to clarify the enhancement of hydrogen bond before and after compression.

Author reply: We are grateful for the reviewer's constructive suggestions. According to the advice, we provide the hydrogen bond binding energy of IPA before pressure treatment, and the calculated result is 1.32 eV.

Comments 5: Some minor errors or typos, examples below:

- a. Page 3 line 45, "remains as a grand challenge" to "as remains a grand challenge";
- b. The representation of T₅ state in figure 4b should be consistent with the one in Supplementary (Figure 14);
- c. Page 3, line 42 "as well as color variability" to "as well as to enable color variability"?

Author reply: Thanks for the helpful comments. According to your suggestion, we have corrected all these errors and revised Fig. 4 in revised manuscript. We also tried our best to check the whole manuscript and Supplementary Information to make them concise and clarity.

The revised details are highlighted in red and can be found in **Lines 42 and 45** of the revised manuscript. The **revised Fig. 4 and its captions** were also shown in the revised manuscript.

Reply to Reviewer #3

Comments: The authors report an hybrid fluorescence and phosphorescence white-light emission with a 75% photoluminescence quantum yield. Through compression and decompression of isophthalic acid (an organic molecule) they are able to mix $n-\pi^*/\pi-\pi^*$ transition configurations, leading to balanced singlet and triplet excitons distribution. The hybrid fluorescence/phosphorescence phenomena has been extensively studied, and most recently the influence of the pressure on fluorescence (Nature Communications, 13, 5234 (2022)). The main contribution of the work presented is mixing both worlds, pressure and hybrid fluorescence/phosphorescence.

Author reply: Thanks for the reviewer's kind comments with the positive affirmation. First of all, we acknowledge your comments and suggestions very much, which are valuable in improving the quality of our manuscript. We carefully studied the references mentioned by the reviewer¹, which is very instructive to our work, so it is cited and highlighted in red and can be found in Lines 61, 486-488 of the revised manuscript. We revised our manuscript in accordance with your instructive guidance. Please see below for the details of all the improvements.

References:

1. Tong, S. et al. Fluorescence-based monitoring of the pressure-induced aggregation microenvironment evolution for an AIEgen under multiple excitation channels. *Nat. Commun.* **13**, 5234 (2022)

Comments 1: The results are well presented, although the hydrogen-bonding/electron delocalization effects should be better explained.

Author reply: We are grateful for the reviewer's constructive suggestions, which are valuable in improving the quality of our manuscript.

First of all, according to the suggestions of the authors, we explain in detail the mechanism of these two effects under this system. In the present work, the hydrogen-bonding effects in the isophthalic acid (IPA) are mainly contributed by electrostatic interactions¹⁻³. After pressure treatment, the hydrogen-bond distance $H_{16}\cdots O_2$ (d_1) decreased from 1.624 Å to 1.613 Å and $H_5\cdots O_3$ (d_2) decreased from 1.705 Å to 1.694 Å (Added Supplementary Fig. 12c). This suggests that the pressure-treated hydrogen bonds

were stronger than the initial state at 1 atm⁴. Based on the obtained X-ray diffraction pattern and IR spectra of the pressure-treated sample, the harvested enlarged 2-theta value of (062) and (042) planes, as well as the rewarding redshift of 6 cm⁻¹ of $\nu(\text{C}=\text{O})$, all these data further manifest the reduced hydrogen-bond distance and enhanced hydrogen bonds in pressure-treated structure. **As the hydrogen bonds strengthen, the positive charge on the hydrogen atom will reinforce the electrostatic attraction on the negative charge in the oxygen atom, thus lowering the energy of the *n*-orbitals provided by the oxygen atoms⁵.**

More importantly, the enhanced hydrogen bonds not only stabilize the *n*-orbitals to the lower energies but also **optimize the planar configuration for electron delocalization of the structure**. After pressure treatment, the strengthened hydrogen bonds reduced the intramolecular dihedral angles (δ_1 and δ_2) between carboxyl group and benzene ring (Revised Fig. 2c). The aromatic ring and carboxyl groups thereby formed a better co-facial geometry after pressure treatment, where the δ_1 decreased from 4.45° to 3.86° and δ_2 decreased from 2.77° to 2.40°. Such intramolecular planarity effectively promoted electronic delocalization in pressure-treated IPA. In addition, the reduced intermolecular dihedral angle θ of 4.73° rendered the pressure-treated IPA an optimized planar configuration, which further benefited the intermolecular population of π electrons⁶. Moreover, the calculated HOMO and LUMO further proved the electronic delocalization after pressure treatment (Added Supplementary Table 6). In this case, **the enhanced hydrogen bonds triggered the reduction of intramolecular and intermolecular dihedral angles of pressure-treated IPA, leading to the enhanced structural planarization. Such planarization configuration further promoted electron delocalization with low-energy π - π^* transitions.**

In conclusion, the enhanced hydrogen bonds synergistically lowered the energy of *n*-orbitals via stronger electrostatic interaction and promoted electron delocalization through structural planarization. Affected by the above changes of orbital energies, the original $^3(n, \pi^*)$ type of triplet states (T_5 and T_6) was converted to a mixture of $^3(n, \pi^*)/^3(\pi, \pi^*)$ (Revised Fig. 4). Such mixed configuration could facilitate the ISC process from S_1 to T_6 or T_5 based on El-Sayed's rule⁷. A high fraction of triplet excitons was then generated for phosphorescence enhancement. In addition, the enhanced hydrogen bonds also suppress the non-radiative dissipation, further benefiting the enhancement of both fluorescence and phosphorescence after pressure treatment. In this regard, we

could boost and balance the population of singlet and triplet excitons and subsequently reap hybrid fluorescence/phosphorescence white-light emission in the pressure-treated IPA.

The revised details are highlighted in red and could be found in Lines 138-207, 209, 212, 215, 216-220, 302-309, 312, Revised Fig. 2, 4 and its captions of the revised manuscript, as well as Added Supplementary Table 6 in Supplementary Information.

Added Supplementary Fig. 12. **a** Pressure-dependent hydrogen-bond distance $H_{16}\cdots O_2$ (d_1) and $H_5\cdots O_3$ (d_2) evolution of IPA. **b** Pressure-dependent $O_4-H_{16}\cdots O_2$ angle (α_1) and $O_1-H_5\cdots O_3$ angle (α_2) evolution of IPA. **c** Schematic diagram of hydrogen bonds of IPA before and after pressure treatment. The hydrogen-bond distance (d_1 and d_2) and angles (α_1 and α_2) are also labeled for clarity.

Revised Fig. 2 | Crystal structure evolution upon compression and decompression. **a** The compression rate of three lattice constants (a , b , c) at different pressures. **b** IR spectra of IPA in the region of C=O stretching vibrational mode $\nu(\text{C}=\text{O})$ at ambient conditions and after pressure was released from 20.0 GPa. **c** Schematic illustrations of the intramolecular dihedral angle (δ_1 and δ_2) between the benzene ring and carboxyl groups as well as the intermolecular dihedral angle (θ) between two benzene rings of the dimer connected by hydrogen bonds. **d** The Jablonski diagram. **e** Hydrogen bond binding energy of the pressure-treated IPA.

Revised Fig. 4 | The SOC and NTOs of the pristine and pressure-treated IPA. **a** The calculated ζ ($\zeta(S_1-T_6)$, $\zeta(S_1-T_5)$, $\zeta(S_1-T_4)$, $\zeta(S_1-T_3)$, $\zeta(S_1-T_2)$, $\zeta(S_1-T_1)$) for the pristine and pressure-treated IPA. **b** The NTOs of T₅ and T₆ state in the pristine and treated IPA. **c** Schematic representation of hydrogen bonds acting on ISC process.

Added Supplementary Table 6. The calculated HOMO and LUMO of IPA at ground state upon compression and decompression.

Pressure	HOMO	LUMO
1 atm		
3.3 GPa		

7.2 GPa		12.4 GPa		Released		
References:

1. Cai, Y. et al. Structural basis for stereoselective dehydration and hydrogen-bonding catalysis by the SAM-dependent pericyclase LepI. *Nat. Chem.* **11**, 812-820 (2019).
2. Saeedi, M. et al. Novel N'-substituted benzylidene benzohydrazides linked to 1,2,3-triazoles: potent α -glucosidase inhibitors. *Sci. Rep.* **13** (2023).
3. Emamian, S., Lu, T., Kruse, H. & Emamian, H. Exploring nature and predicting strength of hydrogen bonds: a correlation analysis between atoms-in-molecules descriptors, binding energies, and energy components of symmetry-adapted perturbation theory. *J. Comput. Chem.* **40**, 2868-2881 (2019).
4. Shi, B. et al. Short hydrogen-bond network confined on COF surfaces enables ultrahigh proton conductivity. *Nat. Commun.* **13**, 6666 (2022).
5. Ma, H. et al. Electrostatic interaction-induced room-temperature phosphorescence in pure organic molecules from QM/MM calculations. *J. Phys. Chem. Lett.* **7**, 2893-2898 (2016).
6. Gong, Y. et al. Crystallization-induced dual emission from metaland heavy atom-free aromatic acids and esters. *Chem. Sci.* **6**, 4438 (2015)

7. El-Sayed, M. A. Triplet state. Its radiative and non-radiative properties. *Acc. Chem. Res.* **1**, 8–16 (1968)

Comments 2: The triplet transitions should also be also more clearly explained, to be understood for a wider audience.

Author reply: We highly appreciate the reviewer for the insightful comments. In fact, triplets are the products of singlet exciton fission, which are ubiquitous but often tend to be an undesirable energy sink because they are spin-forbidden from emitting light¹. Therefore, **forming and stabilizing triplet states** is the key to triplet transitions (Figure R20). To date, efficient intersystem crossing (ISC) and suppressing non-radiative dissipation have been employed as primary concerns to bright triplet excitons for efficient phosphorescent emission^{2,3}. In this regard, large spin-orbital coupling (SOC) and narrowed singlet-triplet energy gap (ΔE_{ST}) have been regarded as necessary for efficient ISC⁴⁻⁷. Rigid crystalline solid-state structures or ultralow temperatures are also needed as the triplet excitons generated in organic molecules are highly sensitive to oxygen and temperature⁸⁻¹⁰.

In the present work, we not only accelerate the ISC with large SOC and narrowed ΔE_{ST} , but also suppress the non-radiative dissipation for efficient triplet transitions by using enhanced hydrogen bonds. First of all, the enhanced hydrogen bonds lower the energy of n -orbitals via enhanced electrostatic attraction¹¹. In addition, it also enhances the degree of conjugation accompanied by low-energy π - π^* transitions in pressure-treated IPA. These changes in orbital energies would further affect the energy level and nature transition-orbital (NTOs) of the singlet and triplet states. In fact, the original ${}^3(n, \pi^*)$ type of triplet states (T_5 and T_6) was converted to a mixture of ${}^3(n, \pi^*)/{}^3(\pi, \pi^*)$ (Revised Fig. 4), which increased their SOC constants $\zeta(S_1-T_5)$ and $\zeta(S_1-T_6)$. The $\zeta(S_1-T_5)$ increased from 4.259 cm^{-1} to 23.836 cm^{-1} and $\zeta(S_1-T_6)$ increased from 0.007 cm^{-1} to 0.076 cm^{-1} (Supplementary Table 9 in the revised Supplementary Information). In addition, the energy gap $\Delta E(S_1-T_5)$ and $\Delta E(S_1-T_6)$ narrowed from 0.3925 eV and 0.3359 eV to 0.3526 eV and 0.3237 eV (Supplementary Table 8 in the revised Supplementary Information). Based on the perturbation theory,

$$K_{ISC} \propto \xi^2 * \exp [-(\Delta E_{ST})^2] \quad (1)$$

the ISC process was thereby accelerated to boost triplet excitons population via the reduced ΔE_{ST} and enhanced SOC constants.

In addition, with the strengthened hydrogen bonds to suppress the non-radiative dissipation, the phosphorescence emission efficiency was further improved in the pressure-treated IPA. In order to quantitatively understand the changes of ISC, radiative and non-radiative transitions before and after pressure treatment, we calculated the values of ISC rate K_{ISC} , radiative rate K_P and non-radiative rate K_{nr} based on the following equation¹²,

$$\Phi_P = \Phi_{ISC} \frac{K_P}{K_P + K_{nr}} \quad (2)$$

$$\tau_P = \frac{1}{K_P + K_{nr}} \quad (3)$$

Based on the measured lifetimes and PLQY of the pristine and pressure-treated sample, we can calculate the radiative decay rate constant of fluorescence $K_F = \Phi_F/\tau_F$, the non-radiative decay rate constant of fluorescence $K_{nr}' = (1 - \Phi_F - \Phi_P)/\tau_F$, the ISC rate constant $K_{ISC} = \Phi_P/\tau_F$, the radiative decay rate constant of phosphorescence $K_P = \Phi_P/\tau_P$, the non-radiative decay rate constant of phosphorescence $K_{nr} = (1 - \Phi_P)/\tau_P$. Consequently, we found that the K_{ISC} was enlarged from 0.010 ns⁻¹ to 0.059 ns⁻¹. The K_{nr} remarkably decreased from 1.047 s⁻¹ to 0.667 s⁻¹, and the K_P increased from 0.079 s⁻¹ to 0.444 s⁻¹ (Supplementary Table 1 in revised Supplementary Information). Therefore, the enhanced hydrogen bonds indeed suppress the non-radiative dissipation for efficient fluorescence and phosphorescence enhancement.

In summary, after pressure treatment, the enhanced hydrogen bonds synergistically lower the energy of n -orbitals via stronger electrostatic interaction and reduced the energy of π - π^* transitions through structural planarization. Affected by the above changes of orbital energies, the original ³(n , π^*) type of triplet states (T_5 and T_6) was converted to a mixture of ³(n , π^*)/³(π , π^*). Such mixed configuration could facilitate the ISC process from S_1 to T_6 or T_5 . A high fraction of triplet excitons was then generated for phosphorescence enhancement. Furthermore, the enhanced hydrogen bonds also suppress the non-radiative dissipation, further benefiting the enhancement of both fluorescence and phosphorescence after pressure treatment. In this regard, we could boost and balance the population of singlet and triplet excitons and subsequently reap hybrid fluorescence/phosphorescence white-light emission in the pressure-treated IPA.

The revised details are highlighted in red and could be found in Lines 178-183, 216-226, 231, 235, 237, 273, 275-281, 289, 291, 293, 295, 297, 301-309 and 312 of the revised manuscript.

Figure R20 Schematic illustrations of singlet and triplet transitions.

References:

1. Thompson, N. J. et al. Energy harvesting of non-emissive triplet excitons in tetracene by emissive PbS nanocrystals. *Nat. Mater.* **13**, 1039-1043 (2014).
2. Kenry, Chen, C. & Liu, B. Enhancing the performance of pure organic room-temperature phosphorescent luminophores. *Nat. Commun.* **10**, 2111 (2019).
3. Zhang, Z. Y. et al. A synergistic enhancement strategy for realizing ultralong and efficient room-temperature phosphorescence. *Angew. Chem. Int. Ed.* **59**, 18748-18754 (2020).
4. Cai, S. et al. Visible-light-excited ultralong organic phosphorescence by manipulating intermolecular interactions. *Adv. Mater.* **29** (2017).
5. Fan, Y. et al. Multi-photoresponsive triphenylethylene derivatives with photochromism, photodeformation and room temperature phosphorescence. *Mater. Horiz.* **9**, 368-375 (2022).
6. Wang, T. et al. Aggregation-induced dual-phosphorescence from organic molecules for nondoped light-emitting diodes. *Adv. Mater.* **31**, e1904273 (2019).
7. Jin, J. et al. Thermally activated triplet exciton release for highly efficient tri-mode organic afterglow. *Nat. Commun.* **11**, 842 (2020).
8. An, Z. et al. Stabilizing triplet excited states for ultralong organic phosphorescence. *Nat. Mater.* **14**, 685-690 (2015).
9. Jin, J. et al. Modulating tri-mode emission for single-component white organic afterglow. *Angew.*

Chem. Int. Ed. **60**, 24984-24990 (2021).

10. Ye, W. et al. Confining isolated chromophores for highly efficient blue phosphorescence. *Nat. Mater.* **20**, 1539-1544 (2021).
11. Ma, H. et al. Electrostatic interaction-induced room-temperature phosphorescence in pure organic molecules from QM/MM calculations. *J. Phys. Chem. Lett.* **7**, 2893-2898 (2016).
12. Zhang, Z. Y. et al. A synergistic enhancement strategy for realizing ultralong and efficient room-temperature phosphorescence. *Angew. Chem. Int. Ed.* **59**, 18748-18754 (2020).

In summary, following the kind suggestions and insightful comments of the reviewers, we carefully rechecked our manuscript and made some amendments to respond to the reviewers' issues. Hopefully, we have addressed all of your concerns. The corresponding revised details are highlighted in red and could be found in Lines 28, 42, 45, 51, 54, 61, 63-64, 67, 82, 84, 94-97, 113, 116, 135, 137-207, 209, 212-226, 231, 235, 237, 241-247, 250-254, 256, 257, 259-262, 269, 271, 273, 275-281, 284-287, 289, 291, 293, 295, 297, 301-309, 312, 320, 335, 336, 358, 364-376, 379, 380, 386, 387, 391, 393, 395, 397, 399, 407-408, 410-411, 445-446, 450-451, 471-472, 474-475, 477-481, 483-560 of the revised manuscript, as well as Lines 21-22, 25-26, 40, 44-45, 52, 60-63, 65-68, 70-73, 75-78, 80-95, 98, 100-115, 117, 120, 127, 131-133, 135, 137, 139-140, 142, 145, 148, 155-166 of the revised Supplementary Information.

REVIEWER COMMENTS

Reviewer #1 (Remarks to the Author):

In this revised version, the authors made a great effort to improve, correct and clarify many points of their research. Said that, I have still serious problems to follow their interpretation that appears, in my opinion, forced in attributing the emission changes to the hydrogen bond enforcement with pressure. To substantiate this idea they provide many many numbers all of them ultimately based on the structural data (I did not consider the IR data because the red shift is really poor to attest for a strong effect on the H-bond strength). I would like to clarify that I am not against their interpretation that hydrogen bonding plays an active role in the process but, in my opinion, to be convincing it is mandatory to provide convincing XRD data.

1. I can trust to their refinement but in order to be convinced I want to see the 2D pattern and judge from its quality how reliable is the Rietveld refinement. They never discuss which kind of sample they have: single crystal, powder, quality of the single crystal, quality of the powder. This can be easily judged by the 2D pattern and it is mandatory to evaluate their analysis. It should be clear that the refinement of the atomic position from powder diffraction data it is a demanding task.

2. The XRD data are the most important for the discussion. It is surprising they present them only on Figure 9 of the supplementary. There is no mention to the way they treated the baseline, and which was the origin of such curious pattern (long modulation).

3. The hydrostaticity of silicon oil is already quite poor between 5 and 10 GPa to improve a little bit between 10 and 12 GPa. Therefore, it is definitely not a good hydrostatic transmitting medium as it is well known within the high-pressure community. Obviously this poses some issues about the reliability of the "structural numbers" obtained and especially on the error bars the authors report (again it is not specified how do they get them and I presume simply computed by the fit uncertainty). The performance of silicon oil I am mentioning derive from a highly cited (>1000) paper: S Klotz et al 2009 J. Phys. D: Appl. Phys. 42 075413.

4. The authors recover a metastable material with different emission properties of the starting one and related to the changes induced by compression (non hydrostatic or anisotropic stress applied to the sample). Did the authors try a thermal annealing of the recovered material to understand how "stable" is the metastable form?

5. The absorption spectra. It would be nice to have an explanation of how the authors got in scale absorption spectra. I presume by reducing the sample thickness, but in which way? Anyway, Figure R16 and R17 do not report the Intensity scale: they cannot use arbitrary units for an absorption spectrum, the scale is in absorbance units and the absorbance values are fundamental to understand the reliability of the data. In addition, the application of the Tauc method is highly disputable in a non semiconducting system. A molecular system absorption structure is made by bands and to evaluate the band gap you need to identify the maximum.

As I said, the manuscript has been improved with respect to the original version but is not yet convincing. There is an extreme superficiality in treating the structural data especially overlooking the possible effects of non-hydrostatic or anisotropic compression. These issues do not alter the self-standing effect discovered, but make the meticulous description of the origin of this effect, attributed to the enforcement of the H-bonds, highly disputable.

Reviewer #2 (Remarks to the Author):

The authors answered all my questions and I am satisfied with all the responses. The paper should be published as is.

Reviewer #3 (Remarks to the Author):

The authors have addressed all the comments and answered the technical questions. I think the paper has improved substantially after revision. I suggest it to be published.

For your guidance, itemized response to reviewer' s comments is appended below.

Reply to Reviewer #1

Comments: In this revised version, the authors made a great effort to improve, correct and clarify many points of their research. Said that, I have still serious problems to follow their interpretation that appears, in my opinion, forced in attributing the emission changes to the hydrogen bond enforcement with pressure. To substantiate this idea they provide many numbers all of them ultimately based on the structural data (I did not consider the IR data because the red shift is really poor to attest for a strong effect on the H-bond strength). I would like to clarify that I am not against their interpretation that hydrogen bonding plays an active role in the process but, in my opinion, to be convincing it is mandatory to provide convincing XRD data.

Author reply: Thanks for the reviewer's kind comments with positive affirmation. We have added experimental 2D ring-type ADXRD patterns to further support our conclusion in accordance with your guidance. The vibrational frequencies of the stretching vibrations of C=O and O-H groups involved in hydrogen bonds can provide a clear-cut signature of the hydrogen-bonding dynamics^{1,2}. The formation of the hydrogen bonds always results in the weakening of the C=O and O-H bonds. This weakening is accompanied by bond elongation and a concomitant decrease of the corresponding stretch vibration frequency compared to the noninteracting species. Note that these "significant" changes of molecular properties upon complex formation are actually quite small: the change in energies, bond lengths, frequencies, and electron densities are two or more orders of magnitude smaller than typical chemical changes². For example, Boldeskul et al. reported that the formation of intermolecular complexes was accompanied by a 3-8 cm⁻¹ shift of the haloform C-H/D stretch vibration³. The redshift of the Si-H bond was estimated to be about 14.4 cm⁻¹ in the formation of Me₃Si-H···HCN Complex¹. Therefore, based on the present hydrogen bonds in the pristine IPA sample, the redshift of 6 cm⁻¹ for IR spectra (Revised Fig. 2b) in the pressure-treated sample is sufficient to indicate that we successfully intercepted the

enhanced hydrogen bonds after pressure treatment. In addition, we also added experimental data to further support our conclusion in accordance with your guidance. Please see below for the details of all the improvements.

References:

1. Civiš, S. et al. Hydrogen Bonding with Hydridic Hydrogen—Experimental Low-Temperature IR and Computational Study: Is a Revised Definition of Hydrogen Bonding Appropriate? *J. Am. Chem. Soc.* (2023).
2. Hobza, P. & Havlas, Z. k. Blue-shifting hydrogen bonds. *Chem. Rev.* **100**, 4253–4264 (2000).
3. Buděšínský, M., Fiedler, P. & Arnold, Z. Triformylmethane: An Efficient Preparation, Some Derivatives, and Spectra. *Synthesis* **1989**, 858-860 (1989).

Comments 1: I can trust to their refinement but in order to be convinced I want to see the 2D pattern and judge from its quality how reliable is the Rietveld refinement. They never discuss which kind of sample they have: single crystal, powder, quality of the single crystal, quality of the powder. This can be easily judged by the 2D pattern and it is mandatory to evaluate their analysis. It should be clear that the refinement of the atomic position from powder diffraction data it is a demanding task.

Author reply: We are grateful for the reviewer’s suggestions, we have added 2D ring-type ADXRD patterns of IPA upon compression and decompression in Revised Supplementary Fig. 11. In order to obtain better ADXRD profiles, the crystal IPA was ground into powder and then loaded into the symmetric diamond anvil cell (DAC) before the measurements. It is worth noting that the grinding is more conducive for us to get clear 2D diffraction rings as shown inset of the Revised Supplementary Fig. 11.

The crystal structure at ambient conditions in the present work was obtained from the Cambridge Crystallographic Data Center (CCDC) with CCDC number 1108747, where the atomic positions were known and definite. The refinement of the XRD patterns at ambient pressure further confirms this structure (Revised Fig. 2a). Based on this standard structure, the crystal structure under different pressures was therefore

refined by using GSAS software^{1,2}. Under high pressure, the symmetry of this phase has not changed, which belonged to monoclinic symmetry with space group $P2_1/c$. The compression process merely changed the atomic positions slightly from their original positions. The refined atomic positions at typical pressures are shown in below Table R1. We also performed optimization of atomic positions using Vienna ab initio Simulation Package (VASP)³ simulation code to further clarify the validity of the results. Both experimental and calculated data confirmed the strengthening of H-bonding.

The revised details are highlighted in red and can be found in Lines 376-377 of the revised manuscript, as well as Lines 86-90 in the revised Supplementary Information.

Revised Supplementary Fig. 11. Rietveld refinements (with baseline subtracted) of the experimental patterns at typical pressures. The red dot, blue line, and black line represent the experimental data, calculated data and difference, respectively. The red vertical markers indicate the corresponding Bragg reflections. The inset (right) shows corresponding 2D ring-type ADXRD pattern.

Table R1. Selected atomic positions of IPA sample upon compression and decompression.

Pressure	Lattice parameters	Atomic positions
1.5 GPa	$a=3.57 \text{ \AA}$ $b=16.20 \text{ \AA}$ $c=11.19 \text{ \AA}$ $\beta=88.90^\circ$ $\alpha=\gamma=90^\circ$	C ₁ (2.70, 5.56, 4.22) C ₂ (2.38, 4.35, 3.69) C ₃ (2.65, 3.19, 4.35) C ₄ (3.25, 3.25, 5.55) C ₅ (3.56, 4.46, 6.07) C ₆ (3.29, 5.61, 5.42) C ₇ (2.37, 6.78, 3.50) C ₈ (2.27, 1.91, 3.76) O ₁ (2.73, 7.87, 4.05) O ₂ (1.79, 6.75, 2.46) O ₃ (1.67, 1.89, 2.74) O ₄ (2.61, 0.85, 4.38)
3.3 GPa	$a=3.44 \text{ \AA}$ $b=16.02 \text{ \AA}$ $c=10.89 \text{ \AA}$ $\beta=88.58^\circ$ $\alpha=\gamma=90^\circ$	C ₁ (2.69, 5.50, 4.10) C ₂ (2.37, 4.30, 3.59) C ₃ (2.65, 3.15, 4.23) C ₄ (3.25, 3.21, 5.40) C ₅ (3.56, 4.41, 5.90) C ₆ (3.29, 5.55, 5.27) C ₇ (2.36, 6.70, 3.41) C ₈ (2.27, 1.89, 3.66) O ₁ (2.72, 7.79, 3.94) O ₂ (1.78, 6.67, 2.40) O ₃ (1.67, 1.87, 2.66) O ₄ (2.61, 0.84, 4.26)
8.0 GPa	$a=3.29 \text{ \AA}$ $b=15.92 \text{ \AA}$ $c=10.53 \text{ \AA}$ $\beta=85.77^\circ$ $\alpha=\gamma=90^\circ$	C ₁ (2.70, 5.46, 3.96) C ₂ (2.38, 4.27, 3.46) C ₃ (2.66, 3.13, 4.09) C ₄ (3.28, 3.19, 5.21) C ₅ (3.59, 4.38, 5.70) C ₆ (3.31, 5.51, 5.09) C ₇ (2.36, 6.66, 3.29) C ₈ (2.29, 1.87, 3.53) O ₁ (2.72, 7.74, 3.80) O ₂ (1.78, 6.63, 2.31) O ₃ (1.68, 1.86, 2.57) O ₄ (2.63, 0.84, 4.11)
Released	$a=3.74 \text{ \AA}$ $b=16.25 \text{ \AA}$ $c=11.69 \text{ \AA}$ $\beta=90.37^\circ$ $\alpha=\gamma=90^\circ$	C ₁ (2.70, 5.58, 4.40) C ₂ (2.39, 4.36, 3.86) C ₃ (2.66, 3.20, 4.55) C ₄ (3.26, 3.26, 5.80) C ₅ (3.56, 4.48, 6.34) C ₆ (3.30, 5.63, 5.66) C ₇ (2.38, 6.80, 3.66) C ₈ (2.28, 1.91, 3.93) O ₁ (2.74, 7.90, 4.23) O ₂ (1.81, 6.77, 2.57) O ₃ (1.69, 1.89, 2.86) O ₄ (2.61, 0.86, 4.58)

References:

1. Toby, B. H. EXPGUI, a graphical user interface for GSAS. *J. Appl. Cryst.* **34**, 210-213 (2001).
2. Yu, Z. et al. Anomalous anisotropic compression behavior of superconducting CrAs under high pressure. *Proc Natl Acad Sci U S A* **112**, 14766-14770 (2015).

3. Kresse, G. & Furthmüller, J. Efficient iterative schemes for ab initio total-energy calculations using a plane-wave basis set. *Phys. Rev. B* **54**, 11169 (1996).

Comments 2: The XRD data are the most important for the discussion. It is surprising they present them only on Figure 9 of the supplementary. There is no mention to the way they treated the baseline, and which was the origin of such curious pattern (long modulation).

Author reply: We appreciate the reviewer for the comments. According to the reviewer's suggestions, we provided the refined ADXRD results with baseline subtracted for IPA upon compression and decompression in Revised Supplementary Fig. 11 (also shown in the above figure). Before refinements of the ADXRD patterns using GSAS software, it is necessary to deduct the baseline as required¹. Further fitting of the baseline was also conducted during the refinement process. In addition, we have added the important XRD data in Revised Fig. 2.

The revised details are highlighted in red and can be found in Lines 145-166, 188, 200-202, 243 and 254 of the revised manuscript, as well as Lines 141-142, and 164 in the revised Supplementary Information.

Revised Fig. 2 | Crystal structure evolution upon compression and decompression.

a Rietveld refinement (with baseline subtracted) of the experimental ADXRD before and after pressure treatment. The red dot, blue line, and black line represent the experimental data, calculated data and difference, respectively. The red vertical markers indicate the corresponding Bragg reflections. The inset (right) shows corresponding 2D ring-type ADXRD patterns. **b** The compression rate of three lattice constants (a , b , c) at different pressures. **c** IR spectra of IPA in the region of C=O stretching vibrational mode $\nu(\text{C}=\text{O})$ at ambient conditions and after pressure was released from 20.0 GPa. **d** Schematic illustrations of the intramolecular dihedral angle (δ_1 and δ_2) between the benzene ring and carboxyl groups as well as the intermolecular dihedral angle (θ) between two benzene rings of the dimer connected by hydrogen bonds. **e** The Jablonski diagram. k_F and k_{nr} represent the rate constants of the radiative and non-radiative transitions from the lowest singlet-excited state (S_1), respectively; k_T and k_{nr} represent the rate constants of the radiative and non-radiative transitions from the lowest triplet-excited state (T_1), respectively; ΔE_{ST} is the singlet-triplet energy gap; k_{ISC} is the rate of ISC; S_0 is the ground state. Ab., absorption.

References:

1. Toby, B. H. EXPGUI, a graphical user interface for GSAS. *J. Appl. Cryst.* **34**, 210-213 (2001).

Comments 3: The hydrostaticity of silicon oil is already quite poor between 5 and 10 GPa to improve a little bit between 10 and 12 GPa. Therefore, it is definitely not a good hydrostatic transmitting medium as it is well known within the high-pressure community. Obviously this poses some issues about the reliability of the “structural numbers” obtained and especially on the error bars the authors report (again it is not specified how do they get them and I presume simply computed by the fit uncertainty). The performance of silicon oil I am mentioning derive from a highly cited (>1000) paper: SKlotz et al 2009 J. Phys. D: Appl. Phys. 42 075413.

Author reply: Thanks for the reviewer’s comments and suggestions. We are aware of

this literature that the reviewer mentioned. In fact, the hydrostatic limits of silicone oil were highly related to their viscosity¹⁻⁸. For example, Shen et al. found that the hydrostaticity of silicone oil with the viscosity of 1 cst could reach 20.0 GPa^{6,9}. In our present work, the applied silicone oil viscosity was 10cst. In order to determine the hydrostatic range of this current silicone oil, we specifically measured the full-width half maximum (FWHM) of R₁ line, R₁-R₂ separation, and the standard deviation of the pressures (σ) according to the previously reported method³ (Figure R1). We find that the σ before ~15.0 GPa is within 0.37 GPa, suggesting a good hydrostatic environment in the sample chamber (Figure R1b). The R₁-R₂ separation keeps steady at pressure < ~15.0 GPa but increases sharply when pressure gets higher (Figure R1c). The behavior of the FWHM of R₁ vs pressure implies a poor hydrostatic status in the sample chamber after ~15.0 GPa. Therefore, the hydrostatic pressure of the silicone oil we used is relatively good below ~15.0 GPa. Notably, the derived structural parameters in this work are obtained within ~12.4 GPa, where the σ is smaller than 0.30 GPa. Therefore, our used silicone oil as the transmitting medium (PTM), could guarantee a good hydrostatic pressure environment in our study range and won't influence the conclusion of the present work.

In the present work, the *in-situ* high-pressure PL experiments were first carried out with silicone oil as PTM. Therefore, we chose silicone oil as PTM in the following ADXRD experiments to ensure the identical experimental environment. In this regard, the relevant harvested structural parameters as well as the refined “structural numbers” could perfectly correspond to its optical properties. In addition, we added new error bars based on the pressure uncertainty (Revised Supplementary Fig. 16). With the increase of pressure, these dihedral angles of δ_1 , δ_2 , and θ still displayed a decreasing trend to form an optimized planarization configuration, which further enhanced the reliability of the harvested structural parameters.

On the other hand, in order to further investigate the influence of PTM on the PL properties of IPA, we have supplemented *in-situ* high-pressure PL experiment by using liquid argon (Revised Supplementary Fig. 5) and nitrogen (Revised Supplementary Fig. 6) as PTM, respectively. We found that the PL evolution from these two experimental

runs was basically consistent with that of silicone oil. After ~ 20.0 GPa of depressurization, we still harvest a bright white-light emission with liquid argon and nitrogen, respectively. Therefore, the different PTMs (silicone oil, liquid argon and liquid nitrogen) could not influence the present conclusions: the PL enhancement under high pressure and the harvest of white-light emission after pressure treatment.

The revised details are highlighted in red and can be found in Lines 118-127 of the revised manuscript, as well as Lines 42-61, 122-123 in the revised Supplementary Information.

Figure R1. **a** The fluorescence spectra of ruby at ambient conditions. **b** Pressure dependence of the standard deviation σ for silicone oil. The inset shows the σ changes below 12.0 GPa. Pressure dependence of **c** the R₁-R₂ separation and **d** the average change in R₁ line width (FWHM) for silicone oil.

Revised Supplementary Fig. 16. The dihedral angles **a** δ_1 , **b** δ_2 and **c** θ with error bars of IPA.

Revised Supplementary Fig. 5. **a** PL spectra of IPA with PTM of liquid argon upon compression to 8.8 GPa. **b** PL spectra of IPA upon compression from 8.8 GPa to 20.6 GPa. **c** PL spectra of IPA upon decompression. The right figure shows the corresponding PL photographs at ambient conditions and released from 20.6 GPa.

Revised Supplementary Fig. 6. **a** PL spectra of IPA with PTM of liquid nitrogen upon compression to 8.7 GPa. **b** PL spectra of IPA upon compression from 8.7 GPa to 20.4 GPa. **c** PL spectra of IPA upon decompression. The right figure shows the corresponding PL photographs at ambient conditions and released from 20.4 GPa.

References:

1. Shu-Jie, Y., Liang-Chen, C. & Chang-Qing, J. Hydrostaticity of Pressure Media in Diamond Anvil Cells. *Chin. Phys. Lett.* Vol. **26**, 096202 (2009).
2. Takemura, K. Evaluation of the hydrostaticity of a helium-pressure medium with powder x-ray diffraction techniques. *J. Appl. Phy.* **89**, 662-668 (2001).
3. Klotz, S., Chervin, J. C., Munsch, P. & Le Marchand, G. Hydrostatic limits of 11 pressure transmitting media. *J. Phy. D: Applied Physics* **42** (2009).
4. Chen, X. et al. Structural transitions of 4:1 methanol–ethanol mixture and silicone oil under high pressure. *Matter Radiat. at Extremes* **6** (2021).
5. Ragan, D. D., Clarke, D. R. & Schiferl, D. Silicone fluid as a high-pressure medium

- in diamond anvil cells. *Rev. Sci. Instrum.* **67**, 494-496 (1996).
6. Shen, Y., Kumar, R. S., Pravica, M. & Nicol, M. F. Characteristics of silicone fluid as a pressure transmitting medium in diamond anvil cells. *Rev. Sci. Instrum.* **75**, 4450-4454 (2004).
 7. Torikachvili, M. S., Kim, S. K., Colombier, E., Bud'ko, S. L. & Canfield, P. C. Solidification and loss of hydrostaticity in liquid media used for pressure measurements. *Rev. Sci. Instrum.* **86** (2015).
 8. Wang, X. et al. Acoustic and elastic properties of silicone oil under high pressure. *RSC Advances* **5**, 38056-38060 (2015).
 9. Hohensee, G. T., Wilson, R. B. & Cahill, D. G. Thermal conductance of metal–diamond interfaces at high pressure. *Nat. Commun.* **6** (2015).

Comments 4: *The authors recover a metastable material with different emission properties of the starting one and related to the changes induced by compression (non hydrostatic or anisotropic stress applied to the sample). Did the authors try a thermal annealing of the recovered material to understand how “stable” is the metastable form?*

Author reply: We are grateful for the reviewer’s advice, the contrast heating/annealing experiments were performed to investigate the stability of the pressure-treated structure. The released sample was heated from room temperature (RT) to 50 °C, 100 °C, 150 °C, 200 °C and 250 °C for 3 hours and annealed to RT (Figure R2). The integrated emission intensity and color remain almost unchanged with a thermal annealing temperature below 150 °C. After annealing from 200 °C, the integrated emission intensity of IPA is about 53% of the initial value. Both the emission intensity and color changed after annealing from 250 °C. This suggested that the targeted PL of the pressure-treated IPA was stable below 150 °C.

Figure R2. The PL spectra of IPA at different conditions. PL spectra of the initial IPA at RT (black line), the pressure-treated IPA at RT (red line), the pressure-treated IPA annealed at 50 °C, 100 °C, 150 °C, 200 °C and 250 °C (orange-blue lines).

Comments 5: The absorption spectra. It would be nice to have an explanation of how the authors got in scale absorption spectra. I presume by reducing the sample thickness, but in which way? Anyway, Figure R16 and R17 do not report the Intensity scale: they cannot use arbitrary units for an absorption spectrum, the scale is in absorbance units and the absorbance values are fundamental to understand the reliability of the data. In addition, the application of the Tauc method is highly disputable in a non semiconducting system. A molecular system absorption structure is made by bands and to evaluate the band gap you need to identify the maximum.

Author reply: Thanks for the reviewer's comments. In order to measure the absorption peak, we actually reduced the sample thickness before the *in situ* high-pressure ultraviolet-visible (UV-vis) absorption experiments. The relatively thick IPA was first pressed into the thin one using the diamond anvil cell (DAC). The PTM and ruby were then loaded into the DAC together with the thin sample for the following experiments. The appropriate thickness of the sample is beneficial to the collection of the absorption peak. The scale of absorption intensity was also provided in both Added

Supplementary Fig. 19 and Revised Fig. 3a.

According to the reviewer's suggestion, we use the absorption peak to evaluate the band gap. As shown in Revised Fig. 3a and Added Supplementary Fig. 19, the absorption peak at 1 atm was located at 295 nm, with an estimated bandgap of 4.20 eV. Below 1.9 GPa, the redshift of the absorption peak proved the decrease of the band gap. As pressure further increased, the cut-off of absorption intensity was inevitable, while the absorption edge kept continuously redshifted during the entire compression process. Especially, we reduced the thickness of the pressure-treated sample and re-measure the absorption spectrum for the identifiable absorption peak in Added Supplementary Fig. 19. The absorption peak of the pressure-treated IPA was located at 317 nm, indicating that the band gap of the targeted sample was 3.91 eV. In this regard, the band gap was successfully reduced by 0.29 eV after pressure treatment.

The corresponding revised details are highlighted in red and can be found in Lines 262-265, and 269-276 of the revised manuscript, as well as Lines 144-148 in the revised Supplementary Information.

Revised Fig. 3 | The evolution of the band gap of IPA upon compression and

decompression. a UV-vis absorption spectra of IPA upon compression. **b** The energies of singlet and triplet states of IPA before and after pressure treatment. **c** The calculated ΔE_{ST} ($\Delta E(S_1-T_6)$, $\Delta E(S_1-T_5)$, $\Delta E(S_1-T_4)$, $\Delta E(S_1-T_3)$, $\Delta E(S_1-T_2)$, $\Delta E(S_1-T_1)$) for the pristine and targeted IPA. **d** Proposed energy transfer processes for fluorescence and phosphorescence in IPA before and after pressure treatment.

Added Supplementary Fig. 19. UV-vis absorption spectra of IPA at ambient conditions and after pressure was completely released. The absorption peak at 1 atm was located at 295 nm (orange line), with an estimated bandgap of 4.20 eV. The absorption peak after completely releasing the pressure was located at 317 nm (blue line), with an estimated bandgap of 3.91 eV.

***Other comments:** As I said, the manuscript has been improved with respect to the original version but is not yet convincing. There is an extreme superficiality in treating the structural data especially overlooking the possible effects of non-hydrostatic or anisotropic compression. These issues do not alter the self-standing effect discovered, but make the meticulous description of the origin of this effect, attributed to the enforcement of the H-bonds, highly disputable.*

Author reply: As for the reviewer's concern, please allow us to answer the question again:

Firstly, the derived structural data in the present work belongs to a nice hydrostatic pressure range of silicone oil, which has been confirmed by the relevant hydrostatic experiments. The comparative experiments with different PTMs, including liquid argon and liquid nitrogen, further preclude the influence of hydrostatic issues on the present conclusions. The silicone oil we used could guarantee a nice hydrostatic pressure with a maximum pressure difference of 0.37 GPa below ~15.0 GPa. The derived structural parameters in this work were obtained within ~12.4 GPa, belonging to the hydrostatic pressure range of silicone oil. In addition, the PL evolution with liquid argon and nitrogen as PTMs was basically consistent with that of silicone oil. After ~20.0 GPa of depressurization, we still harvest a bright white-light emission with liquid argon and nitrogen, respectively. Therefore, the present used PTMs (silicone oil, liquid argon and liquid nitrogen) could not influence the current conclusions: the PL enhancement under high pressure and the harvest of white-light emission after pressure treatment. Notably, the same PTM used for both photoluminescence (PL) and ADXRD experiments is beneficial to assure the uniformity of the experimental environment.

Secondly, the obtained reduced hydrogen-bond distance derived from the corresponding nice ADXRD data is reliable, which has been further evidenced by the added 2D XRD patterns. The clear and identifiable 2D diffraction rings illustrate the good quality of the diffraction data, which further solidifies the conclusion on the H-bond strengthening. Notably, the initial crystal structure at ambient conditions in the present work was obtained from the Cambridge Crystallographic Data Center (CCDC) with CCDC number 1108747, the atomic positions of the pristine sample were known and definite. The pressure process merely changed the atomic positions slightly from their original positions. Thus, based on this standard structure and the well-performing XRD patterns, the crystal structure under different pressures was therefore refined by using GSAS software. We also performed optimization of atomic positions using VASP simulation code to further clarify the validity of the results. Coupled with the redshift

of $\nu(\text{C}=\text{O})$ in IR spectra, both experimental and calculated data confirmed the strengthening of hydrogen bonds.

In this regard, the enhanced hydrogen bonds were then acting on the electronic structures to boost the target emission of IPA. On the one hand, the reduced hydrogen-bond distance reinforced the electrostatic attraction on the lone pair (n) electrons in oxygen atom, thus stabilizing the n-orbitals to lower energies. Moreover, the strengthened hydrogen bonds further promoted the planarization of the structure, which favored electron delocalization accompanied by low-energy π - π^* transitions. Affected by the above changes of orbital energies, the original $^3(\text{n}, \pi^*)$ type of triplet states (T_5 and T_6) was converted to a mixture of $^3(\text{n}, \pi^*)/^3(\pi, \pi^*)$. Such mixed configuration could facilitate the intersystem crossing process from S_1 to T_6 or T_5 . A high fraction of triplet excitons was then generated for phosphorescence enhancement.

In summary, following the suggestions and comments of the reviewer, we carefully rechecked our manuscript and made some amendments to respond to the reviewer's issues. Hopefully, we have addressed all of your concerns. The corresponding revised details are highlighted in red and could be found in **Lines 22-26, 28-29, 55-59, 68-78, 96, 97, 99, 101, 102, 110-111, 115-131, 133-134, 141, 142, 145-152, 153-166, 169, 175, 184, 188, 200-202, 205, 213, 217, 228, 231, 243, 254, 262-265, 269-276, 280-282, 284, 288-289, 291, 302, 303, 309, 312, 323, 333-337, 341, 346-347, 369, 376-378, 381, 391, 392, 398, 399, 403, 405, 407, 409, 411, 413-417, 530-577** of the revised manuscript, as well as **Lines 21, 42-61, 64, 69, 77, 82, 86-90, 93, 98, 103, 114, 120, 122-123, 126, 130, 136, 139, 141-142, 144-148, 151, 164, 168** of the revised Supplementary Information.

Reply to Reviewer #2

The authors answered all my questions and I am satisfied with all the responses. The paper should be published as is.

Author reply: We are appreciated to the reviewer's positive evaluation and helpful work.

Reply to Reviewer #3

The authors have addressed all the comments and answered the technical questions. I think the paper has improved substantially after revision. I suggest it to be published.

Author reply: We thank the reviewer for all his/her input during the manuscript's revision process.

Reviewers' comments:

Reviewer #1 (Remarks to the Author):

In my previous two reports, I consistently acknowledged the undeniable effect observed following pressure treatment of IPA, deeming it a valuable outcome. However, my primary concerns revolved around the authors' extensive efforts to establish that this effect resulted from strengthening hydrogen bonds. The foundation for this assertion relied solely on data obtained from XRD experiments, as spectroscopic data provided only suggestive, yet inconclusive, evidence. Regrettably, it was only in the second revision that these crucial XRD data were presented, and, as I had suspected, they failed to support the analysis presented.

a) The 2D images presented in Figure 2a and S11 reveal that the investigated sample is a powder of notably poor quality, marked by significant texturing.

b) The number reflections is pretty low, and conducting a Rietveld refinement necessitates substantial manipulation (constrains).

c) The considerable broadness of the peaks (crystallites of nanometric dimensions) and the relatively high background intensity lend to a favorable Rwp. Nevertheless, this is not a proof of the accuracy of the fit, as illustrated by B. H. Toby in Powder Diffraction (2006, 21, 67). As a matter of fact important residuals are observed in all the patterns presented in S11.

d) While the data can yield reliable lattice parameters, they fail to provide the essential atomic positions required for precise determination of hydrogen bond distances. It is disconcerting that on page 10, these distances are reported with three decimal places, devoid of any indication of uncertainty or error, which is not acceptable. Moreover, the lattice parameters reported in Figure 2 are rather scattered adding further doubts about the reliability of the internal parameters derived from this analysis. Notably, there is no mention of the curve used to fit these data. The conspicuous absence of error bars throughout the manuscript is particularly vexing, especially given the minor nature of the proposed effects. For instance, the dihedral angles shift by less than 0.5 degrees.

e) I mentioned in a previous report that another paper (now included as reference 34) presented a thorough investigation of IPA afterglow phosphorescence. The mechanism behind this phosphorescence was assigned, thanks to magnetic measurements, to the spin exchange between a radical ion pair favored by the packing of IPA molecules and promoting the formation of charge transfer complexes, or their derived radical pairs. In this way the two electron spins over the pair are only weakly coupled allowing the spin exchange and then the intersystem crossing which is at the basis of the phosphorescence. These effects are expected to be amplified with increasing pressure. I am perplexed by the fact that this paper's results are not discussed in relation to the proposed interpretation, which I believe should be an essential consideration when an alternative mechanism is put forth.

In my opinion this manuscript is not worth to be published.

Reviewer #4 (Remarks to the Author):

Reviewer #1 has correctly signaled the issues with the low quality XRD data presented in the paper. These XRD data are of poor quality (relatively small number of weak broad peaks, which do not provide information on atomic positions, especially hydrogen). The results of the Rietveld refinements consequently cannot be reliable. In addition, the crystallographic data are not even provided (fractional atomic coordinates and atomic displacement parameters with their ESD's) nor deposited as CIFs. The data shown can only be used to provide lattice parameters with the additional complication that the pressure medium used is non-hydrostatic. The structural results presented in the paper are thus not sound as it is not possible to obtain accurate bond distances and angles from these XRD data.

For your guidance, itemized response to reviewer' s comments is appended below.

Reply to Reviewer #1

Comments: In my previous two reports, I consistently acknowledged the undeniable effect observed following pressure treatment of IPA, deeming it a valuable outcome. However, my primary concerns revolved around the authors' extensive efforts to establish that this effect resulted from strengthening hydrogen bonds. The foundation for this assertion relied solely on data obtained from XRD experiments, as spectroscopic data provided only suggestive, yet inconclusive, evidence. Regrettably, it was only in the second revision that these crucial XRD data were presented, and, as I had suspected, they failed to support the analysis presented.

Author reply: Many thanks to you for your tireless work and guidance. We appreciate for your high evaluation with our work. We understand your concerns, and supplement more experimental evidence to solidify our conclusion on the strengthening of hydrogen bonds. To systematically investigate the mechanism of the enhanced phosphorescence emission following pressure treatment of IPA, we have supplemented more experimental and theoretical evidence, including *in situ* **time-of-flight neutron diffraction** (conducted at BL11 PLANET of the Japan Proton Accelerator Research Complex), **magnetic-field photoluminescence (MPL) experiments** (performed at Wuhan National High Magnetic Field Center), and **TD-DFT calculations**. Given that organic IPA consists solely of light elements such as C, H, and O, we further performed *in situ* TOF neutron diffraction to delve more deeply into the structural evolution upon compression and decompression. And the results are consistent with our synchrotron X-ray diffraction (XRD) data, which are included in the supplementary information for each revision. Based on these optical experiments, structural analysis, and theoretical calculations, we comprehensively analyzed the impact of pressure treatment on intersystem crossing (ISC) from three aspects: **spin-orbit coupling (SOC), hyperfine coupling (HFC), and singlet-triplet energy gap (ΔE_{ST})**.

(1) Intermolecular interaction changes:

Based on the *in situ* TOF neutron diffractions and XRD data, we have confirmed the conclusion of hydrogen bond enhancement after pressure treatment. After pressure was completely released, the $D_{17\cdots O_9}$ (d_1) decreased from 1.59(4) Å to 1.47(4) Å and $D_{21\cdots O_5}$ (d_2) decreased from 1.77(4) Å to 1.56(3) Å. In addition, the pressure-treatment engineering rendered a lessened spatial overlap of the π -stacked motif in targeted IPA, where the parallel misalignment angle (σ) decreased from 67.3(8)° to 64.9(8)°.

(2) Mechanism of the phosphorescence enhancement after pressure treatment:

i. SOC: On one hand, the decreased hydrogen-bond distances would drive strengthened electrostatic interactions between the proton and the lone pair electrons. The n -orbitals would like to be stabilized to the lower energies by the enhanced electrostatic environments, leading to the consequent changes of SOC as well as ISC process¹. After pressure treatment, the SOC coefficients $\zeta(S_1-T_2)$, $\zeta(S_1-T_3)$, $\zeta(S_1-T_5)$, and $\zeta(S_1-T_6)$ have been improved to varying degrees. Particularly, we found that the T_5 and T_6 states underwent a noticeable change from $^3(n, \pi^*)$ in the original IPA to the mixture of $^3(n, \pi^*)$ and $^3(\pi, \pi^*)$ in pressure-treated IPA (Revised Fig. 3b). According to the El-Sayed's rule², **the effective mixture of $n-\pi^*/\pi-\pi^*$ transition configurations tremendously promotes the increase of SOC for accelerated spin-flipping process.**

ii. HFC: Secondly, we have observed the magnetic-field effect in the pristine IPA based on MPL tests, consistent with the previous report (Revised Fig. 4a). This undoubtedly confirms the influence of HFC on the initial phosphorescence emission before pressure treatment^{3,4}. However, the pressure-treated IPA did not show significant magnetic-field effects in the MPL spectra, hinting at a negligible influence of HFC on the enhanced phosphorescence emission of the targeted IPA. According to the TD-DFT calculations, we found that **the pressure-treated IPA mainly shows an intramolecular $n-\pi^*$ transition, which is not conducive to the existence of the weakly coupled radical pairs** (Supplementary Fig. 19). Therefore, **the HFC effect has**

negligible influence on the pressure-treated IPA due to the parallel-displaced configuration.

iii. ΔE_{ST} : Besides HFC effect, the pressure-treated parallel-displaced configuration also played a significant role in **minimizing energy difference $\Delta E(\pi, \pi^*)$** of the bonding and antibonding π -type molecular orbitals⁵. The enhanced hydrogen bonds would also **stabilize the n-orbitals to the lower energies** by the enhanced electrostatic environments. Therefore, the energies for both singlet and triplet excited states as well as ΔE_{ST} were recalculated in detail (Revised fig. 4b, c). Compared with the initial states, the treated sample featured various degrees of lowered energy levels. **The $\Delta E(S_1-T_6)$, $\Delta E(S_1-T_5)$, $\Delta E(S_1-T_4)$, and $\Delta E(S_1-T_3)$ were also decreased after pressure treatment, further contributing to accelerating the ISC process.**

In this regard, **attributed to the enhanced hydrogen bonds and the parallel-displaced arrangement, the pressure treatment engineering primarily accelerates the ISC process by enhancing SOC and narrowing ΔE_{ST} of the targeted IPA, thus harvesting the improved phosphorescence emission.**

Revised Fig. 4 | The effects of HFC and ΔE_{ST} on ISC process of IPA before and after pressure treatment. **a** Magnetic-field effects on the PL intensity of IPA. The data at 1 atm and released represent the mean value from five and three experiments, respectively. **b** The energies of singlet and triplet states of IPA before and after pressure treatment. **c** The calculated ΔE_{ST} ($\Delta E(S_1-T_6)$, $\Delta E(S_1-T_5)$, $\Delta E(S_1-T_4)$, $\Delta E(S_1-T_3)$, $\Delta E(S_1-T_2)$, $\Delta E(S_1-T_1)$) for the pristine and targeted IPA. **d** Proposed energy transfer processes for fluorescence and phosphorescence in IPA before and after pressure treatment.

Supplementary Fig. 19. The molecular orbitals for the π - π staking dimers in the pristine and pressure-treated IPA.

References:

1. Ma, H. et al. Electrostatic interaction-induced room-temperature phosphorescence in pure organic molecules from QM/MM calculations. *J. Phys. Chem. Lett.* **7**, 2893-2898 (2016).
2. Schmidt, K. et al. Intersystem crossing processes in nonplanar aromatic heterocyclic molecules. *J. Phys. Chem. A* **111**, 10490-10499 (2007).
3. Hu, B., Yan, L. & Shao, M. Magnetic-Field Effects in Organic Semiconducting Materials and Devices. *Advanced Materials* **21**, 1500-1516 (2009).
4. Kuno, S., Akeno, H., Ohtani, H. & Yuasa, H. Visible room-temperature phosphorescence of pure organic crystals via a radical-ion-pair mechanism. *Phys.*

Chem. Chem. Phys. **17**, 15989 (2015).

5. Lutz, P. B. & Bayse, C. A. Orbital-based insights into parallel-displaced and twisted conformations in π - π interactions. *Physical Chemistry Chemical Physics* **15** (2013).

Comments 1: a) *The 2D images presented in Figure 2a and S11 reveal that the investigated sample is a powder of notably poor quality, marked by significant texturing.*

b) *The number reflections is pretty low, and conducting a Rietveld refinement necessitates substantial manipulation (constrains).*

c) *The considerable broadness of the peaks (crystallites of nanometric dimensions) and the relatively high background intensity lead to a favorable Rwp. Nevertheless, this is not a proof of the accuracy of the fit, as illustrated by B. H. Toby in Powder Diffraction (2006, 21, 67). As a matter of fact important residuals are observed in all the patterns presented in S11.*

d) *While the data can yield reliable lattice parameters, they fail to provide the essential atomic positions required for precise determination of hydrogen bond distances. It is disconcerting that on page 10, these distances are reported with three decimal places, devoid of any indication of uncertainty or error, which is not acceptable. Moreover, the lattice parameters reported in Figure 2 are rather scattered adding further doubts about the reliability of the internal parameters derived from this analysis. Notably, there is no mention of the curve used to fit these data. The conspicuous absence of error bars throughout the manuscript is particularly vexing, especially given the minor nature of the proposed effects. For instance, the dihedral angles shift by less than 0.5 degrees.*

Author reply: We appreciate your suggestions. The problems you proposed are common phenomena, especially for high-pressure experiments on light-weight elements, so we resort to **neutron diffraction**. Neutron diffraction stands out as a powerful technique for materials analysis due to the pronounced sensitivity to light elements, such as hydrogen, which enables detailed insights into their spatial arrangements within crystal structures¹. Because the large incoherent scattering cross section of hydrogen causes a large contribution to the background in the neutron

diffraction data, the deuterated IPA (IPA- d_6) was used in the *in situ* TOF neutron diffraction experiment².

As pressure was increased from ambient conditions to 19.6 GPa, all neutron diffraction patterns moved toward smaller d -spacing, indicating lattice compression (Supplementary Fig. 13). After pressure was completely released, several peaks were located at smaller d -values, where the d -spacings of (033), (110) and (022) planes narrowed ~ 0.1 Å (Figure R1). Notably, the orientations of (033) and (022) lattice planes are almost perpendicular to the direction of hydrogen bonds, and the evolution of their d -spacings is closely related to the change of hydrogen-bond distance (Supplementary Fig. 16). The decreased d -spacing of (110) should be attributed to both closed π stacking as well as reduced hydrogen bonds.

Based on the *in situ* TOF neutron diffraction experiments, we further performed Rietveld refinements of neutron diffraction patterns to obtain detailed lattice parameters and other structural information upon compression and decompression (Supplementary Fig. 15 and Figure R2). The lattice parameters evolution based on the neutron diffraction is consistent with the refined results of ADXRD data, further confirming the lattice shrinkage under high pressure (Figure R2a, b, d, and e). We improved the methods of the Rietveld refinements based on this neutron diffraction experiment, and the results showed that the variation of intermolecular/intramolecular dihedral angles is indeed within the error range. Thus, we revisited the structural evolution under high pressure and the relationship between structural changes and photoluminescence. Considering the error of dihedral angle, we added restrictions of bond length and bond angle of IPA when refining the atom positions, with the aim of keeping their variation within a reasonable range and avoiding molecular disintegration.

(1) Hydrogen bonds evolution: Considering the error of refinement, we assessed the arrangement changes of the correlative intermolecular distances $D_{17\cdots O_9}$ (d_1), $D_{21\cdots O_5}$ (d_2), and the parallel misalignment angle (σ) of IPA upon compression and decompression (Figure R2c, f). We found that both d_1 and d_2 displayed a decreasing trend with the increase of pressure, suggesting the strengthened hydrogen bonds at high pressure. After pressure was completely released, d_1 decreased from 1.59(4) Å to 1.47(4)

\AA and d_2 decreased from 1.77(4) \AA to 1.56(3) \AA , consistent with the observed decrease (~ 0.1 \AA) in the d -spacings of (033) and (022) planes. Therefore, **the hydrogen bonds in pressure-treated IPA were stronger than those ones before pressure treatment.**

(2) **π - π staking arrangement evolution:** Moreover, the σ also tended to decrease upon compression, and the lesser σ value of $67.3(8)^\circ$ rendered a lessened spatial overlap of the π -stacked motif in pressure-treated IPA. Meanwhile, the intermolecular perpendicular distance between benzene rings was compressed from ~ 3.5 \AA to ~ 3.3 \AA . The intermolecular distance $d_{\text{C}\cdots\text{C}}$ between benzene rings did not change significantly, which varied from 3.77(6) \AA to 3.64(4) \AA (Supplementary Fig. 17).

Figure R1. Time-of-flight (TOF) neutron diffraction patterns of IPA- d_6 .

Supplementary Fig. 15. Rietveld refinement plot of IPA- d_6 before and after pressure treatment.

Supplementary Fig. 16. **a** The lattice planes of (033) and (022) in IPA. **b** The lattice planes of (110) in IPA.

Figure R2. Crystal structure evolution upon compression and decompression. a The compression rate of three lattice constants (a , b , c) at different pressures. **b** Pressure-dependent β evolution of IPA. **c** Pressure-dependent hydrogen-bond distances $D_{17}\cdots O_9$ (d_1) and $D_{21}\cdots O_5$ (d_2) evolution of IPA. **d** The compression rate of three lattice constants (a , b , c) at different pressures. **e** Pressure-dependent β evolution of IPA. **f** Pressure-dependent σ evolution of IPA. The parameters in **a** and **b** were determined by Rietveld refinement of ADXRD patterns. The parameters in **c-f** were determined by Rietveld refinement of neutron diffraction patterns.

Supplementary Fig. 17. The crystal structure of IPA. The black arrows indicate the C···C distance.

The revised details are highlighted in red and can be found in Lines 55-56, 72-76, 99-101, 116-117, 181-222, 329 of the revised manuscript, as well as Lines 91-135, 141-143, 151-155 in the revised Supplementary Information.

References:

1. Haberl, B., Guthrie, M. & Boehler, R. Advancing neutron diffraction for accurate structural measurement of light elements at megabar pressures. *Scientific Reports* **13** (2023).
2. Ting, V. P.; Henry, P. F.; Schmidtman, M.; Wilson, C. C.; Weller, M. T. Probing Hydrogen Positions in Hydrous Compounds: Information from Parametric Neutron Powder Diffraction Studies. *Phys. Chem. Chem. Phys.* **2012**, *14*, 6914-6921.

Comments 2: I mentioned in a previous report that another paper (now included as reference 34) presented a thorough investigation of IPA afterglow phosphorescence. The mechanism behind this phosphorescence was assigned, thanks to magnetic measurements, to the spin exchange between a radical ion pair favored by the packing

of IPA molecules and promoting the formation of charge transfer complexes, or their derived radical pairs. In this way the two electron spins over the pair are only weakly coupled allowing the spin exchange and then the intersystem crossing which is at the basis of the phosphorescence. These effects are expected to be amplified with increasing pressure. I am perplexed by the fact that this paper's results are not discussed in relation to the proposed interpretation, which I believe should be an essential consideration when an alternative mechanism is put forth.

Author reply: We appreciate the reviewer for the comments. According to the reviewer's suggestions, we provided the magnetic-field photoluminescence (MPL) experiments of IPA and analyzed the corresponding **influence of hyperfine coupling (HFC)** in the revised manuscript. The combination of these data proposes that the mechanism of the phosphorescence emission enhancement of the pressure-treated IPA should be mainly derived from the strengthened hydrogen bonds, and the detailed analysis as follows:

(1) **The HFC effect indeed plays a key role at ambient conditions as the previous paper reported and the reviewer proposed, but this effect was negligible after pressure treatment.** Before pressure treatment, the MPL intensity of the pristine sample changed with the magnetic flux density modulation of up to ~ 2 T (Figure R3). This magnetic-field effect proved the presence of singlet and triplet radical ion pairs, which undoubtedly confirmed the existence of HFC mechanism on the initial phosphorescence emission before pressure treatment^{1,2}.

However, after pressure treatment, we noticed minimal variation in pressure-treated IPA (Figure R3). Notably, an external magnetic field has little influence on the singlet and triplet exciton ratios through ISC in excitonic states when polaron-pair states are absent under photoexcitation². Therefore, **the contribution of the radical-pairs-derived supplementary HFC channel to ISC processes of the pressure-treated IPA will also be reduced.**

According to the structural analysis, the pressure treatment engineering rendered a lessened spatial overlap of the π -stacked motif in the targeted IPA, accompanied by a lesser σ value of $64.9(8)^\circ$. In order to explore the relationship between the parallel-

displaced arrangement and the radical pairs, we conducted the molecular orbitals of IPA dimer before and after pressure treatment. We found that the pristine IPA mainly exhibited charge-transfer-like π - π^* transition between two IPA molecules, thus benefiting the presence of radical pairs and contributing to the ISC process. However, **the pressure-treated IPA mainly shows an intramolecular n- π^* transition, which is not conducive to the existence of the weakly coupled radical pairs and the supplemental HFC channel (Supplementary Fig. 19). Therefore, the effect of HFC was negligible after pressure treatment owing to the misaligned π - π stacking arrangement.**

(2) All the present experimental and theoretical data have indicated that the strengthened electrostatic interaction played a significant role in phosphorescence emission for IPA. As mentioned above, after pressure treatment, the decreased hydrogen-bond distances have driven strengthened electrostatic interactions between the proton and the lone pair electrons³, leading to the consequent **enhancement in SOC and decrease in ΔE_{ST}** (Figure R4). In addition, the parallel-displaced configuration would also contribute to minimizing energy difference $\Delta E(\pi, \pi^*)$ of the bonding and antibonding π -type molecular orbitals⁴, thereby further contributing to the reduction of the ΔE_{ST} (Figure R4b). Therefore, **the mechanism of the phosphorescence emission enhancement of the pressure-treated IPA should be mainly derived from the strengthened hydrogen bonds.**

Figure R3. Magnetic-field effects on the PL intensity of IPA. The data at 1 atm and released represent the mean value from five and three experiments, respectively.

Supplementary Fig. 19. The molecular orbitals for the π - π staking dimers in the pristine and pressure-treated IPA.

Figure R4. a The calculated ξ ($\xi(S_1-T_6)$, $\xi(S_1-T_5)$, $\xi(S_1-T_4)$, $\xi(S_1-T_3)$, $\xi(S_1-T_2)$, $\xi(S_1-T_1)$) for the pristine and pressure-treated IPA. **b** The calculated ΔE_{ST} ($\Delta E(S_1-T_6)$, $\Delta E(S_1-T_5)$, $\Delta E(S_1-T_4)$, $\Delta E(S_1-T_3)$, $\Delta E(S_1-T_2)$, $\Delta E(S_1-T_1)$) for the pristine and targeted IPA.

The revised details are highlighted in red and can be found in Lines 55-56, 72-76, 99-101, 116-117, 223, 229-239, 261-264, 268-292, 302-307 of the revised manuscript, as well as Lines 84-89, 160-164 in the revised Supplementary Information.

References:

1. Kuno, S., Akeno, H., Ohtani, H. & Yuasa, H. Visible room-temperature phosphorescence of pure organic crystals via a radical-ion-pair mechanism. *Phys. Chem. Chem. Phys.* **17**, 15989 (2015).
2. Hu, B., Yan, L. & Shao, M. Magnetic-Field Effects in Organic Semiconducting Materials and Devices. *Advanced Materials* **21**, 1500-1516 (2009).
3. Ma, H. et al. Electrostatic interaction-induced room-temperature phosphorescence in pure organic molecules from QM/MM calculations. *J. Phys. Chem. Lett.* **7**, 2893-2898 (2016).
4. Lutz, P. B. & Bayse, C. A. Orbital-based insights into parallel-displaced and twisted conformations in π - π interactions. *Phys. Chem. Chem. Phys.* **15** (2013).
5. Schmidt, K. et al. Intersystem crossing processes in nonplanar aromatic heterocyclic molecules. *J. Phys. Chem. A* **111**, 10490-10499 (2007).

Reply to Reviewer #4

Reviewer #1 has correctly signaled the issues with the low quality XRD data presented in the paper. These XRD data are of poor quality (relatively small number of weak broad peaks, which do not provide information on atomic positions, especially hydrogen). The results of the Rietveld refinements consequently cannot be reliable. In addition, the crystallographic data are not even provided (fractional atomic coordinates and atomic displacement parameters with their ESD's) nor deposited as CIFs. The data shown can only be used to provide lattice parameters with the additional complication that the pressure medium used is non-hydrostatic. The structural results presented in the paper are thus not sound as it is not possible to obtain accurate bond distances and angles from these XRD data.

Author reply: Thanks for the reviewer's comments. Given that organic IPA consists solely of light elements such as C, H, and O, which displayed a weaker scatter ability in ADXRD experiments. We further performed *in situ* TOF neutron diffraction to delve more deeply into the structural evolution upon compression and decompression. Neutron diffraction stands out as a powerful technique for materials analysis due to the pronounced sensitivity to light elements, such as hydrogen, which enables detailed insights into their spatial arrangements within crystal structures¹. Because the large incoherent scattering cross section of hydrogen causes a large contribution to the background in the neutron diffraction data, the deuterated IPA (IPA-*d*₆) was used in the *in situ* neutron diffraction experiment². **According to the reviewer's suggestions, the crystallographic data are provided below based on Rietveld refinements of neutron diffraction patterns (Table R1 and R2), and these data are also included in the revised manuscript.**

As pressure was increased from ambient conditions to 19.6 GPa, all neutron diffraction patterns moved toward smaller *d*-spacing, indicating lattice compression (Supplementary Fig. 13). After pressure was completely released, several peaks were located at smaller *d*-values, where the *d*-spacings of (033), (110) and (022) planes narrowed ~0.1 Å (Figure R1). Notably, the orientations of (033) and (022) lattice planes are almost perpendicular to the direction of hydrogen bonds, and the evolution of their

d -spacings is closely related to the change of hydrogen-bond distance (Supplementary Fig. 16). The decreased d -spacing of (110) should be attributed to both closed π stacking as well as reduced hydrogen bonds.

Based on the *in situ* TOF neutron diffraction experiments, we further performed Rietveld refinements of neutron diffraction patterns to obtain detailed lattice parameters and other structural information upon compression and decompression (Supplementary Fig. 15 and Figure R2). The lattice parameters evolution based on the neutron diffraction is consistent with the refined results of ADXRD data, further confirming the lattice shrinkage under high pressure (Figure R2a, b, d, and e). We improved the methods of the Rietveld refinements based on this neutron diffraction experiment, and the results showed that the variation of intermolecular/intramolecular dihedral angles is indeed within the error range. Thus, we revisited the structural evolution under high pressure and the relationship between structural changes and photoluminescence. Considering the error of dihedral angle, we added restrictions of bond length and bond angle of IPA when refining the atom positions, with the aim of keeping their variation within a reasonable range and avoiding molecular disintegration.

(1) The evolution of the hydrogen bonds: Considering the error of refinement, we assessed the arrangement changes of the correlative intermolecular distances $D_{17\cdots O_9}$ (d_1), $D_{21\cdots O_5}$ (d_2), and the parallel misalignment angle (σ) of IPA upon compression and decompression (Figure R2c, f). We found that both d_1 and d_2 displayed a decreasing trend with the increase of pressure, suggesting the strengthened hydrogen bonds at high pressure. After pressure was completely released, the d_1 decreased from 1.59(4) Å to 1.47(4) Å and d_2 decreased from 1.77(4) Å to 1.56(3) Å. Therefore, **the hydrogen bonds in pressure-treated IPA were stronger than those ones before pressure treatment.**

(2) The evolution of the π - π stacking arrangement: Moreover, the σ also tended to decrease upon compression, and the lesser σ value of 67.3(8)° rendered a lessened spatial overlap of the π -stacked motif in pressure-treated IPA. Meanwhile, the intermolecular perpendicular distance between benzene rings was compressed from ~3.5 Å to ~3.3 Å. The intermolecular distance $d_{(C\cdots C)}$ between benzene rings did not

change significantly, which varied from 3.77(6) Å to 3.64(4) Å (Supplementary Fig. 17).

Table R1. The crystallographic data of IPA at 1 atm.

Atomic label	Xfrac+ESD	Xfrac+ESD	Xfrac+ESD
C1	0.759(6)	0.3471(10)	0.3724(19)
C5	0.692(8)	0.2704(10)	0.3247(17)
C9	0.757(10)	0.1988(11)	0.3904(23)
C13	0.888(10)	0.2054(12)	0.5076(23)
C17	-0.012(12)	0.2825(11)	0.5524(20)
C21	0.909(8)	0.3537(12)	0.4867(21)
C25	0.659(7)	0.4220(11)	0.3046(23)
C29	0.659(7)	0.1167(13)	0.3401(21)
D1	0.556(7)	0.2672(13)	0.2391(21)
D5	-0.018(8)	0.1500(15)	0.5527(24)
D9	0.139(7)	0.2870(17)	0.6367(20)
D13	0.003(14)	0.4137(15)	0.5175(29)
D17	0.660(7)	0.5398(14)	0.3172(26)
D21	0.631(8)	0.0065(14)	0.3658(22)
O1	0.743(8)	0.4952(18)	0.3631(24)
O5	0.502(10)	0.4189(17)	0.2113(26)
O9	0.498(10)	0.1169(17)	0.2481(28)
O13	0.723(8)	0.0528(16)	0.4036(24)

Table R2. The crystallographic data of IPA after releasing pressure completely.

Atomic label	Xfrac+ESD	Xfrac+ESD	Xfrac+ESD
C1	0.723(7)	0.3396(14)	0.8734(19)
C5	0.628(9)	0.2661(14)	0.8290(19)
C9	0.720(7)	0.1936(14)	0.8877(24)
C13	0.874(7)	0.1984(11)	0.9985(20)
C17	0.948(7)	0.2757(14)	1.0490(19)
C21	0.893(8)	0.3442(15)	0.9864(22)
C25	0.652(6)	0.4146(13)	0.8077(21)
C29	0.604(6)	0.1145(13)	0.8381(18)
D1	0.541(7)	0.2633(16)	0.7401(21)
D5	0.931(9)	0.1421(13)	1.0452(23)
D9	0.123(7)	0.2796(16)	1.1236(24)
D13	0.960(7)	0.4037(17)	1.0212(22)
D17	0.671(7)	0.5346(16)	0.8153(23)
D21	0.623(7)	-0.0043(15)	0.8479(18)
O1	0.756(6)	0.4836(19)	0.8608(21)
O5	0.497(8)	0.4161(14)	0.7202(22)
O9	0.445(10)	0.1134(17)	0.7384(26)
O13	0.705(7)	0.0476(15)	0.8917(18)

Figure R1. Time-of-flight (TOF) neutron diffraction patterns of IPA- d_6 .

Supplementary Fig. 15. Rietveld refinement plot of IPA- d_6 before and after pressure treatment.

Supplementary Fig. 16. **a** The lattice planes of (033) and (022) in IPA. **b** The lattice planes of (110) in IPA.

Figure R2. Crystal structure evolution upon compression and decompression. **a** The compression rate of three lattice constants (a , b , c) at different pressures. **b** Pressure-dependent β evolution of IPA. **c** Pressure-dependent hydrogen-bond distances $D_{17}\cdots O_9$ (d_1) and $D_{21}\cdots O_5$ (d_2) evolution of IPA. **d** The compression rate of three lattice constants (a , b , c) at different pressures. **e** Pressure-dependent β evolution of IPA. **f** Pressure-dependent σ evolution of IPA. The parameters in **a** and **b** were determined by Rietveld refinement of ADXRD patterns. The parameters in **c-f** were determined by

Rietveld refinement of neutron diffraction patterns.

Supplementary Fig. 17. The crystal structure of IPA. The black arrows indicate the C···C distance.

The revised details are highlighted in red and can be found in Lines 55-56, 72-76, 99-101, 116-117, 181-222, 329 of the revised manuscript, as well as Lines 91-135, 141-143, 151-155 in the revised Supplementary Information.

References:

1. Haberl, B., Guthrie, M. & Boehler, R. Advancing neutron diffraction for accurate structural measurement of light elements at megabar pressures. *Scientific Reports* **13** (2023).
2. Ting, V. P.; Henry, P. F.; Schmidtman, M.; Wilson, C. C.; Weller, M. T. Probing Hydrogen Positions in Hydrous Compounds: Information from Parametric Neutron Powder Diffraction Studies. *Phys. Chem. Chem. Phys.* **2012**, *14*, 6914-6921.

In summary, following the suggestions and comments of the reviewer, we carefully rechecked our manuscript and made some amendments to respond to the reviewer's

issues. Hopefully, we have addressed all of your concerns. The corresponding revised details are highlighted in red and could be found in Lines 5-6, 11-16, 55-56, 58, 59, 61, 63, 72-76, 85, 99-101, 116-117, 155-156, 169-171, 176, 180, 181-223, 225, 229-239, 242, 244, 247, 253, 259, 261-264, 268-292, 293, 295-298, 302-307, 329, 341, 346, 369, 375-407, 410, 414, 471-518, 522-567, 572-579, 581-582 of the revised manuscript, as well as Lines 7-8, 13-18, 84-135, 138, 141-143, 149, 151-157, 160-164, 170-177 of the revised Supplementary Information.

REVIEWER COMMENTS

Reviewer #5 (Remarks to the Author):

The authors present a new series of materials which show increased white light emission after pressure treatment.

The manuscript has been reviewed previously and in the most recent submission neutron time-of-flight data has been presented as part of the study. In particular this data has been used to support the result that upon recovery the hydrogen bonds are strengthened as highlighted by their reduction in length compared to the as synthesised material (prior to pressure treatment).

The quality of the neutron diffraction data and its interpretation as presented should be questioned to ensure that the hydrogen bond length determined is correctly determined.

Firstly, there is not enough technical information in the SI to determine how the experiment was performed.

a) How was the pressure determined - the authors state using the known pressure load curve for the anvils and setup - it is well known that the pressure load performance will depend on packing, the hardness of the material and the PTM (as well as variability in the gasket etc). The experiment should have been performed with a pressure marker in the sample volume as is the normal for such experiments.

b) The lack of pressure marker then leads to a further question - how do we know when you recover the gasket back to ambient conditions and determine the structure it is actually at ambient pressure. Unless the gasket is removed from the cell assembly (a detail missing if it is) a significant amount of residual pressure may be held by the press even with no applied load. Residual pressure will shorten the hydrogen bond length itself and there may well be significant hysteresis in the H-bond behaviour.

c) Pressure transmitting media - no details as to the pressure transmitting media that was used in the neutron experiment is given. The peaks in the diffraction pattern appear to broaden and hence suggest a non hydrostatic environment. Whilst this may not affect the change in the white light emission properties it will dramatically alter the quality of the diffraction data and hence the determined structural parameters and reliability of the fit etc.

d) It appears that only the longer time window of the PLANET instrument has been presented - what about the data in the lower t-o-f window? In this d-spacing range is where a significant amount of the detailed structural information is to be found when performing Rietveld analysis on a powder diffraction data set.

Regarding the presentation of the data

a) I can understand that only the two ambient pressure data sets are shown in the main manuscript but more should be shown in the SI - show all of the high pressure data sets (likewise provide all of the cif files for the high pressure data sets)

b) Where are the fits to the in situ pressure data sets - these need to be shown to allow the reader to evaluate the work experimentally.

c) There are typos in Supp Table 2 - see the c vs β angle for the Released data set....

d) I assume that Supp Table 3 was determined by neutron diffraction data. The determined values are very low in precision - how were the values determined? Directly from the refined structure or from lattice plane spacing?

e) What are the figures of merit for the fits?

Overall, I think that whilst the experiment of the high pressure neutron may have been performed

well - the current presentation of results do not do it justice and by doing so actually leave the reader to ask questions about their validity and analysis. More information and robustness is required to justify publication in its current form.

Reviewer #6 (Remarks to the Author):

The Editor asked me to evaluate quality and validity of quantum-chemical simulations of characterizing energy and properties of singlet and triplet states supporting experimental data. Upon inspection of the revised MS, SI and the response letter, I find the following:

1. An addition of quantum-chemical simulations analyzing the effects of SOC (spin-orbit coupling), HFC (hyperfine coupling) and ΔE_{ST} (singlet-triplet gap) has significantly substantiated and supported the original hypothesis that hydrogen bond enhancement is the main cause of phosphorescence enhancement. These calculations nicely demonstrate an increased effective mixture of $n\text{-}\pi^*$ and $\pi\text{-}\pi^*$ character that likely leads to more effective intersystem crossing owing to El-Sayed rule. Increased effective amount of SOC favors this as well. Calculations also do not find any substantial influence of HFC in the pressure-treated samples. Finally, calculations point out to the uniform decrease of first singlet (S_1) to several triplet states gaps pointing to stabilization of the resulting triplet states. Overall, presenting quantum chemical modeling added value to science and improved the article by invoking basic quantum-mechanical analysis and establishing connections to known physical modes.

2. The authors have used Gaussian 09 for DFT and TDDFT calculations of a model system (dimer). B3LYP/6-31G(d, p) model chemistry was invoked, which I find appropriate for these simulations (indeed, NTOs do not show traces of charge-transfer character). Calculations were followed by NTO analysis which is an appropriate methodology to examine the orbital character for TDDFT derived electronic states. Finally, the SOC coefficients were evaluated with Beijing density function(BDF) program. I am not familiar and have not used this software, however, by examining their webpage I found BDF to be adequate for this purpose.

3. While I am not questioning the results of quantum-mechanical modeling, I would make several suggestions to improve computational work mostly answering question on how stable are the identified trends and what is an uncertainty of the derived numbers with respect to the chosen DFT framework. These additional results may be placed in SI with some brief referencing from the main text.

a) The authors are using a model dimer system, neglecting other molecules and generally dielectric surrounding. I would suggest to check their results against a single point calculation of 4 molecules ($\times 2$) size and against a single point calculation of a dimer within polarizable continuum model (PCM). These things are standard within Gaussian software and will not take a lot of computational power.

b) A more subtle point is a value of singlet-triplet gap: ΔE_{ST} . I would remind that the value of this gap in the TDDFT framework depends a lot on the fraction of the orbital exchange present in the DFT model used. For example, B3LYP uses 20% of this exchange. I would expect that models with a larger fraction of exchange (like asymptotically corrected functionals) would lead to much larger ΔE_{ST} , whereas models with lower fractions (like GGA functionals) will lead to very small gaps. While B3LYP could be a reasonable compromise, I would suggest to make additional estimates of ΔE_{ST} using i) several other functionals (e.g. HSE, PBE0, WB97X) to estimate uncertainty of numbers and ii) using Delta SCF approach (different from TDDFT) to evaluate ΔE_{ST} ($S_1\text{-}T_1$). Delta SCF should be a lot more stable with respect to the choice of DFT model compared to TDDFT method (but it will not suit for estimation of ΔE_{ST} ($S_1\text{-}T_5/6$) gaps). Again these are trivial single point calculations that can be performed within a few days.

Altogether I believe that these minor additional simulations will make the currently present modeling more solid and trustable.

For your guidance, itemized response to reviewer' s comments is appended below.

Reply to Reviewer #5

Comments 1: The authors present a new series of materials which show increased white light emission after pressure treatment.

The manuscript has been reviewed previously and in the most recent submission neutron time-of-flight data has been presented as part fo the study. In particular this data has been used to support the result that upon recovery the hydrogen bonds are strengthened as highlighted by their reduction in length compared to the as synthesised material (prior to pressure treatment).

The quality of the neutron diffraction data and its interpretation as presented should be questioned to ensure that the hydrogen bond length determined is correctly determined.

Firstly, there is not enough technical information in the SI to determine how the experiment was performed.

Author reply: Thanks for the reviewer's kind comments with the positive affirmation. First of all, we acknowledge your comments and suggestions very much, which are valuable in improving the quality of our manuscript. According to your suggestions, we added the new data and related technical information in the Supplementary Information, including the pressure calibration method, pressure transmitting media and Rietveld analysis of the high-pressure neutron diffraction and revised our manuscript in accordance with your guidance. Please see below for the details of all the improvements.

Comments 1 (a): How was the pressure determined - the authors state using the known pressure load curve for the anvils and setup - it is well known that the pressure load performance will depend on packing, the hardness of the material and the PTM (as well as variability in the gasket etc). The experiment should have been performed with a pressure marker in the sample volume as is the normal for such experiments.

Author reply: Thanks for your comments. In most case of the high-pressure neutron diffraction experiment based on the Paris-Edinburgh (PE) Press, the cell pressure in the neutron diffraction experiment is estimated from the Edinburgh group calibration curve based on the applied oil pressure, which is considered as reliable and trustworthy¹⁻⁶.

This method establishes the relationship of the cell pressure and applied load (or hydraulic oil pressure) prior to the high-pressure experiment. The Edinburgh group calibration curve used in our neutron experiment is depicted in Figure R1. Therefore, the cell pressure can be estimated based on the known oil pressure and the Edinburgh group calibration curve¹. This pressure calibration method has been commonly used in high-pressure neutron experiments with PE press²⁻⁶. In addition, before measuring our sample, we have calibrated the cell pressure of high-pressure devices by measuring the EOS of pressure markers of Pb. However, there is a considerable overlap of the neutron diffraction patterns between Pb and the IPA sample, which is not conducive for the subsequent analysis of the structural evolution. Therefore, we can't load Pb together with the sample, and estimate the cell pressure via the Edinburgh group calibration curve and oil pressure. **In the present work, we have an important reason to confirm that the determined pressure value is reliable, and couldn't be affected by the packing state as follows:**

We have compared the P - V data derived from Rietveld refinements of angle-dispersive X-ray diffraction (ADXRD) and neutron diffraction patterns. The pressure calibration of the ADXRD experiments was determined utilizing the standard ruby fluorescent technique⁷. As shown in Supplementary Fig. 15, the cell volume evolution based on the neutron diffraction fits extremely well with the refined results of ADXRD data, confirming the reliability of this pressure calibration method in the neutron diffraction experiment.

The corresponding revised details are highlighted in red and could be found in **Lines 120-130** of the revised Supplementary Information.

Figure R1. The Edinburgh group calibration curve used in the present cell. The pressure in the sample chamber as a function of the oil pressure. The horizontal coordinate is the oil pressure, and the vertical coordinate is the actual pressure of the sample chamber.

Added Supplementary Fig. 15. Cell volume evolutions of IPA upon compression. Blue symbols represent the P - V data from ADXR experiment, and green symbols indicate the P - V data from the neutron diffraction experiment.

References:

1. Hattori, T. et al. Development of a Technique for High Pressure Neutron Diffraction at 40 GPa with a Paris-Edinburgh Press. *High Press. Res.* **2019**, *39*, 417-425.
2. Fitzgibbons, T. C. et al. Benzene-derived carbon nanothreads. *Nat. Mater.* **14**, 43-

47 (2014).

3. Li, X. et al. Mechanochemical Synthesis of Carbon Nanothread Single Crystals. *J. Am. Chem. Soc.* **139**, 16343-16349 (2017).
4. Zhang, P. et al. Distance-Selected Topochemical Dehydro-Diels–Alder Reaction of 1,4-Diphenylbutadiyne toward Crystalline Graphitic Nanoribbons. *J. Am. Chem. Soc.* **142**, 17662-17669 (2020).
5. Zhang, Z. et al. Thermal batteries based on inverse barocaloric effects. *Sci. Adv.* **9**, eadd0374 (2023).
6. Sano-Furukawa, A. et al. Direct observation of symmetrization of hydrogen bond in delta-AlOOH under mantle conditions using neutron diffraction. *Sci Rep* **8**, 15520 (2018).
7. Mao, H. K., Xu, J. & Bell, P. M. Calibration of the ruby pressure gauge to 800 kbar under quasi-hydrostatic conditions. *JGR: Solid Earth* **91** (1986).

Comments 1 (b): *The lack of pressure marker then leads to a further question - how do we know when you recover the gasket back to ambient conditions and determine the structure it is actually at ambient pressure. Unless the gasket is removed from the cell assembly (a detail missing if it is) a significant amount of residual pressure may be held by the press even with no applied load. Residual pressure will shorten the hydrogen bond length itself and there may well be significant hysteresis in the H-bond behaviour.*

Author reply: Thanks very much for your professional knowledge and constructive suggestions. The high-pressure neutron diffraction experiment was performed at BL11 PLANET in the Materials and Life Science Experimental Facility (MLF) at Japan Proton Accelerator Research Complex (J-PARC)¹. In the experiment, the PE press needs to be placed laterally with the aim of ensuring that the neutron beam passes through the cross-section of the PE press (Figure R2). Therefore, an oil pressure of 0.1 MPa was applied to hold the recovered sample within the PE Press. This is the lowest oil pressure, which does not have a great impact on the pressure of the sample.

[REDACTED]

Figure R2. a. VX4 PE Press equipped with the automatic hydraulic oil syringe pump.
b. VX4 PE Press^{1,2}.

In order to eliminate the possible influence of the residual pressure derived from the PE press, we further performed the neutron diffraction experiments of the recovered sample after removing the gasket. Therefore, the completely recovered data of the pressure-treated IPA was collected in Supplementary Fig. 19. Notably, the d -spacings of (033), (110) and (022) planes located at smaller values, which indicated the narrowed lattice spacings after pressure treatment. The orientations of (033) and (022) lattice planes are almost perpendicular to the direction of hydrogen bonds (Supplementary Fig. 17). Therefore, the reduced d -spacings of these almost vertical planes in the targeted IPA could confirm the result of strengthened hydrogen bonds after pressure treatment. Therefore, the hydrogen bonds in the targeted IPA are actually stronger than those ones before pressure treatment.

On the other hand, our IR spectra data clearly indicates the evolution of the hydrogen bond, which further proves the results from XRD and neutron diffraction. Based on the IR spectra (Figure R3), the stretching vibration of C=O bonds redshifted continuously below 20.0 GPa. This suggested that the O–H···O=C hydrogen bonds were continuously strengthened with the increase of pressure. During the decompression process, this vibration mode in IR spectra exhibited a relatively small blue shift (Figure R3). More importantly, after pressure was completely released, the C=O stretching vibrational mode in the pressure-treated IPA was redshifted by 6 cm^{-1} compared to the original state (Figure R4). This demonstrates that the increase of the pressure enhances the hydrogen bonds, and during the decompression process the

enhanced hydrogen bonds will experience some weakening but will not return to their original strength. Therefore, we could retain the enhanced hydrogen bonds down to the ambient conditions after pressure treatment.

The corresponding revised details are highlighted in red and can be found in Lines 200-201, 203-204, 208, 219, 227, 230-236, 418-431 of the revised manuscript, as well as Lines 152-159 in the revised Supplementary Information.

Supplementary Fig. 19. Neutron diffraction patterns of IPA-*d*₆ before and after pressure treatment.

Supplementary Fig. 17. **a** The lattice planes of (033) and (022) in IPA. **b** The lattice planes of (110) in IPA.

Figure R3. Selected IR spectra of IPA in the region of C=O stretching vibrational mode ($\nu(\text{C}=\text{O})$) upon compression and decompression. The right figure shows the corresponding evolution of hydrogen bonds.

Figure R4. IR spectra of IPA in the region of C=O stretching vibrational mode $\nu(\text{C}=\text{O})$ at ambient conditions and after pressure was released from 20.0 GPa.

References:

1. Hattori, T. et al. Design and Performance of High-Pressure PLANET Beamline at Pulsed Neutron Source at J-PARC. *Nuclear Instrum. Methods Phys. Res., Sect. A* **2015**, 780, 55-67.
2. Shi, Y., Chen, X.-P., Xie, L., Sun, G.-A. & Fang, L.-M. High-pressure neutron diffraction techniques based on Paris-Edinburgh press. *Acta Physica Sinica* **68** (2019).

Comments 1 (c): Pressure transmitting media - no details as to the pressure transmitting media that was used in the neutron experiment is given. The peaks in the diffraction pattern appear to broaden and hence suggest a non hydrostatic environment. Whilst this may not affect the change in the white light emission properties it will dramatically alter the quality of the diffraction data and hence the determined structural parameters and reliability of the fit etc.

Author reply: We appreciate for your comments. We apologize for the lack of details regarding the pressure-transmitting media (PTM) used in the neutron diffraction experiment. In fact, we did not employ a PTM in the high-pressure neutron diffraction experiment due to the specific challenges encountered. Initially, we attempted to utilize a PTM such as a mixture of methanol and ethanol (methanol: ethanol = 4:1), which is commonly used for PTM in high-pressure neutron diffraction experiments. However, this PTM would dissolve the sample and cause substantial alterations in its optical properties. As shown in Figure R5, the PL spectrum of the IPA with the PTM is significantly different from that of the original IPA. Therefore, we carried out the high-pressure neutron diffraction without PTM.

In addition, we have performed the in situ high-pressure PL spectra without PTM in Figure R6. When the pressure was completely released from ~20 GPa, the treated IPA still exhibited a white-light emission. In the previous study, we have explored the influence of PTM on the PL properties of IPA in Supplementary Fig. 5 and 6. **These results have indicated that the different PTMs (silicone oil, liquid argon and liquid nitrogen) could not influence the present conclusions: the PL enhancement under high pressure and the harvest of white-light emission after pressure treatment.**

Therefore, although the absence of the PTM will bring a certain error, considering

the negative impact of the PTM on IPA, this operation is more advantageous and reliable for the subsequent structural analysis.

The revised details are highlighted in red and can be found in Lines 410-411 of the revised manuscript.

Figure R5. PL spectra of IPA with (methanol: ethanol = 4:1, green line) and without PTM (blue line).

Figure R6. **a** PL spectra of IPA without PTM upon compression to 10.0 GPa. **b** PL spectra of IPA upon compression from 10.0 GPa to 20.3 GPa. **c** PL spectra of IPA upon decompression. **d** the chromaticity coordinates of IPA after pressure treatment (0.28, 0.35).

Comments 1 (d): It appears that only the longer time window of the PLANET instrument has been presented - what about the data in the lower t -o-f window? In this d -spacing range is where a significant amount of the detailed structural information is to be found when performing Rietveld analysis on a powder diffraction data set.

Author reply: Thanks for the reviewer's comments. According to your suggestions, the neutron diffraction data in the lower d -spacing range has been provided in the revised supplementary Fig. 13. In addition, we also revised the related Rietveld refinements in the Supplementary Fig. 16. The revised details are highlighted in red and can be found in **Lines 105, 132-133** of the revised Supplementary Information.

Revised Supplementary Fig. 13. Time-of-flight (TOF) neutron diffraction patterns of IPA- d_6 upon compression.

Comments 2: Regarding the presentation of the data

Comments 2: (a) I can understand that only the two ambient pressure data sets are shown in the main manuscript but more should be shown in the SI - show all of the high

pressure data sets (likewise provide all of the cif files for the high pressure data sets).
(b) Where are the fits to the *in situ* pressure data sets - these need to be shown to allow the reader to evaluate the work experimentally.

Author reply: We acknowledge your insightful suggestions very much, which are valuable in improving the quality of our manuscript. In our original Supplementary Information, we have provided *in situ* high-pressure neutron diffraction data in Supplementary Fig. 13. According to your suggestions, we added the *in situ* high-pressure Rietveld refinement plot in the revised Supplementary Fig. 16. In addition, the high-pressure cif data were also provided in the revised Supplementary Information.

The revised details are highlighted in red and can be found in Lines 422-423 of the revised manuscript and Lines 132-133, 190-579 of the revised Supplementary Information.

Revised Supplementary Fig. 16. Rietveld refinement plot of IPA-d₆ upon compression and decompression.

Comments 2 (c): *There are typos in Supp Table 2 - see the c vs beta angle for the Released data set...*

Author reply: Thanks very much for your professional knowledge and great patience in spotting the mistakes. We have revised the beta angle for the Released data in Supplementary Table 2. In addition, we also tried our best to check the whole manuscript and Supplementary Information to make them concise and clarity. The revised details are highlighted in red and can be found in **Line 177** of the revised Supplementary Information.

Comments 2 (d): *I assume that Supp Table 3 was determined by neutron diffraction data. The determined values are very low in precision - how were the values determined? Directly from the refined structure or from lattice plane spacing?*

Author reply: Thanks for your comments. The data in Supplementary Table 3 were directly derived from the refined structure.

In the present work, the Rietveld refinements of the neutron diffraction patterns were performed using the General Structure Analysis System II (GSAS II) software. The final structure refinement turned out to converge successfully with low R and GOF values (Supplementary Fig. 16). Therefore, we collected the structural information at various pressures based on these refinements, including the lattice parameters, intermolecular hydrogen-bond distance (d_1 and d_2), intermolecular the C \cdots C distance ($d_{(C\cdots C)}$) and the parallel misalignment angle (σ). Furthermore, the error values of each structural parameter were also provided by GSAS II, reflecting the precision is acceptable and trustworthy, which is also consistent with previous work^{1,2}.

References:

1. Kloß, S. D. et al. Preparation of iron(IV) nitridoferrate Ca_4FeN_4 through azide-mediated oxidation under high-pressure conditions. *Nat. Commun.* **12** (2021).
2. Lokshin, K. A. et al. Structure and dynamics of hydrogen molecules in the novel clathrate hydrate by high pressure neutron diffraction. *Phys. Rev. Lett.* **93**, 125503 (2004).

Comments 2 (e): *What are the figures of merit for the fits?*

Author reply: Thanks for the helpful comments. According to your suggestion, we have included the R_{wp} and GOF values in the revised Rietveld Supplementary Fig. 16 (also shown in the above figure). The revised details are highlighted in red and can be found in **Lines 132-133** of the revised Supplementary Information.

Overall, I think that whilst the experiment of the high pressure neutron may have been performed well - the current presentation of results do not do it justice and by doing so actually leave the reader to ask questions about their validity and analysis. More information and robustness is required to justify publication in its current form.

Author reply: Thanks for your kind comments with the positive affirmation of our high-pressure neutron diffraction experiments. Following your kind suggestions, we carefully rechecked our manuscript, added the corresponding structural data and made some amendments to respond to the reviewer's issues. Hopefully, we have addressed all the concerns of the reviewer. The corresponding revised details are highlighted in red and can be found in **Lines 188, 200-201, 203-204, 208, 219, 227, 230-236, 410-411, 418-431** of the revised manuscript, as well as **Lines 95, 105, 120-130, 132-133, 152-159, 177, 190-579** in the revised Supplementary Information.

Reply to Reviewer #6

The Editor asked me to evaluate quality and validity of quantum-chemical simulations of characterizing energy and properties of singlet and triplet states supporting experimental data. Upon inspection of the revised MS, SI and the response letter, I find the following:

1. An addition of quantum-chemical simulations analyzing the effects of SOC (spin-orbit coupling), HFC (hyperfine coupling) and ΔE_{ST} (singlet-triplet gap) has significantly substantiated and supported the original hypothesis that hydrogen bond enhancement is the main cause of phosphorescence enhancement. These calculations nicely demonstrate an increased effective mixture of $n\text{-}\pi^$ and $\pi\text{-}\pi^*$ character that likely leads to more effective intersystem crossing owing to El-Sayed rule. Increased effective amount of SOC favors this as well. Calculations also do not find any substantial influence of HFC in the pressure-treated samples. Finally, calculations point out to the uniform decrease of first singlet (S_1) to several triplet states gaps pointing to stabilization of the resulting triplet states. Overall, presenting quantum chemical modeling added value to science and improved the article by invoking basic quantum-mechanical analysis and establishing connections to known physical modes.*

2. The authors have used Gaussian 09 for DFT and TDDFT calculations of a model system (dimer). B3LYP/6-31G(d, p) model chemistry was invoked, which I find appropriate for these simulations (indeed, NTOs do not show traces of charge-transfer character). Calculations were followed by NTO analysis which is an appropriate methodology to examine the orbital character for TDDFT derived electronic states. Finally, the SOC coefficients were evaluated with Beijing density function(BDF) program. I am not familiar and have not used this software, however, by examining their webpage I found BDF to be adequate for this purpose.

3. While I am not questioning the results of quantum-mechanical modeling, I would make several suggestions to improve computational work mostly answering question on how stable are the identified trends and what is an uncertainty of the derived numbers with respect to the chosen DFT framework. These additional results may be placed in SI with some brief referencing from the main text.

Author reply: Thank you very much for your thorough evaluation and insightful comments on our manuscript. We truly appreciate the time and effort you dedicated to reviewing our work. Your suggestions for further improving the computational work are invaluable and important. We have added additional calculated results in the Supplementary Information to provide a more comprehensive understanding of the stability of identified trends and the uncertainties associated with our calculations.

Comments 1: The authors are using a model dimer system, neglecting other molecules and generally dielectric surrounding. I would suggest to check their results against a single point calculation of 4 molecules (x2) size and against a single point calculation of a dimer within polarizable continuum model (PCM). These things are standard within Gaussian software and will not take a lot of computational power.

Author reply: Thanks for your professional knowledge and constructive suggestions. In the present work, the IPA material we studied is a crystal sample, so we do not use the PCM model in the calculation. Therefore, following your suggestions, we have added the single-point calculations of the dimer model and 4 molecules (x2) size in order to check our results, as shown in below Figure R1 and Table R1. Their energies after pressure treatment were both higher than those before pressure treatment, which indicates that their trends are consistent. Therefore, considering the other molecules and generally dielectric surroundings, the IPA is still in a metastable state after pressure treatment. In addition, we also revised our calculation of SOC coefficients using 4 molecules models (Table R2).

The revised details are highlighted in red and can be found in **Lines 238, 263-266** of the revised manuscript and **Lines 161, 181-182** of the revised Supplementary Information.

Figure R1. 4-molecules models at 1 atm and after pressure treatment.

Table R1. The energies of dimer and 4 molecules before and after pressure treatment.

Energy (eV)	dimer	4 molecules
1 atm	-33165.50	-66331.52
Released	-33165.27	-66331.15

Table R2. The SOC constants (ζ) of the pristine and pressure-treated IPA.

SOC coefficients (ζ)	1atm	Released
S_1 to T_1	0.73 cm^{-1}	1.92 cm^{-1}
S_1 to T_2	5.42 cm^{-1}	19.35 cm^{-1}
S_1 to T_3	13.64 cm^{-1}	18.55 cm^{-1}
S_1 to T_4	5.27 cm^{-1}	11.34 cm^{-1}
S_1 to T_5	28.53 cm^{-1}	29.18 cm^{-1}
S_1 to T_6	0.08 cm^{-1}	7.71 cm^{-1}

Comments 2: A more subtle point is a value of singlet-triplet gap: ΔE_{ST} . I would remind that the value of this gap in the TDDFT framework depends a lot on the fraction of the orbital exchange present in the DFT model used. For example, B3LYP uses 20% of this exchange. I would expect that models with a larger fraction of exchange (like asymptotically corrected functionals) would lead to much larger ΔE_{ST} .

E_{ST} , whereas models with lower fractions (like GGA functionals) will lead to very small gaps. While B3LYP could be a reasonable compromise, I would suggest to make additional estimates of ΔE_{ST} using i) several other functionals (e.g. HSE, PBE0, ω B97X) to estimate uncertainty of numbers and ii) using Delta SCF approach (different from TDDFT) to evaluate ΔE_{ST} (S_1-T_1). Delta SCF should be a lot more stable with respect to the choice of DFT model compared to TDDT method (but it will not suit for estimation of ΔE_{ST} ($S_1-T_{5/6}$) gaps). Again these are trivial single point calculations that can be performed within a few days. Altogether I believe that these minor additional simulations will make the currently present modeling more solid and trustable.

Altogether I believe that these minor additional simulations will make the currently present modeling more solid and trustable.

Author reply: We highly appreciate you have proposed insightful suggestions. In order to estimate the uncertainty of the excited state energy, we have performed the calculations using other functionals (e.g. HSE, PBE0, ω B97X) that you mentioned. In addition, with the aim to take more account of the correction of hydrogen bonds, we further add additional calculations using CAM-B3LYP functionals. The results are displayed in the below Table R3-R6.

After pressure treatment, the treated IPA displays various degrees of lowered energy levels, which is consistent with results using B3LYP function as well as the experimental results. The energy values of the excited states calculated by these methods (e.g. B3LYP, HSE, PBE0) gradually increase ($B3LYP < HSE < PBE0$), but the energy gap is not distinctly pronounced. Furthermore, results from ω B97X calculations tend to overestimate the excited state energies and ΔE_{ST} , possibly derived from greater consideration of long-range correction.

As pointed out by the reviewer, $\Delta E(S_1-T_1)$ was recalculated using Δ SCF approach in comparison with TDDFT results. Because this result depends on functional selection, we consider the functional of different exact exchange components for comparison¹. At the same time, we also take Range-Separated functional into consideration, because the factors of long-range correction also affect the accuracy of

$\Delta E(S_1-T_1)$. The different Range-Separated functional adjusts the difference of μ values according to the following formula, r_{12} being the interelectronic distance². The calculation results are displayed in Table R7.

$$\frac{1}{r_{12}} = \frac{1 - [\alpha + \beta \cdot \text{erf}(\mu r_{12})]}{r_{12}} + \frac{\alpha + \beta \cdot \text{erf}(\mu r_{12})}{r_{12}}$$

From the results above, the values of $\Delta E(S_1-T_1)$ calculated by this method are greater than the results of TDDFT, but the gap is not obvious. Furthermore, as the rate of exact exchange components increases, estimates of $\Delta E(S_1-T_1)$ values are also regionally stable around 1.2 eV. This is very close to the result when long range correction is taken into account. The current calculation results show that the M062X results underestimate $\Delta E(S_1-T_1)$. While the results of $\Delta E(S_1-T_1)$ calculation after considering the weak interaction of the system are very robust, and basically reproduce all our explanations in this paper.

By integrating all the calculation results, we have adopted the calculation results of TDDFT based on CAM-B3LYP. We also have updated figures 3 and 4 in the revised manuscript

The revised details are highlighted in red and can be found in Lines 274, 309-311, 436, 438 of the revised manuscript and Lines 184-188 of the revised Supplementary Information.

Revised Fig. 3 | The SOC and NTOs of the pristine and pressure-treated IPA. a The calculated ξ ($\xi(S_1-T_6)$, $\xi(S_1-T_5)$, $\xi(S_1-T_4)$, $\xi(S_1-T_3)$, $\xi(S_1-T_2)$, $\xi(S_1-T_1)$) for the pristine and pressure-treated IPA. **b** The NTOs of T₅ and T₆ in the pristine and treated IPA. **c** Schematic representation of hydrogen bonds acting on ISC process.

Revised Fig. 4 | The effects of HFC and ΔE_{ST} on ISC process of IPA before and after pressure treatment. **a** Magnetic-field effects on the PL intensity of IPA. The data at 1 atm and released represent the mean value from five and three experiments, respectively. **b** The energies of singlet and triplet states of IPA before and after pressure treatment. **c** The calculated ΔE_{ST} ($\Delta E(S_1-T_6)$, $\Delta E(S_1-T_5)$, $\Delta E(S_1-T_4)$, $\Delta E(S_1-T_3)$, $\Delta E(S_1-T_2)$, $\Delta E(S_1-T_1)$) for the pristine and targeted IPA. **d** Proposed energy transfer processes for fluorescence and phosphorescence in IPA before and after pressure treatment.

Table R3. The energies of singlet and triplet states of the pristine IPA by using different functions.

Function \ Energy	S ₁ (eV)	T ₁ (eV)	T ₂ (eV)	T ₃ (eV)	T ₄ (eV)	T ₅ (eV)	T ₆ (eV)
B3LYP	4.55	3.43	3.44	3.47	3.48	4.07	4.08
HSE	4.63	3.27	3.28	3.30	3.31	4.08	4.09
PBE0	4.67	3.29	3.30	3.32	3.33	4.10	4.11
ω B97X	5.13	3.26	3.27	3.30	3.31	4.34	4.35
CAM-B3LYP	4.62	3.44	3.48	3.65	3.72	3.83	3.96

Table R4. The energies of singlet and triplet states of the pressure-treated IPA by using different functions.

Function \ Energy	S ₁ (eV)	T ₁ (eV)	T ₂ (eV)	T ₃ (eV)	T ₄ (eV)	T ₅ (eV)	T ₆ (eV)
B3LYP	3.63	2.47	2.48	2.60	2.61	3.44	3.45
HSE	3.69	2.32	2.33	2.44	2.45	3.45	3.46
PBE0	3.82	2.33	2.34	2.45	2.46	3.48	3.49
ω B97X	4.48	2.23	2.24	2.34	2.35	3.77	3.79
CAM-B3LYP	3.59	2.39	2.54	3.17	3.31	3.46	3.51

Table R5. The $\Delta E(S_1-T_n)$ of the pristine IPA by using different functions.

Energy Function	$\Delta E(S_1-T_1)$ (eV)	$\Delta E(S_1-T_2)$ (eV)	$\Delta E(S_1-T_3)$ (eV)	$\Delta E(S_1-T_4)$ (eV)	$\Delta E(S_1-T_5)$ (eV)	$\Delta E(S_1-T_6)$ (eV)
B3LYP	1.12	1.11	1.08	1.07	0.48	0.47
HSE	1.36	1.35	1.33	1.32	0.55	0.54
PBE0	1.38	1.37	1.35	1.34	0.57	0.56
ω B97X	1.87	1.86	1.83	1.82	0.79	0.78
CAM- B3LYP	1.18	1.14	0.97	0.90	0.79	0.66

Table R6. The $\Delta E(S_1-T_n)$ of the pressure-treated IPA by using different functions.

Energy Function	$\Delta E(S_1-T_1)$ (eV)	$\Delta E(S_1-T_2)$ (eV)	$\Delta E(S_1-T_3)$ (eV)	$\Delta E(S_1-T_4)$ (eV)	$\Delta E(S_1-T_5)$ (eV)	$\Delta E(S_1-T_6)$ (eV)
B3LYP	1.16	1.15	1.03	1.02	0.19	0.18
HSE	1.37	1.36	1.25	1.24	0.24	0.23
PBE0	1.49	1.48	1.37	1.36	0.34	0.33
ω B97X	2.25	2.24	2.14	2.13	0.71	0.69
CAM- B3LYP	1.20	1.05	0.42	0.28	0.13	0.08

Table R7. $\Delta E(S_1-T_1)$ values under Δ SCF approach.

Functionals with the exact exchange rate	$\Delta E(s_1-t_1)$ values under Δ SCF approach
B3LYP (20%)	1atm: 1.191 eV
	Released: 1.120 eV
TPSSh (10%)	1atm: 1.173 eV
	Released: 1.172 eV
PBE0 (25%)	1atm: 1.197 eV

	Released: 1.199 eV
M06 (27%)	1atm: 1.205 eV
	Released: 1.209 eV
M062X (54%)	1atm: 1.188 eV
	Released: 1.197 eV
ω B97X ($\mu=0.3$)	1atm: 1.194 eV
	Released: 1.211 eV
ω B97X-D ($\mu=0.2$)	1atm: 1.191 eV
	Released: 1.207 eV
CAM-B3LYP ($\mu=0.33$)	1atm: 1.199 eV
	Released: 1.205 eV

References:

1. Wang, Y., Jin, X., Yu, H. S., Truhlar, D. G. & He, X. Revised M06-L functional for improved accuracy on chemical reaction barrier heights, noncovalent interactions, and solid-state physics. *PNAS* **114**, 8487-8492 (2017).
2. Santra, G., Calinsky, R. & Martin, J. M. L. Benefits of Range-Separated Hybrid and Double-Hybrid Functionals for a Large and Diverse Data Set of Reaction Energies and Barrier Heights. *J. Phys. Chem. A* **126**, 5492-5505 (2022).

In summary, following the suggestions and comments of the reviewer, we carefully rechecked our manuscript and made some amendments to respond to the reviewer's issues. Hopefully, we have addressed all of your concerns. The corresponding revised details are highlighted in red and could be found in Lines 5-6, 16, 19-20, 29-30, 34-39, 188, 200-201, 203-204, 208, 219, 227, 230-236, 238, 258, 263-266, 274, 296, 309-311, 364, 388, 402, 410-411, 418-431, 434, 436, 438, 439, 445-447, 569, 582-589, 608-613, 615-622 of the revised manuscript, as well as Lines 7-8, 18, 21-22, 95, 105, 117, 120-130, 132-133, 146, 150, 152-159, 161, 162, 166, 173, 177, 181-579, 590-594 of the revised Supplementary Information.

REVIEWER COMMENTS

Reviewer #5 (Remarks to the Author):

The authors have made considerable effort to address the concerns of the referees.

A few final comments.

a) As far as I understand a pressure load curve is not an accurate and reliable method to understand the pressure load behaviour of a PE press given the different hardness of materials and the effect of packing has on the pressure performance. A pressure marker should be used especially considering point b below.

b) To believe that at the low pressure applied by the press at recovery to give negligible pressure is not true - experience and publications suggest that hysteresis of the PE press and gasket behaviour means that significant pressure is held even at low pressure.

c) The authors should address the fact that the diffraction peaks are broadened and may give misleading results by performing a Rietveld fit to the data sets (ADP and atomic position determination) where this is observed and in particular the determined bond distances.

Reviewer #6 (Remarks to the Author):

The authors adequately responded to my comments (Reviewer 6) and made a fair attempt to expand and substantiate theoretical modeling. The revision benefitted from updated figures and additional summaries in SI. A minor comment is that something like polarizable continuum model (PCM) is a very cheap computational alternative to evaluate the dielectric effect of crystal environment vs simulations in vacuum. Overall, the main conclusions coming from additional atomistic simulations of electronic structure remain essentially the same to the previous modeling results, and continue supporting scenarios drawn from experimental part.

I would also say that Reviewer 5 brought several very important comments on the experimental part. Here I defer evaluation of the revision to the expert.

For your guidance, itemized response to reviewer' s comments is appended below.

Reply to Reviewer #5

The authors have made considerable effort to address the concerns of the referees.

A few final comments

a) As far as I am understand a pressure load curve is not an accurate and reliable method to understand the pressure load behaviour of a PE press given the different hardness of materials and the effect of packing has on the pressure performance. A pressure marker should be used especially considering point b below. b) To believe that at the low pressure applied by the press at recovery to give negligible pressure is not true - experience and publications suggest that hysteresis of the PE press and gasket behaviour means that significant pressure is held even at low pressure. c) The authors should address the fact that the diffraction peaks are broadened and may give misleading results by performing a Rietveld fit to the data sets (ADP and atomic position determination) where this is observed and in particular the determined bond distances

Author reply: Thank you so much for taking lots of time to revise our manuscript! We are really appreciated all your comments and suggestions, which have improved our work greatly. According to your suggestions, we have conducted additional high-pressure neutron experiments incorporating the pressure marker (Pb) and pressure-transmitting medium (Ar). Meanwhile, the lattice parameters, atomic positions, and structural information were also provided in the revised manuscript and Supplementary Information.

In these experiments, the pressure marker Pb was used to accurately monitor pressure changes. After releasing the pressure, the diffraction peaks of Pb returned to their original positions (Fig. 2d), further confirming the complete pressure release in the recovered IPA. Before the high-pressure experiment, the fresh sample with a Pb marker was loaded into a vanadium tube with a diameter of 6 mm to collect the neutron diffraction data. The data were used as an ambient-pressure

reference. For the high-pressure experiment in PE cell with Pb and Ar, the pressure was calibrated by the equation of state (EOS) of Pb (Table R1)¹. After pressure was completely released, the peaks of Pb moved back to the original state, just the same as the ambient-pressure reference (diffraction pattern shown in Fig. 2d). Notably, the d -spacings of (033), (110), and (022) planes become smaller after pressure treatment, indicating a narrowing of lattice spacings in the recovered IPA. In addition, we also compared these new experimental results with the neutron diffraction spectrum of the pressure-treated sample after removing the gasket. As shown in Supplementary Fig. 17, the consistent shift in the (033), (110), and (022) planes confirms the reliability of our results. Considering that the orientations of (033) and (022) lattice planes are almost perpendicular to the direction of hydrogen bonds (Supplementary Fig. 18), these reduced d -spacings undoubtedly prove the mechanism we proposed that the strengthened hydrogen bonds after pressure treatment.

In order to eliminate your concern about the peak broadening caused by the possible pressure gradient, we utilize a pressure-transmitting medium (PTM) of Ar in this high-pressure neutron diffraction experiment (Figure R1 b). The addition of the PTM (Ar) improves the broadening of neutron diffraction peaks under high pressure (Figure R1). Based on this experimental run, we did Rietveld refinements (Supplementary Fig. 16) and recalibrated the lattice parameters and atomic positions (Supplementary Tables 2 and 3). As shown in Fig. 2e and f, the results are consistent with our previous conclusions: the hydrogen bonds strengthened gradually with the increase of pressure. The reduced parallel misalignment angle (σ) indicated a lessened spatial overlap of the π -stacked motif upon compression. After pressure was completely released, the D₁₇···O₉ (d_1) of the recovered sample decreased from 1.73(4) Å to 1.49(5) Å and D₂₁···O₅ (d_2) decreased from 1.77(4) Å to 1.69(5) Å, indicating the enhanced hydrogen bonds. And the σ decreased from 67.7(8) to 65.7(9)°, rendering a lessened spatial overlap of the π -stacked motif in the targeted IPA. **Therefore, we retain the enhanced hydrogen bonds and misaligned π -stacked motif down to the ambient conditions after pressure treatment.**

Moreover, we have to restate that our IR spectra data clearly indicates the

evolution of the hydrogen bond, which further proves the results from XRD and neutron diffraction. Based on the IR spectra (Figure R2), the stretching vibration of C=O bonds redshifted continuously below 20.0 GPa. This suggested that the O–H···O=C hydrogen bonds were continuously strengthened with the increase of pressure. During the decompression process, this vibration mode in IR spectra exhibited a relatively small blue shift (Figure R2). More importantly, after pressure was completely released, the C=O stretching vibrational mode in the pressure-treated IPA was redshifted by 6 cm⁻¹ compared to the original state (Fig. 2a). This demonstrates that the increase of the pressure enhances the hydrogen bonds, and during the decompression process, the enhanced hydrogen bonds will experience some weakening but will not return to their original strength. Therefore, we could retain the enhanced hydrogen bonds down to the ambient conditions after pressure treatment.

The corresponding revised details are highlighted in red and can be found in Lines 188, 193-197, 206-212, 215, 226-227, 230, 232, 234, 406, 408-410, 413-414, 416-419, 423-424, 441-442, 580-581 of the revised manuscript, as well as Lines 104, 106-108, 117, 121-122, 124-129, 140-146, 152-153, 173-177, 187-579, 585-586 in the revised Supplementary Information.

Revised Fig. 2 | Crystal structure evolution upon compression and decompression.

a IR spectra of IPA in the region of C=O stretching vibrational mode $\nu(\text{C}=\text{O})$ at ambient

conditions and after pressure was released from 20.0 GPa. **b** Molecular packing along the b axis (left). The schematic diagram of the parallel misalignment angle (σ) of IPA (right). **c** The compression rate of three lattice constants (a , b , c) at different pressures. **d** Time-of-flight (TOF) neutron diffraction patterns of IPA- d_6 . The TOF neutron diffraction patterns of the pristine (blue line) and recovered (magenta line) samples were collected at BL11 PLANET in the MLF at J-PARC. The asterisks indicate the peaks of the Pb marker. The neutron diffraction patterns of the pristine (green line) and recovered (orange line) samples were collected at High pressure neutron diffractometer (Fenghuang) at CMRR neutron science platform. **e** Pressure-dependent hydrogen-bond distances $D_{17}\cdots O_9$ (d_1) and $D_{21}\cdots O_5$ (d_2) evolution of IPA. **f** Pressure-dependent σ evolution of IPA. The d_1 , d_2 , and σ of IPA- d_6 were determined by Rietveld refinement of neutron diffraction patterns.

Revised Supplementary Fig. 17. Neutron diffraction patterns of IPA- d_6 before and after pressure treatment. The neutron diffraction patterns of the pristine (green line) and recovered (orange line) samples were collected at High pressure neutron diffractometer (Fenghuang) at China Mianyang Research Reactor's (CMRR) neutron science platform. The neutron diffraction pattern of the recovered (blue line) sample was collected at BL11 PLANET in the Materials and Life Science Experimental Facility (MLF) at Japan Proton Accelerator Research Complex (J-PARC). The black asterisks indicate the peaks of the Pb marker.

Revised Supplementary Fig. 18. **a** The lattice planes of (033) and (022) in IPA. **b** The lattice planes of (110) in IPA.

Figure R1. **a** Time-of-flight (TOF) neutron diffraction patterns of IPA- d_6 upon compression without pressure marker and PTM. **b** Time-of-flight (TOF) neutron diffraction patterns of IPA- d_6 upon compression with pressure marker of Pb and PTM of Ar. The black asterisks indicate the peaks of the Pb marker. The magenta asterisks indicate the peaks of the Ar². The blue asterisks indicate the peaks of the diamond. The pressure was estimated based on the EOS of Pb.

Revised Supplementary Fig. 16. Rietveld refinement plot of IPA- d_6 upon compression and decompression. Before the high-pressure experiment, the fresh sample with a Pb marker was loaded into a vanadium tube with a diameter of 6 mm to collect the neutron diffraction data. The data were used as an ambient-pressure reference. The high-pressure neutron diffraction patterns were collected by loading the sample with Pb and Ar in VX4 Paris-Edinburgh (PE) Press.

Figure R2. Selected IR spectra of IPA in the region of C=O stretching vibrational mode ($\nu(\text{C}=\text{O})$) upon compression and decompression. The right figure shows the corresponding evolution of hydrogen bonds.

Table R1. Lattice parameters of Pb upon compression. The pressure was calibrated by the EOS of Pb

Pressure (GPa)	a (Å)	volume (Å ³)
1 atm	4.96330	122.268
1.5	4.90930	118.320
3.4	4.84392	114.361
7.7	4.75302	107.377
12.3	4.67040	101.874
16.3	4.61168	98.079
19.8	4.56770	95.300

Revised Supplementary Table 2. Lattice parameters of IPA upon compression and decompression. These values were determined by Rietveld refinement using the GSASII software package based on neutron diffraction data.

Pressure (GPa)	a (Å)	b (Å)	c (Å)	β (°)	$\alpha=\gamma$ (°)
1 atm	3.770	16.404	11.766	90.155	90.00
1.5	3.528	16.268	11.136	88.439	90.00
3.4	3.421	16.128	10.847	87.404	90.00
7.7	3.283	15.931	10.466	85.631	90.00
12.3	3.164	15.716	10.2406	83.685	90.00
Released	3.755	16.283	11.741	89.800	90.00

Revised Supplementary Table 3. The d_1 , d_2 , σ , and $d_{(C\cdots C)}$ values of IPA upon compression and decompression.

Pressure (GPa)	d_1	d_2	σ (°)	$d_{(C\cdots C)}$
1 atm	1.73(4)	1.77(4)	67.7(8)	3.77(5)
1.5	1.67(4)	1.71(5)	63.5(7)	3.53(4)
3.4	1.49(3)	1.54(3)	62.4(7)	3.42(4)
7.7	1.45(3)	1.48(3)	62.0(6)	3.28(3)
12.3	1.37(4)	1.40(5)	60.7(9)	3.16(4)
Released	1.49(5)	1.69(5)	65.7(9)	3.75(5)

References:

1. Vohra, Y. K. & Ruoff, A. L. Static compression of metals Mo, Pb, and Pt to 272 GPa: Comparison with shock data. *Phys. Rev. B* **42**, 8651-8654 (1990).
2. Dewaele, A. et al. Stability and equation of state of face-centered cubic and hexagonal close packed phases of argon under pressure. *Sci. Rep.* **11** (2021).

Reply to Reviewer #6

The authors adequately responded to my comments (Reviewer 6) and made a fair attempt to expand and substantiate theoretical modeling. The revision benefitted from updated figures and additional summaries in SI. A minor comment is that something like polarizable continuum model (PCM) is a very cheap computational alternative to evaluate the dielectric effect of crystal environment vs simulations in vacuum. Overall, the main conclusions coming from additional atomistic simulations of electronic structure remain essentially the same to the previous modeling results, and continue supporting scenarios drawn from experimental part.

Author reply: Thank you very much for your positive assessment of the revisions. In order to further evaluate the results, we have added the single-point calculations of the dimer model and 4 molecules (x2) size using PCM methods (B3-GD3 function), where n-hexane (HEX), diethyl ether (ETE), tetrahydrofuran (THF) and acetonitrile were utilized as the solvent environment. As shown in below Table R1 and R2, their energies after pressure treatment were both higher than those before pressure treatment. Furthermore, the energy of the pressure-treated IPA using different PCM methods calculated under the dimer model is about 0.2 eV higher than that of the pristine IPA. Based on the 4 molecules (x2) model, the estimates of the energy differences were stable around ~0.4 eV. Therefore, considering the generally dielectric surroundings, the IPA is still in a metastable state after pressure treatment.

Table R1. The energies of dimer in different solvent environments before and after pressure treatment.

	1 atm	Released
Dimer-vacuum	-33165.50 eV	-33165.27 eV
PCM-HEX	-33160.80 eV	-33160.60 eV
PCM-ETE	-33160.96 eV	-33160.79 eV
PCM-THF	-33161.03 eV	-33160.86 eV
PCM-acetonitrile	-33161.11 eV	-33160.96 eV

Table R2. The energies of 4 molecules in different solvent environments before and after pressure treatment.

	1 atm	Released
4 molecules-vacuum	-66331.52 eV	-66331.15 eV
PCM-HEX	-66322.56 eV	-66322.07 eV
PCM-ETE	-66322.88 eV	-66322.44 eV
PCM-THF	-66323.03 eV	-66322.60 eV
PCM-acetonitrile	-66323.21 eV	-66322.81 eV

In summary, following the suggestions and comments of the reviewer, we carefully rechecked our manuscript and made some amendments to respond to the reviewer's issues. Hopefully, we have addressed all of your concerns. The corresponding revised details are highlighted in red and could be found in Lines 188, 193-197, 206-212, 215, 226-227, 230, 232, 234, 406, 408-410, 413-414, 416-419, 423-424, 441-442, 580-581 of the revised manuscript, as well as Lines 104, 106-108, 117, 121-122, 124-129, 140-146, 152-153, 173-177, 187-579, 585-586 in the revised Supplementary Information.

REVIEWERS' COMMENTS

Reviewer #5 (Remarks to the Author):

I thank the authors for their additional experiments - which have greatly improved the quality and reliability in the diffraction results.

The study is greatly improved and is now of a high standard and the results support sufficiently the claims made.

Reviewer #6 (Remarks to the Author):

The authors have adequately responded to my comments. I am fine with publication of this article in Nature Communications.